# PCA of Probability Measures: Sparse and Dense Sampling Regimes

Erell Gachon [1]   Jérémie Bigot [1]   Elsa Cazelles [2]

## Abstract

A common approach to perform PCA on probability measures is to embed them into a Hilbert space where standard functional PCA techniques apply. While convergence rates for estimating the embedding of a single measure from $m$ samples are well understood, the literature has not addressed the setting involving multiple measures. In this paper, we study PCA in a double asymptotic regime where $n$ probability measures are observed, each through $m$ samples. We derive convergence rates of the form $n^{-1/2} + m^{-\alpha}$ for the empirical covariance operator and the PCA excess risk, where $\alpha > 0$ depends on the chosen embedding. This characterizes the relationship between the number $n$ of measures and the number $m$ of samples per measure, revealing a sparse (small $m$) to dense (large $m$) transition in the convergence behavior. Moreover, we prove that the dense-regime rate is minimax optimal for the empirical covariance error. Our numerical experiments validate these theoretical rates and demonstrate that appropriate subsampling preserves PCA accuracy while reducing computational cost.

## 1. Introduction

In this paper, we are interested in Principal Component Analysis (PCA) of a set of probability measures supported on $\mathbb{R}^d$. Such data models arise naturally in various applied fields, like signal and image processing, computer vision, and computational biology (Khamis et al., 2024; Kolouri et al., 2017; Montesuma et al., 2024). The goal of PCA of probability measures is to identify the main modes of variability, with outputs analogous to classical PCA : principal directions, component scores and modes of variation. These outputs enable low-dimensional representation, interpreta-

tion of variability and feature extraction for downstream tasks such as classification. In standard Euclidean settings, PCA relies on the eigenspectrum and eigenvectors of the covariance matrix of the data. Extending this idea to probability measures therefore requires a meaningful notion of covariance between random measures. A common strategy is to embed a given random measure $\mu$ into a Hilbert space $\mathcal{H}$ through a map $\Phi$. In this embedded space, the covariance operator

$$\Sigma = \mathbb{E}[\Phi(\mu) \otimes \Phi(\mu)] \tag{1}$$

becomes the fundamental descriptor of variability.

We will focus on three specific embeddings of probability measures in Hilbert spaces. The *Kernel Mean Embedding* (Muandet et al., 2017) relies on kernel methods to map a probability measure into a *Reproducing Kernel Hilbert Space* (RKHS). Another approach is the *Linearized Optimal Transport* (LOT) embedding (Wang et al., 2013), which leverages the Riemannian-like geometry of the Wasserstein space. Finally, the *Sliced Wasserstein* (SW) (Rabin et al., 2011) embedding, based on the Sliced Wasserstein distance, has recently gained popularity due to its computational efficiency and its ability to capture geometric features of probability measures. Most existing statistical results on these embeddings (Manole et al., 2024; Berlinet & Thomas-Agnan, 2011; Chewi et al., 2024) focus on the classical setting where one estimates $\Phi(\mu)$ from $m$ samples drawn from a *single* probability measure $\mu$. However, many modern applications involve a collection of $n$ probability measures rather than a single one, calling for a statistical framework which accounts both intra- and inter-measure variability.

A concrete example of data naturally represented as a set probability distributions arises in biology through flow cytometry, where each patient corresponds to a distribution of hundreds of thousands of cells, see experiments in Section 4.2. Here, while the number of patients $n$ is fixed, each patient contributes a distribution composed of a large number $m$ of cells, highlighting the need for computationally efficient strategies such as subsampling within each distribution. This naturally leads to the question: how many points per measure can be safely retained without significantly degrading PCA performance?

Motivated by this application, we consider $n$ measures $\mu_1, \cdots, \mu_n$ that can be viewed as independent copies of

---

[1]Institut de Mathématiques de Bordeaux, Université de Bordeaux, CNRS (UMR 5251) [2]CNRS, IRIT (UMR 5505), Université de Toulouse. Correspondence to: Erell Gachon <erell.gachon@math.u-bordeaux.fr>.

*Proceedings of the 43rd International Conference on Machine Learning*, Seoul, South Korea. PMLR 306, 2026. Copyright 2026 by the author(s).

a random measure $\mu$. Then, $m$ independent samples $X_{i,1}, \cdots, X_{i,m}$ are drawn from each $\mu_i$. This setting introduces a double asymptotic framework, corresponding to both the sampling of distributions ($n \to \infty$) and the sampling within each distribution ($m \to \infty$). The relative growth rates of $n$ and $m$ give rise to a transition phenomenon between two regimes. In the *dense regime*, each measure is observed through a large number of samples ($m$ large relative to $n$), so that the intra-measure sampling error is negligible and the convergence is driven by the parametric rate in $n$. In the *sparse regime*, each measure is observed through only a few samples ($m$ small relative to $n$), so that the intra-measure sampling error dominates and the convergence rate is driven by $m$ and the embedding choice. While such transition phenomena have been extensively studied for functional PCA (Belhakem et al., 2025; Yao et al., 2005; Hall et al., 2006), they remain largely unexplored for PCA of random probability measures.

Formally, i.i.d. copies of the random measure $\mu$ are drawn, and each is then approximated by $m$ samples, that is:

$$\forall 1 \leq i \leq n, \qquad \hat{\mu}_i = \frac{1}{m} \sum_{j=1}^{m} \delta_{X_{i,j}}.$$

where, conditionally on $\mu_i$, $X_{i,j} \overset{iid}{\sim} \mu_i$ for $1 \leq j \leq m$. The natural estimator of the population covariance operator (1) is the empirical covariance

$$\hat{\Sigma} = \frac{1}{n} \sum_{i=1}^{n} \Phi(\hat{\mu}_i) \otimes \Phi(\hat{\mu}_i).$$

Our first goal is to understand how accurately $\hat{\Sigma}$ approximates the population covariance $\Sigma$ with respect to the expected Hilbert–Schmidt (HS) error $\mathbb{E}\|\hat{\Sigma} - \Sigma\|_{\mathrm{HS}}$.

This provides a direct measure of the statistical error. Beyond covariance estimation, PCA performance is naturally captured through the reconstruction risk, which quantifies how much of the data's variability is preserved after dimensionality reduction. Concretely, PCA searches for a rank-$q$ orthogonal projection that retains as much information as possible. Let $\mathcal{P}_q = \{P : \mathcal{H} \to \mathcal{H} \mid P$ is an orthogonal projector of rank $q\}$. Given $P \in \mathcal{P}_q$, we define the population and empirical reconstruction errors of PCA as follows:

$$R(P) = \mathbb{E}\|\Phi(\mu) - P\Phi(\mu)\|_{\mathcal{H}}^2, \tag{2}$$

$$\hat{R}(P) = \frac{1}{n} \sum_{i=1}^{n} \|\Phi(\hat{\mu}_i) - P\Phi(\hat{\mu}_i)\|_{\mathcal{H}}^2. \tag{3}$$

We can then define the following optimal projectors as minimizers of the reconstruction errors:

$$P_{\leq q} \in \underset{P \in \mathcal{P}_q}{\arg\min} \, R(P), \qquad \hat{P}_{\leq q} \in \underset{P \in \mathcal{P}_q}{\arg\min} \, \hat{R}(P). \tag{4}$$

Finally we define the excess risk of the empirical PCA projector $\hat{P}_{\leq q}$ as:

$$\mathcal{E}_q^{\mathrm{PCA}} = \mathbb{E}\Big[R(\hat{P}_{\leq q}) - R(P_{\leq q})\Big]. \tag{5}$$

This quantity measures how much reconstruction accuracy is lost by using the empirical covariance operator instead of its population counterpart.

### 1.1. Contributions

We quantify how the empirical covariance $\hat{\Sigma}$ and the empirical projector $\hat{P}_{\leq q}$ converge to their population counterparts, using the HS distance and excess risk. Our contributions are the following ones:

(i) Under mild assumptions on $\Phi$, we show that the empirical covariance $\hat{\Sigma}$ approximates the population covariance $\Sigma$ with the following convergence rate (see Theorem 3.3):

$$\mathbb{E}\|\hat{\Sigma} - \Sigma\|_{\mathrm{HS}} \lesssim n^{-1/2} + m^{-\alpha} \tag{6}$$

where the exponent $\alpha > 0$ depends on the chosen embedding $\Phi$ (see Table 1). This result reveals a transition phenomenon between the dense ($m > n^{1/2\alpha}$) and sparse ($m < n^{1/2\alpha}$) regimes.

(ii) We further show (Theorem 3.4) that in the dense regime $m > n^{1/2\alpha}$, the rate $n^{-1/2}$ in the upper bound (6) is minimax optimal.

(iii) Under mild assumptions on $\Phi$ and $q$, and assuming a polynomial decay of the eigenvalues of $\Sigma$, we show that the excess risk of PCA based on $\hat{\Sigma}$ achieves the following convergence rate (see Theorem 3.8):

$$\mathcal{E}_q^{\mathrm{PCA}} \lesssim n^{-1/2} + \sqrt{q} \, m^{-\alpha},$$

revealing a similar transition phenomenon between the dense and sparse regimes. Theorem 3.8 is stated here in a simplified form (under assumptions on the eigenvalues decay), more general conditions also apply.

(iv) We validate these predictions on simulated and real data, demonstrating the transition phenomenon and showing that subsampling to the dense regime threshold maintains statistical accuracy while reducing computational cost.

### 1.2. Related works

A natural approach to PCA in the space of probability measures is to rely on KME (Muandet et al., 2017), which maps measures into an RKHS where classical PCA applies. Another approach is to endow this space with the Wasserstein metric. Although this space is non-linear, its Riemannian-like structure can be exploited to define Geodesic PCA (Bigot et al., 2017; Seguy & Cuturi, 2015; Vesseron et al., 2025), where the principal modes of variation are geodesics

that best capture the variability in the data. These formulations involve significant computational complexity. Another common approach (Wang et al., 2013) consists in selecting a reference measure and embedding the data from the Wasserstein space into its tangent space at this point, where classical functional PCA can be applied.

So far, the behavior of PCA in the double asymptotic regime has been studied in the framework of functional PCA (Belhakem et al., 2025; Yao et al., 2005; Hall et al., 2006), for which the data are $n$ random functions belonging to the Hilbert space of squared integrable functions on the real line, observed at $m$ evenly spaced or randomly distributed points. For instance, the recent minimax analysis in (Belhakem et al., 2025) shows that, under a bivariate Hölder-type regularity $\alpha$ for the covariance kernel, the optimal (minimax) rate for estimating an eigenfunction of $\Sigma$ is of the order $n^{-1} + m^{-2\alpha}$, that also reveals a transition phenomenom between the sparse and dense sampling regimes of the observed functions. However, standard FPCA fails on probability measures: due to nonlinear geometry, FPCA components do not remain within the space of probability measures, rendering their interpretation difficult.

### 1.3. Notation

Let $\mathcal{X} \subseteq \mathbb{R}^d$ and $\mathcal{M}(\mathcal{X})$ be the set of Borel probability measures supported on $\mathcal{X}$. We equip $\mathcal{M}(\mathcal{X})$ with the Borel $\sigma$-algebra $\mathcal{B}(\mathcal{M}(\mathcal{X}))$ generated by the weak topology. We will also note an arbitrary separable Hilbert space $\mathcal{H}$ endowed with inner-product $\langle \cdot, \cdot \rangle_{\mathcal{H}}$ and corresponding norm $\| \cdot \|_{\mathcal{H}}$. We equip $\mathcal{H}$ with its Borel $\sigma$-algebra $\mathcal{B}(\mathcal{H})$ generated by the norm topology. We note $\| \cdot \|_{\mathrm{HS}}$ and $\langle \cdot, \cdot \rangle_{\mathrm{HS}}$ the HS norm and inner-product for linear operators. The tensor product operator is denoted $\otimes$ and defined such that for all $f, g, h \in \mathcal{H}$, $(f \otimes g)h = \langle f, h \rangle_{\mathcal{H}} g$. Reminders on the theory of operators on Hilbert spaces are given in Appendix E. For $a, b \in \mathbb{R}$, we write $a \lesssim b$ when there exists a constant $C > 0$ such that $a \leq Cb$, and we write $a \asymp b$ when $b \lesssim a \lesssim b$.

## 2. Background

We consider a Hilbert space embedding of the probability measure space $\mathcal{M}(\mathcal{X})$:

$$\Phi : \big(\mathcal{M}(\mathcal{X}), \mathcal{B}(\mathcal{M}(\mathcal{X}))\big) \to \big(\mathcal{H}, \mathcal{B}(\mathcal{H})\big).$$

We require the embedding $\Phi$ to be measurable, that is for any $B \in \mathcal{B}(\mathcal{H})$, $\Phi^{-1}(B) \in \mathcal{B}(\mathcal{M}(\mathcal{X}))$. In what follows, we will consider random probability measures in $\mathcal{M}(\mathcal{X})$.

**Definition 2.1** ((Panaretos & Zemel, 2020)). A random measure $\mu$ is any measurable map from a probability space $(\Omega, \mathcal{F}, \mathbb{P})$ to $\mathcal{M}(\mathcal{X})$, endowed with its Borel $\sigma$-algebra.

Let $\mu$ be a random probability measure. We consider $n$ i.i.d. copies $\mu_1, \cdots, \mu_n$ of $\mu$. For each $1 \leq i \leq n$, we draw $m$

independent samples $X_{i,1}, \cdots, X_{i,m}$ of $\mu_i$ and define the corresponding empirical measure $\hat{\mu}_i = (1/m) \sum_{j=1}^m \delta_{X_{i,j}}$. Then, we shall consider the following embeddings: $\Phi(\mu)$, $\Phi(\mu_i)$ and $\Phi(\hat{\mu}_i)$ for $1 \leq i \leq n$. As $\Phi$ is measurable, the embedding $\Phi(\mu)$ is a random variable in $\mathcal{H}$, that we suppose centered for simplicity. Finally, we define the quantity

$$r_m(\Phi) := \sqrt{\mathbb{E}\|\Phi(\mu_i) - \Phi(\hat{\mu}_i)\|_{\mathcal{H}}^2}, \qquad (7)$$

which measures the sampling error introduced by estimating the embedding from a finite number $m$ of samples from the measure $\mu_i$ and plays a central role in controlling the overall estimation error. We summarize existing results in the literature on the decay of $r_m(\Phi)$ in Table 1.

*Remark* 2.2. For simplicity, we assume a constant number of samples $m$ per measure, but our results extend to varying $m_i$. In that case, the intra-measure approximation error term becomes

$$\frac{1}{n} \sum_{i=1}^n r_{m_i}(\Phi) \leq r_{\underline{\mathbf{m}}}(\Phi) \quad \text{with} \quad \underline{\mathbf{m}} = \min_{1 \leq i \leq n} m_i.$$

### 2.1. Embeddings

We now describe the three specific embeddings considered in this work.

#### 2.1.1. KERNEL MEAN EMBEDDING

Given a positive definite kernel function $k : \mathcal{X} \times \mathcal{X} \to \mathbb{R}$, we note $\mathcal{H}_k$ the associated reproducing kernel Hilbert space (RKHS). The kernel mean embedding (Muandet et al., 2017) (KME) is then the following mapping:

$$\begin{aligned}
\Phi^{\mathrm{KME}} : \mathcal{M}(\mathcal{X}) &\to \mathcal{H}_k \\
\mu &\mapsto \int_{\mathcal{X}} k(x, \cdot) \mathrm{d}\mu(x).
\end{aligned}$$

#### 2.1.2. OT-BASED EMBEDDINGS

We start this section by a brief introduction to optimal transport (OT) (see (Villani et al., 2008)). Let $\rho$ and $\mu$ be two probability measures supported on $\mathcal{X}$. The *2-Wasserstein distance* $W_2$ is defined as (Kantorovich, 1942) as:

$$W_2^2(\rho, \mu) = \min_{\pi \in \Pi(\rho, \mu)} \int_{\mathcal{X} \times \mathcal{X}} \|x - y\|^2 \mathrm{d}\pi(x, y), \quad (8)$$

where $\Pi(\rho, \mu)$ is the set of probability measures on $\mathcal{X} \times \mathcal{X}$ with respective marginals $\rho$ and $\mu$, that is for any $A \subset \mathcal{X}$ (resp. $B \subset \mathcal{X}$), $\pi(A \times \mathcal{X}) = \rho(A)$ (resp. $\pi(\mathcal{X} \times B) = \mu(B)$). Another well-known formulation of OT is the so-called Monge problem (Monge, 1781):

$$\min_{T_{\#}\rho = \mu} \int_{\mathcal{X}} \|x - T(x)\|^2 \mathrm{d}\rho(x), \qquad (9)$$

where $T_{\#}\rho$ denotes the pushforward measure defined such that for all Borelian $B \subset \mathcal{X}, T_{\#}\rho(B) = \rho(T^{-1}(B))$. If it exists, a solution of (9) is called a *Monge map*. When $\rho$ is absolutely continuous, Brenier's theorem (Brenier, 1991) connects the two formulations of OT by stating that the optimal solution $\pi$ of (8) can be written as $\pi = (\mathrm{Id}, T)_{\#}\rho$, where $T$ is the unique Monge map.

**Linearized OT embedding** Fixing an absolutely continuous measure $\rho$, the linearized OT (LOT) (Wang et al., 2013) embedding consists of a mapping from the space of probability measures to the linear space $L^2(\rho)$ as follows:

$$\Phi^{\mathrm{LOT}} : \mathcal{M}(\mathcal{X}) \to L^2(\rho)$$
$$\mu \mapsto T_{\mu} - \mathrm{Id},$$

where $T_{\mu}$ is the Monge map from $\rho$ to $\mu$.

**Sliced Wasserstein embedding** To mitigate the high computational cost of OT, the *sliced Wasserstein distance* (SW)(Rabin et al., 2011) uses the property that Wasserstein distances between measures supported on the real line admit closed-form solutions. This approach alleviates the curse of dimensionality inherent to the LOT embedding: the sampling error (7) for the SW embedding is independent of the ambient dimension, see Table 1.

Let $\mathbb{S}^{d-1}$ denote the unit sphere in $\mathbb{R}^d$. For $\theta \in \mathbb{S}^{d-1}$, we define the projection $P^{\theta} : x \in \mathbb{R}^d \mapsto \langle \theta, x \rangle \in \mathbb{R}$. The SW distance $\mathrm{SW}_2$ between $\mu, \nu \in \mathcal{M}(\mathcal{X})$ is:

$$\mathrm{SW}_2^2(\mu, \nu) = \int_{\mathbb{S}^{d-1}} W_2^2(P_{\#}^{\theta}\mu, P_{\#}^{\theta}\nu)\mathrm{d}\sigma(\theta)$$
$$= \int_{\mathbb{S}^{d-1}} \int_0^1 (F_{\mu,\theta}^{-1}(t) - F_{\nu,\theta}^{-1}(t))^2 \mathrm{d}t\mathrm{d}\sigma(\theta)$$

where $\sigma$ is the uniform measure on $\mathbb{S}^{d-1}$ and $F_{\mu,\theta}^{-1}$ is the quantile function of $P_{\#}^{\theta}\mu$. Then, the SW embedding $\Phi^{\mathrm{SW}}$ maps probability measures to $L^2([0,1] \times \mathbb{S}^{d-1})$ in the following way:

$$\Phi^{\mathrm{SW}} : \mathcal{M}(\mathcal{X}) \to L^2([0,1] \times \mathbb{S}^{d-1})$$
$$\mu \mapsto \Phi(\mu)(t, \theta) = F_{\mu,\theta}^{-1}(t).$$

## 3. Theoretical results

In this section, we establish convergence rates for the empirical covariance operator and the PCA excess risk in the double asymptotic regime where both $n$ (the number of measures) and $m$ (the number of samples per measure) grow. Our analysis reveals how these two sources of variability interact and identifies the role of the embedding choice in determining overall statistical performance.

### 3.1. Statistical performance of the empirical covariance operator in the double asymptotic setting

We make the following fourth-moment assumption on the embedding $\Phi$.

**Assumption 3.1.** There exists a real number $R > 0$ such that the random probability measure $\mu$ verifies, for any positive integer $m$,

$$\mathbb{E}\|\Phi(\mu)\|_{\mathcal{H}}^4 \le R \quad \text{and} \quad \mathbb{E}\|\Phi(\hat{\mu})\|_{\mathcal{H}}^4 \le R,$$

where $\hat{\mu} = (1/m)\sum_{j=1}^m \delta_{X_j}$ denotes the empirical measure obtained by sampling $m$ points $X_1 \cdots, X_m$ iid from $\mu$.

*Remark* 3.2. Assumption 3.1 is typically satisfied by KME when using a bounded kernel, and by LOT and SW embeddings when the space $\mathcal{X}$ is bounded.

Note that the fourth-moment Assumption 3.1 ensures that the covariance operator of the random variable $\Phi(\mu)$ is trace-class (see Appendix E). All quantities involving HS norms and inner-products are therefore well-defined. The following theorem provides an upper bound on the HS distance between the empirical covariance operator and its population counterpart.

**Theorem 3.3.** *Under Assumption 3.1, we have that:*

$$\mathbb{E}\|\Sigma - \hat{\Sigma}\|_{\mathrm{HS}} \le R^{1/2}n^{-1/2} + 2R^{1/4}r_m(\Phi).$$

The statistical error has two sources: (i) sampling variability across the $n$ measures, and (ii) approximation error due to estimating each embedding from $m$ samples. The upper bound in Theorem 3.3 illustrates this by separating the $n^{-1/2}$ parametric term from the sampling error term $r_m(\Phi)$. In Table 1, we summarize existing results on the decay of $r_m(\Phi)$ and refer to Appendix F for more details.

*Sketch of Proof.* Theorem 3.3 is proved in Appendix A. We decompose the expected HS error into two terms by introducing the intermediate random covariance operator

$$\Sigma^n = \frac{1}{n}\sum_{i=1}^n \Phi(\mu_i) \otimes \Phi(\mu_i)$$

using the triangle inequality

$$\mathbb{E}\|\hat{\Sigma} - \Sigma\|_{\mathrm{HS}} \le \mathbb{E}\|\Sigma - \Sigma^n\|_{\mathrm{HS}} + \mathbb{E}\|\Sigma^n - \hat{\Sigma}\|_{\mathrm{HS}}.$$

Lemma A.1 bounds the first term $\mathbb{E}\|\Sigma - \Sigma^n\|_{\mathrm{HS}}$ by $R^{1/2}n^{-1/2}$ using the Jensen's inequality, an expansion of the squared HS norm and Assumption 3.1. For the second term, Lemma A.2 provides the bound $\mathbb{E}\|\Sigma^n - \hat{\Sigma}\|_{\mathrm{HS}} \le 2R^{1/4}r_m(\Phi)$. □

When $r_m(\Phi) \lesssim n^{-1/2}$ (which corresponds to the dense regime), the upper bound provided by Theorem 3.3 is dominated by the parametric rate $n^{-1/2}$. We complement this

upper bound with a minimax lower bound showing that the rate $n^{-1/2}$ cannot be improved in general when the per-measure estimation error is negligible, that is when $m$ is sufficiently large with respect to $n$. To clarify notation, we write $\Sigma = \Sigma_\mu$ to emphasize the dependence on the random measure $\mu$ in what follows.

**Theorem 3.4.** *Assume one of the following sufficient conditions on $R$ holds:*

$$R \geq \begin{cases} 4\,(d^2 + 2d) & \text{for the KME and LOT embeddings,} \\ 17 & \text{for the SW embedding.} \end{cases}$$

*Let $\mathcal{D}(R)$ be the class of random measures $\mu$ supported on $\mathbb{R}^d$ satisfying $\mathbb{E}\|\Phi(\mu)\|_{\mathcal{H}}^4 \leq R$. Then,*

$$\inf_{\hat{\Sigma}} \sup_{\mu \in \mathcal{D}(R)} \mathbb{E}\Big[\|\hat{\Sigma} - \Sigma_\mu\|_{\mathrm{HS}}\Big] \geq C n^{-1/2},$$

*where $\hat{\Sigma}$ denotes any covariance estimator (a measurable function of the data $(X_{ij})_{1 \leq i \leq n, 1 \leq j \leq m}$) of $\Sigma_\mu$ and $C > 0$ is an explicit constant depending on $d$ and the embedding $\Phi$, see Equation (27) in the Appendix.*

*Sketch of Proof.* The full proof is given in Appendix B. We follow the classical scheme from nonparametric statistics to obtain lower bounds on a minimax risk (Tsybakov, 2003) by reducing to a two-hypothesis testing problem. We construct two Gaussian models $H_0$ and $H_1$ such that the corresponding covariance operators of the embeddings are separated in HS distance by $2Cn^{-1/2}$ for a universal constant $C > 0$. We then show that the KL-divergence between these two Gaussian distributions is upper-bounded by a constant. These two conditions, combined with Markov's inequality and information-theoretic arguments from (Tsybakov, 2003), imply that no estimator can distinguish the two hypotheses with probability strictly better than a positive constant, yielding the desired minimax lower bound. □

*Remark* 3.5. The lower-bound conditions on $R$ in Theorem 3.4 ensure that the probability measures used to construct the two-hypothesis testing problem in the above proof belong to the class $\mathcal{D}(R)$.

*Remark* 3.6. Theorems 3.3 and 3.4 remain true if one replaces the HS norm by the operator norm $\|\cdot\|_{\mathrm{op}}$. Indeed, since $\|A\|_{\mathrm{op}} \leq \|A\|_{\mathrm{HS}}$ for any compact operator $A$, Theorem 3.3 immediately yields an operator norm upper bound with the same rate. The minimax lower bound can also be obtained by applying the same two-hypothesis model used for Theorem 3.4.

### 3.2. PCA excess risk

In addition to Assumption 3.1, bounding the excess risk requires the embedding $\Phi(\mu)$ to be subgaussian.

**Definition 3.7.** A random variable $\phi$ is subgaussian if $\mathbb{E}[\|\phi\|_{\mathcal{H}}^2]$ is finite and there exists a constant $C > 0$ such that for all $f \in \mathcal{H}$,

$$\sup_{k \geq 1} k^{-1/2} \mathbb{E}\big[|\langle \phi, f \rangle_{\mathcal{H}}|^k\big]^{1/k} \leq C \mathbb{E}\big[\langle \phi, f \rangle_{\mathcal{H}}^2\big]^{1/2}.$$

The following result provides an upper bound on the excess risk (5) of the empirical PCA projector relative to the population projector.

**Theorem 3.8.** *Under Assumption 3.1 and if $\Phi(\mu)$ is subgaussian, we have that:*

$$\mathcal{E}_q^{\mathrm{PCA}} \lesssim \sum_{j=1}^q \max\left\{ \sqrt{\frac{\lambda_j \sum\limits_{k \geq j} \lambda_k}{n}}, \frac{\sum\limits_{k \geq j} \lambda_k}{n} \right\} + 4R^{1/4}\sqrt{q}\, r_m(\Phi),$$

*where $(\lambda_j)_{j \geq 0}$ are the eigenvalues of the covariance operator $\Sigma$ defined in (1).*

In the context of PCA, the chosen number of components $q$ is typically small as the primary goal is dimensionality reduction and visualization. In particular, $q$ is often much smaller than $n$, the number of observed measures. The dependance on $q$ in the upper bound therefore may not significantly affect the overall excess risk in practice.

Mirroring Theorem 3.3, Theorem 3.8 again reveals two distinct sources of error. The first term captures the sampling variability across the $n$ observed measures, expressed through the eigenvalues of the population covariance operator. The second term reflects the intra-measure approximation error arising since each $\mu_i$ is observed through $m$ samples.

*Sketch of Proof.* Theorem 3.8 is proved in Appendix C. We decompose the excess risk of the empirical PCA projector into two terms by introducing the intermediate reconstruction error and its minimizer

$$R^n(P) = \frac{1}{n} \sum_{i=1}^n \|\Phi(\mu_i) - P\Phi(\mu_i)\|_{\mathcal{H}}^2,$$

and $P_{\leq q}^n \in \operatorname*{argmin}_{P \in \mathcal{P}_q} R^n(P)$ in Lemma C.1. Then Lemma C.2 provides a bound on the first term of this decomposition and reveals the interplay between the global $n^{-1/2}$ rate and the local $n^{-1}$ rate. This result relies on the variational characterization of partial traces and a concentration inequality for covariance operators, following the sketch of the proof in (Reiss & Wahl, 2020)[Proposition 2.5]. Lemma C.3 focuses on bounding the second term of this decomposition which depends on the convergence properties of the chosen embedding. It does so by bringing forward the 1-Wasserstein

distance between the empirical distribution of the $\Phi(\mu_i)$'s and the empirical distribution of the $\Phi(\hat{\mu}_i)$'s. $\qquad\square$

As a consequence from Theorem 3.8, we can also control the projection error.

**Corollary 3.9.** *The projection error inherits the bounds of Theorem 3.8 via the following:*

$$\|P_{\leq q} - \hat{P}_{\leq q}\|_{\mathrm{HS}}^2 \leq \frac{2\mathcal{E}_q^{\mathrm{PCA}}}{\lambda_{q+1} - \lambda_q}.$$

Corollary 3.9 is proven in Appendix C.4.

### 3.3. Implications of the main theorems

3.3.1. CHOICE OF $m$ FOR A FIXED $n$.

For a fixed number of observed measures $n$, the rate

$$n^{-1/2} + r_m(\Phi)$$

highlights how the choice of the number of samples $m$ per measure affects the overall error. Using the rates summarized in Table 1, we can ensure that the embedding approximation error is of the same order as the parametric $n^{-1/2}$ term, as described in the following.

(i) For the **LOT embedding**, we have $r_m(\Phi^{\mathrm{LOT}}) \asymp m^{-1/d}$. One should then choose

$$m \gtrsim n^{d/2}.$$

(ii) For the **KME and SW embeddings**, we have $r_m(\Phi^{\mathrm{KME}}) \asymp r_m(\Phi^{\mathrm{SW}}) \asymp m^{-1/2}$. One should then choose

$$m \gtrsim n.$$

This shows that the LOT embedding requires a larger sample size per measure for high dimensional data, while empirical PCA using either the SW embedding or KME converges faster with fewer samples.

3.3.2. EIGENVALUES DECAY

To clarify the implications of Theorem 3.8, we evaluates the first term in the upper bound of the excess risk, which depends on the eigenvalues of the covariance operator $\Sigma$:

$$\sum_{j=1}^q \max\left\{ \sqrt{\frac{\lambda_j \sum_{k\geq j}\lambda_k}{n}}, \frac{\sum_{k\geq j}\lambda_k}{n} \right\}.$$

We study this dependence by considering two classical cases of eigenvalue decay: polynomial and exponential. The following corollary gives explicit rates for the excess risk in these two scenarios.

**Corollary 3.10.** *Under the assumptions of Theorem 3.8, the followings hold:*

*(i) **Polynomial decay.** If $\lambda_j \asymp j^{-\alpha}$ for some $\alpha > 3/2$ and $q \leq n$, then*

$$\mathcal{E}_q^{\mathrm{PCA}} \lesssim n^{-1/2} + 2R\sqrt{q}\, r_m(\Phi).$$

*(ii) **Exponential decay.** If $\lambda_j \asymp e^{-\alpha j}$ for some $\alpha > 0$, then*

$$\mathcal{E}_q^{\mathrm{PCA}} \lesssim \begin{cases} n^{-1} + 2R\sqrt{q}\, r_m(\Phi), & \text{if } n \leq (1-e^{-\alpha})^{-1}, \\ n^{-1/2} + 2R\sqrt{q}\, r_m(\Phi), & \text{if } n \geq (1-e^{-\alpha})^{-1}. \end{cases}$$

*Remark* 3.11. Polynomial or exponential eigenvalue decay is standard in FPCA (Bosq, 2000; Ramsay & Silverman, 2005), where the covariance operator can be viewed as an integral operator with some kernel. The smoothness of this kernel governs the decay of its eigenvalues: smoother kernels lead to faster decay. Assuming $\alpha > 3/2$ corresponds to a mild smoothness condition on the embedded random object $\Phi(\mu)$. Such assumptions are classical and commonly used to control truncation errors in spectral approximation.

## 4. Numerical experiments

We conduct experiments on a simulated dataset of Gaussian measures and on real datasets, including flow cytometry datasets and 3D point clouds of objects. Additional experiments on simulated data and RGB images are provided in Appendix G.

### 4.1. Numerical experiments on simulated data

In this section, bold symbols $\boldsymbol{\mu}, \boldsymbol{b},...$ will denote random objects. To illustrate the transition phenomenon highlighted in the previous section, we consider the random measure $\boldsymbol{\mu} = \mathcal{N}(\boldsymbol{b}, \boldsymbol{\sigma}^2 I_2)$ where $\boldsymbol{b} = (\boldsymbol{b}_1, \boldsymbol{b}_2)^T$ is a random vector of $\mathbb{R}^2$ with independent entries and $\boldsymbol{\sigma} \in \mathbb{R}$ is a real random variable. This setting allows closed-form expressions for the embeddings and their covariance operators, see Propositions D.1, D.2 and D.3 in Appendix D.

We conduct numerical experiments to validate the theoretical convergence rates established in Theorems 3.3 and 3.8. We generate $n$ random Gaussian measures $\boldsymbol{\mu}_1, \ldots, \boldsymbol{\mu}_n$, with $\boldsymbol{\mu}_i = \mathcal{N}(\boldsymbol{b}_i, \boldsymbol{\sigma}_i^2 I_2)$, $\boldsymbol{b}_i \sim \mathcal{N}(0, \tau_b^2 I_2)$ and $\boldsymbol{\sigma}_i \sim \mathcal{N}(1, \tau_\sigma^2)$ independently. For each measure $\boldsymbol{\mu}_i$, we draw $m$ i.i.d. samples $\boldsymbol{X}_{i1}, \ldots, \boldsymbol{X}_{im}$ and construct the empirical measure $\hat{\boldsymbol{\mu}}_i = \frac{1}{m} \sum_{j=1}^m \delta_{\boldsymbol{X}_{ij}}$. The embeddings are then computed from these empirical measures.

Since $\Phi^{\mathrm{KME}}, \Phi^{\mathrm{LOT}}$ and $\Phi^{\mathrm{SW}}$ are infinite-dimensional functions, we approximate them in practice by computing their evaluation on a finite set of points. For the KME, we randomly sample $m_0$ points from $\rho = \mathcal{N}(0, I_2)$ and evaluate $\Phi^{\mathrm{KME}}$ on these points. For the LOT embedding, we consider the empirical measure $\hat{\rho}_{m_0}$ constructed from these

*Table 1.* Convergence rates of $\mathbb{E}\|\Phi(\mu) - \Phi(\hat{\mu})\|_{\mathcal{H}}^2$ and $r_m(\Phi) = \sqrt{\mathbb{E}\|\Phi(\mu) - \Phi(\hat{\mu})\|_{\mathcal{H}}^2}$ for different embeddings ($d \geq 5$), where $\hat{\mu}$ denotes an estimator of $\mu$ based on $m$ i.i.d. samples. For the LOT embedding, the reference measure $\rho$ is in practice also approximated from $m_0$ samples. For simplicity, we assume here that $m_0 = m$. Detailed assumptions and results are in Appendix F.

| Embedding | Ref | Assumptions | Measure estimator | Estimator of $\Phi$ | $\mathbb{E}\|\Phi(\mu) - \Phi(\hat{\mu})\|_{\mathcal{H}}^2$ | $r_m(\Phi)$ |
|---|---|---|---|---|---|---|
| KME | (Berlinet & Thomas-Agnan, 2011) | Bounded kernel | $\mu$: empirical measure | $\frac{1}{m}\sum_{j=1}^{m} k(x_j, \cdot)$ | $m^{-1}$ | $m^{-1/2}$ |
| LOT | (Manole et al., 2024) | F.2, F.4 | $\rho$: true measure $\mu$: empirical measure | True OT map | $m^{-2/d}$ | $m^{-1/d}$ |
| | (Manole et al., 2024) | F.4 $g$ $s$-smooth | $\rho$: true measure $\mu$: wavelet estimator | True OT map | $m^{-\frac{2s}{2(s-1)+d}}$ | $m^{-\frac{s}{2(s-1)+d}}$ |
| | (Manole et al., 2024) | F.2, F.3, F.4 $f$ bounded | $\rho$: empirical measure $\mu$: empirical measure | 1NN | $\log(m)m^{-2/d}$ | $\log(m)^{1/2}m^{-1/d}$ |
| | (Balakrishnan & Manole, 2025) | F.2, F.4 $f$ and $g$ $s$-smooth | $\rho$: any a.c. estimator $\mu$: any estimator | True OT map | $m^{-\frac{2(s+1)}{2s+d}}$ | $m^{-\frac{s+1}{2s+d}}$ |
| | (Pooladian et al., 2023) | F.2 $f$ bounded $\mu$ discrete | $\rho$: empirical measure $\mu$: empirical measure | Barycentric projection of the entropic optimal coupling $\varepsilon \asymp m^{-1/2}$ | $m^{-1/2}$ | $m^{-1/4}$ |
| SW | (Chewi et al., 2024) | Finite 2nd moment | $\mu$: empirical measure | $\Phi^{SW}(\hat{\mu})$ | $m^{-1}$ | $m^{-1/2}$ |

same $m_0$ points. When $m_0 \neq m$, Monge's problem (9) might not admit a solution. However, one can define a transport map from the optimal transport plan $\pi$ between $\hat{\rho}_{m_0}$ and $\hat{\mu}_i$ via the barycentric projection (Deb et al., 2021):

$$T_i(x) = \int_{\mathcal{X}} y \frac{\mathrm{d}\pi(x, y)}{\mathrm{d}\hat{\rho}_{m_0}(x)\mathrm{d}\hat{\mu}_i(y)}\mathrm{d}\hat{\mu}_i(y). \qquad (10)$$

Estimation rates for the barycentric projection do not appear in Table 1 but are detailed in Appendix F. Finally, for the SW embedding, $\Phi^{SW}$ is evaluated on $p$ random directions on the unit sphere $\mathbb{S}^1$ and $T$ evenly-spaced quantiles for each projection.

All results are averaged over 10 independent trials, and metrics are normalized to $[0, 1]$ using min-max scaling. We compute $\mathbb{E}\|\hat{\Sigma} - \Sigma\|_{HS}$ and the PCA excess risk $\mathcal{E}_q^{PCA}$ for $q = 1$ using the closed-form expression of $\Sigma$ from Propositions D.1, D.2 and D.3. We consider two sampling regimes:

(i) *Dense sampling regime* : We fix $m = 1000$ and vary $n$ from 10 to 1000. The global bound is dominated by the term in $n$, so the intra-measure sampling error is negligible compared to the inter-measure variability. The discretization parameters are $m_0 = 100$, $T = 10$, and $p = 10$. Figure 1 displays the results, and we observe slopes close to $-1/2$ for the covariance risk, and slopes close to $-1$ for the excess risk, which is consistent with our theory.

(ii) *Sparse sampling regime* : We fix $n = 500$ and vary $m$ from 10 to 500. The global bound is governed by the $r_m(\Phi)$ term so the intra-measure sampling error dominates. The discretization parameters are $m_0 = 500$, $T = 40$, and $p = 40$. Figure 2 displays the results.

Additionally, we study the variation of both risks with regards to the ambient dimension $d$, while keeping $n$ and $m$ fixed, see Table 2. The observations are consistent with Table 1. In particular, LOT exhibits a significant degrada-

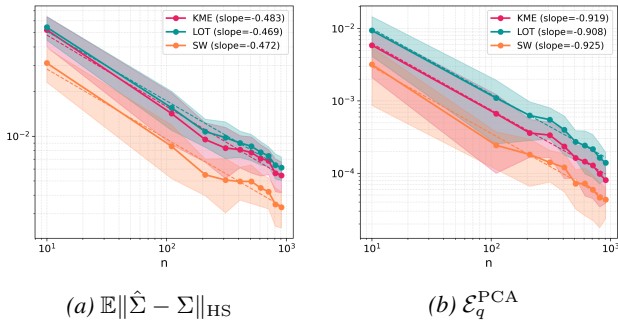

*(a)* $\mathbb{E}\|\hat{\Sigma} - \Sigma\|_{HS}$     *(b)* $\mathcal{E}_q^{PCA}$

*Figure 1.* **Dense sampling regime** ($m = 1000$ fixed, $n$ varies from 10 to 1000). The figures are plotted in log-log scale.

tion as $d$ increases, while KME and SW remain essentially stable.

We perfom an additional experiment to study the behaviour of the PCA excess risk as a function of the number of components $q$, while keeping $n$, $m$ and $d$ fixed, see Table 3.

### 4.2. Numerical experiments on real data

In this section, we focus on the results of PCA after subsampling the measures $\hat{\mu}_i$, followed by their embedding into a Hilbert space using the KME, the LOT and the SW embeddings. To assess the stability of PCA representations with respect to random subsampling, we use the following methodology throughout our experiments. For each subsample size $m$, we generate $N = 20$ independent random subsamples from each measure, compute the corresponding embedding (KME, LOT, or SW), and apply PCA to obtain a two-dimensional representation. For iteration $k$ and subsample size $m$, let $Y_k^{(m)} \in \mathbb{R}^{n \times 2}$ denote the two-dimensional PCA representation obtained from the $k$-th random subsample of size $m$. We compute the Procrustes (Gower, 1975) disparity $d_{kl}^{(m)} = d(Y_k^{(m)}, Y_l^{(m)})$ between iterations $k$ and

*Table 2.* Comparison of KME, LOT and SW across dimensions for covariance and excess risks. All reported values are multiplied by 100 for readability.

| | Covariance risk | | | Excess risk | | |
|---|---|---|---|---|---|---|
| $d$ | KME | LOT | SW | KME | LOT | SW |
| 2 | $1.43 \pm 0.31$ | $2.40 \pm 0.17$ | $0.71 \pm 0.12$ | $0.02 \pm 0.03$ | $0.07 \pm 0.02$ | $0.02 \pm 0.01$ |
| 5 | $2.60 \pm 0.32$ | $11.94 \pm 0.13$ | $0.66 \pm 0.15$ | $0.13 \pm 0.12$ | $0.38 \pm 0.05$ | $0.01 \pm 0.01$ |
| 10 | $4.29 \pm 0.32$ | $34.72 \pm 0.38$ | $0.55 \pm 0.05$ | $0.17 \pm 0.11$ | $0.71 \pm 0.05$ | $0.02 \pm 0.01$ |
| 50 | $12.64 \pm 0.78$ | $235.64 \pm 0.97$ | $0.52 \pm 0.03$ | $0.71 \pm 0.11$ | $0.93 \pm 0.01$ | $0.02 \pm 0.01$ |

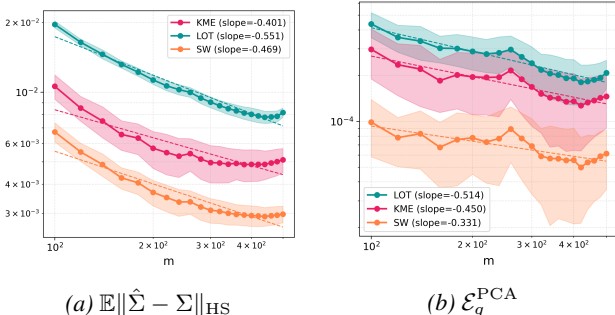

*(a)* $\mathbb{E}\|\hat{\Sigma} - \Sigma\|_{\mathrm{HS}}$   *(b)* $\mathcal{E}_q^{\mathrm{PCA}}$

*Figure 2.* **Sparse sampling regime** ($n = 500$ fixed, $m$ varies from 100 to 500). The figures are plotted in log-log scale.

*Table 3.* Comparison of KME, LOT and SW for the excess risk across different values of $q$. All reported values are multiplied by 100 for readability.

| $q$ | KME | LOT | SW |
|---|---|---|---|
| 1 | $16.23 \pm 11.66$ | $4.13 \pm 0.11$ | $0.13 \pm 0.07$ |
| 2 | $24.03 \pm 8.57$ | $8.28 \pm 0.15$ | $0.44 \pm 0.23$ |
| 5 | $49.22 \pm 8.39$ | $20.20 \pm 0.32$ | $2.31 \pm 0.19$ |
| 10 | $53.79 \pm 8.54$ | $39.09 \pm 0.47$ | $7.91 \pm 0.00$ |

$l$, with

$$d(Y_k^{(m)}, Y_l^{(m)}) = \min_{R,s,t} \|Y_k^{(m)} - sY_l^{(m)} - \mathbf{1}t^T\|^2,$$

where the minimum is taken over rotations $R \in \mathbb{R}^{2\times 2}$, scaling factors $s \in \mathbb{R}_+$ and translations $t \in \mathbb{R}^2$. Here, $\|\cdot\|_F$ denotes the Frobenius norm and $\mathbf{1} \in \mathbb{R}^n$ is the vector of ones. The quantity $d$ measures the dissimilarity between two datasets after optimal alignment. The mean Procrustes disparity for subsample size $m$ is then computed as:

$$\overline{d}^{(m)} = \frac{2}{N(N-1)} \sum_{1 \le k < l \le N} d_{kl}^{(m)},$$

and its standard deviation is computed in a similar way.

**Flow cytometry datasets.** We use publicly available flow cytometry datasets (Brusic et al., 2014) from the T-cell panel of the Human Immunology Project Consortium (HIPC). Seven laboratories each stained three replicates of three cryo-preserved biological samples (denoted patient 1, 2, and 3), yielding a total of $n = 7 \times 3 \times 3 = 63$ datasets.

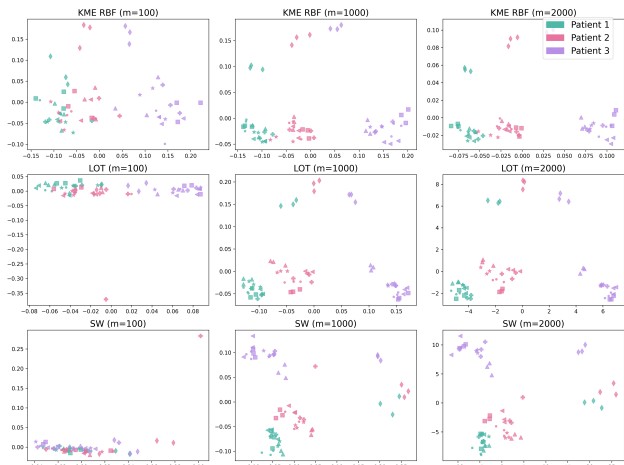

*Figure 3.* 2D PCA representation of the $n = 63$ HIPC measures for different subsample sizes $m$ (per column) and three embeddings (KME, LOT, SW, per row respectively). Each plot shows the projection onto the first two principal components, with different markers indicating different laboratories.

Each dataset consists of measurements of 7 markers for approximately 50,000 cells, which we treat as a point cloud in dimension $d = 7$. We perform PCA on these $n$ measures using the three embeddings considered in this work and study the effect of subsampling the cells from each measure. For KME, we use the radial basis function (RBF) kernel $k(x, y) = \exp(-\|x - y\|^2/(2\sigma^2))$ with bandwidth $\sigma = 1$. As in the simulated experiments, we sample $m_0 = 100$ points from $\rho = \mathcal{N}(0, I_d)$ to approximate the KME and LOT embeddings. For the SW embedding, we sample $p = 50$ random projections and $T = 100$ quantiles.

We start by visualizing the 2-dimensional PCA representation for each embedding and different subsample sizes in Figure 3. We observe that the representation stabilizes around $m = 50$, which is nearly identical to the $m = 2000$ case, representing a substantial reduction since only 0.5% of the approximately 50,000 available cells are needed while maintaining high-quality representations.

The mean Procrustes disparity and corresponding standard deviation are plotted in Figure 4a. We observe the expected decrease in disparity as $m$ increases, with stability around $m = 400$, confirming that relatively small subsample sizes

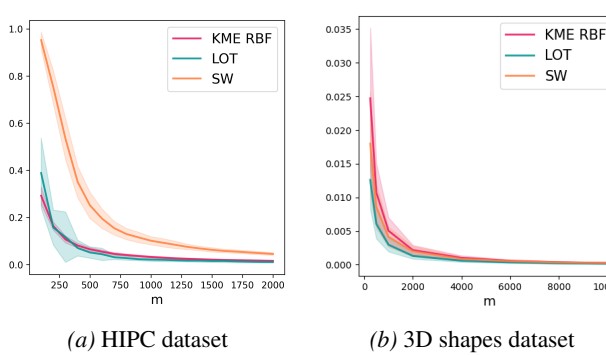

*(a)* HIPC dataset     *(b)* 3D shapes dataset

*Figure 4.* Evolution of mean (solid line) and standard deviation (shaded region) of Procrustes disparity for different subsample sizes on two datasets.

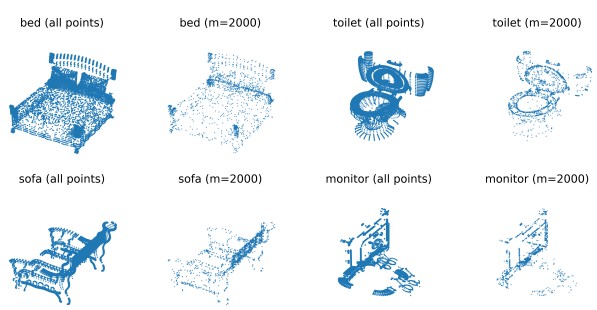

*Figure 5.* Example of 3D shapes from the ModelNet10 dataset and their point cloud representation obtained by sampling $m = 2000$ points.

are sufficient to obtain stable PCA representations.

**3D shapes dataset.** We consider the ModelNet10 dataset (Wu et al., 2015), which consists of 3D point clouds representing objects from different categories. We focus on 4 object categories and randomly select 5 shapes from each category, corresponding to the measures $\hat{\mu}_i$. We select shapes of approximately 50,000 points. For KME, we use the RBF kernel with bandwidth $\sigma = 0.5$. For the LOT embedding, the reference measure is the uniform measure on $[-1, 1]^d$ from which we sample $m_0 = 100$ points. For the SW embedding, the number of quantiles and the number of projections are respectively $T = 20$ and $p = 20$. We visualize the 2-dimensional PCA representation for each embedding and different subsample sizes in Figure 6. We also compute the mean and standard deviation of the Procrustes disparity as in the previous section, with results shown in Figure 4b. Similar to the HIPC dataset, we observe that the PCA representation stabilizes for relatively small subsample sizes. Figure 5 shows examples of shapes sampled with $m = 2000$ points, which corresponds to the stabilization regime indicated by the Procrustes analysis in Figure 4b.

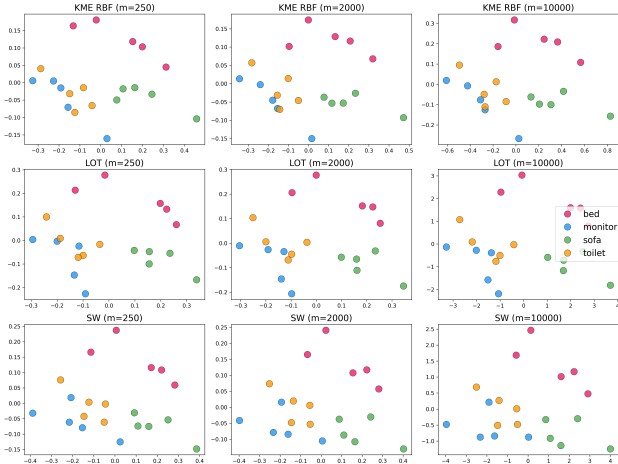

*Figure 6.* 2D PCA representation of the ModelNet10 dataset for different subsample sizes $m$ and three embeddings (KME, LOT, SW). Each plot shows the projection onto the first two principal components.

## 5. Conclusion

Understanding the interplay between the number $n$ of measures and the number $m$ of samples per measure is essential for PCA of probability measures via Hilbert space embeddings. In this work, we characterized the convergence behavior of the empirical covariance operator and the PCA excess risk, establishing rates of the form $n^{-1/2} + m^{-\alpha}$, where $\alpha > 0$ depends on the chosen embedding. In practice, the number $n$ of available distributions is often fixed. Our results then provide insight into choosing $m$: it should be large enough to accurately approximate the population PCA, while remaining small to limit computational costs. Through our numerical experiments, we observe that relatively small values of $m$ are sufficient to obtain high-quality PCA estimators.

## Acknowledgements

This work benefited from financial support from the French government managed by the National Agency for Research (ANR) under the France 2030 program, with the reference ANR-23-PEIA-0004.

## Impact statement

This paper presents work whose goal is to advance the field of Machine Learning. There are many potential societal consequences of our work, none which we feel must be specifically highlighted here.

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

## A. Proof of Theorem 3.3

To simplify the presentation in the proofs, we use the following notation for the embeddings of $\mu$, $\mu_i$ and $\hat{\mu}_i$:

$$\phi := \Phi(\mu) \qquad \phi_i := \Phi(\mu_i) \qquad \hat{\phi}_i := \Phi(\hat{\mu}_i)$$

Theorem 3.3 is obtained by first applying the triangle inequality

$$\mathbb{E}\|\Sigma - \hat{\Sigma}\|_{\mathrm{HS}} \le \mathbb{E}\|\Sigma - \Sigma^n\|_{\mathrm{HS}} + \mathbb{E}\|\Sigma^n - \hat{\Sigma}\|_{\mathrm{HS}},$$

where

$$\Sigma^n = \frac{1}{n} \sum_{i=1}^{n} \phi_i \otimes \phi_i$$

is the intermediate covariance operator. Then, the proof consists in bounding each of the above terms using Lemmas A.1 and A.2, whose proofs are deferred to Sections A.1 and A.2.

**Lemma A.1.** *Under Assumption 3.1,*
$$\mathbb{E}\|\Sigma - \Sigma^n\|_{\mathrm{HS}} \le R^{1/2} n^{-1/2}.$$

**Lemma A.2.** *Under Assumption 3.1,*

$$\mathbb{E}\|\Sigma^n - \hat{\Sigma}\|_{\mathrm{HS}} \le 2R^{1/4} \, r_m(\Phi).$$

### A.1. Proof of Lemma A.1

First, using Jensen's inequality, one has that:

$$\mathbb{E}\|\Sigma - \Sigma^n\|_{\mathrm{HS}} \le \sqrt{\mathbb{E}\big[\|\Sigma - \Sigma^n\|_{\mathrm{HS}}^2\big]}. \qquad (11)$$

Let us now focus on $\mathbb{E}\big[\|\Sigma - \Sigma^n\|_{\mathrm{HS}}^2\big]$, as follows.

$$\mathbb{E}\big[\|\Sigma - \Sigma^n\|_{\mathrm{HS}}^2\big] = \mathbb{E}\left[\left\|\Sigma - \frac{1}{n} \sum_{i=1}^{n} \phi_i \otimes \phi_i\right\|_{\mathrm{HS}}^2\right]$$

$$= \mathbb{E}\left[\|\Sigma\|_{\mathrm{HS}}^2\right] - 2\mathbb{E}\left[\left\langle \Sigma, \frac{1}{n} \sum_{i=1}^{n} \phi_i \otimes \phi_i \right\rangle_{\mathrm{HS}}\right] + \mathbb{E}\left[\left\|\frac{1}{n} \sum_{i=1}^{n} \phi_i \otimes \phi_i\right\|_{\mathrm{HS}}^2\right]$$

We have for the first two terms:

$$\mathbb{E}\left[\|\Sigma\|_{\mathrm{HS}}^2\right] - 2\mathbb{E}\left[\left\langle \Sigma, \frac{1}{n} \sum_{i=1}^{n} \phi_i \otimes \phi_i \right\rangle_{\mathrm{HS}}\right] = \|\Sigma\|_{\mathrm{HS}}^2 - 2\left\langle \Sigma, \frac{1}{n} \sum_{i=1}^{n} \mathbb{E}[\phi_i \otimes \phi_i] \right\rangle_{\mathrm{HS}}$$

$$= \|\Sigma\|_{\mathrm{HS}}^2 - 2\left\langle \Sigma, \frac{1}{n} \sum_{i=1}^{n} \Sigma \right\rangle_{\mathrm{HS}}$$

$$= -\|\Sigma\|_{\mathrm{HS}}^2$$

For the third term, we can write:

$$\mathbb{E}\left[\left\|\frac{1}{n} \sum_{i=1}^{n} \phi_i \otimes \phi_i\right\|_{\mathrm{HS}}^2\right] = \frac{1}{n^2}\mathbb{E}\left[\left\|\sum_{i=1}^{n} \phi_i \otimes \phi_i\right\|_{\mathrm{HS}}^2\right]$$

$$= \frac{1}{n^2}\mathbb{E}\left[\sum_{i=1}^{n}\|\phi_i \otimes \phi_i\|_{\mathrm{HS}}^2\right] + \frac{1}{n^2}\sum_{i=1}^{n}\sum_{j=1, j\neq i}^{n} \mathbb{E}\left[\langle \phi_i \otimes \phi_i, \phi_j \otimes \phi_j \rangle_{\mathrm{HS}}\right]$$

Using the independence of the $\phi_i$'s, we have:

$$\mathbb{E}\left[\left\|\frac{1}{n}\sum_{i=1}^{n}\phi_i \otimes \phi_i\right\|_{\mathrm{HS}}^2\right] = \frac{1}{n^2}\sum_{i=1}^{n}\mathbb{E}\left[\|\phi_i\|_{\mathcal{H}}^4\right] + \frac{1}{n^2}\sum_{i=1}^{n}\sum_{j=1,j\neq i}^{n}\left\langle\mathbb{E}\left[\phi_i \otimes \phi_i\right], \mathbb{E}\left[\phi_j \otimes \phi_j\right]\right\rangle_{\mathrm{HS}}$$

$$= \frac{1}{n^2}\sum_{i=1}^{n}\mathbb{E}\left[\|\phi_i\|_{\mathcal{H}}^4\right] + \frac{1}{n^2}\sum_{i=1}^{n}\sum_{j=1,j\neq i}^{n}\langle\Sigma,\Sigma\rangle_{\mathrm{HS}}$$

$$= \frac{\mathbb{E}\left[\|\phi\|_{\mathcal{H}}^4\right]}{n} + \frac{n-1}{n}\|\Sigma\|_{\mathrm{HS}}^2$$

where we used that $\|\phi \otimes \phi\|_{\mathrm{HS}}^2 = \|\phi\|_{\mathcal{H}}^4$, which follows from the definition of the HS norm: for any orthonormal basis $(f_j)_{j\geq 1}$ of $\mathcal{H}$, we have

$$\|\phi \otimes \phi\|_{\mathrm{HS}}^2 = \sum_{j\geq 1}\|(\phi \otimes \phi)f_j\|_{\mathcal{H}}^2 = \sum_{j\geq 1}|\langle\phi, f_j\rangle_{\mathcal{H}}|^2\|\phi\|_{\mathcal{H}}^2 = \|\phi\|_{\mathcal{H}}^4.$$

Then,

$$\mathbb{E}\left[\|\Sigma - \Sigma^n\|_{\mathrm{HS}}^2\right] = \mathbb{E}\left[\|\Sigma\|_{\mathrm{HS}}^2\right] - 2\mathbb{E}\left[\left\langle\Sigma, \frac{1}{n}\sum_{i=1}^{n}\phi_i \otimes \phi_i\right\rangle_{\mathrm{HS}}\right] + \mathbb{E}\left[\left\|\frac{1}{n}\sum_{i=1}^{n}\phi_i \otimes \phi_i\right\|_{\mathrm{HS}}^2\right]$$

$$= -\|\Sigma\|_{\mathrm{HS}}^2 + \frac{\mathbb{E}\left[\|\phi\|_{\mathcal{H}}^4\right]}{n} + \frac{n-1}{n}\|\Sigma\|_{\mathrm{HS}}^2$$

$$= \frac{1}{n}\left(\mathbb{E}\left[\|\phi\|_{\mathcal{H}}^4\right] - \|\Sigma\|_{\mathrm{HS}}^2\right)$$

$$\leq \frac{\mathbb{E}\left[\|\phi\|_{\mathcal{H}}^4\right]}{n}$$

$$\leq Rn^{-1}.$$

Thus, we can conclude with (11) that

$$\mathbb{E}\|\Sigma - \Sigma^n\|_{\mathrm{HS}} \leq R^{1/2}n^{-1/2}$$

## A.2. Proof of Lemma A.2

For each $1 \leq i \leq n$ we have, by linearity of the tensor product operator, the identity

$$\phi_i \otimes \phi_i - \hat{\phi}_i \otimes \hat{\phi}_i = (\phi_i - \hat{\phi}_i) \otimes \phi_i \ + \ \hat{\phi}_i \otimes (\phi_i - \hat{\phi}_i).$$

Summing over $i$ and dividing by $n$ yields

$$\Sigma^n - \hat{\Sigma} = \frac{1}{n}\sum_{i=1}^{n}\left[(\phi_i - \hat{\phi}_i) \otimes \phi_i \ + \ \hat{\phi}_i \otimes (\phi_i - \hat{\phi}_i)\right].$$

Since the equality $\|u \otimes v\|_{\mathrm{HS}} = \|u\|_{\mathcal{H}}\|v\|_{\mathcal{H}}$ holds, it follows that

$$\|\phi_i \otimes \phi_i - \hat{\phi}_i \otimes \hat{\phi}_i\|_{\mathrm{HS}} \ \leq \ \|\phi_i - \hat{\phi}_i\|_{\mathcal{H}}\left(\|\phi_i\|_{\mathcal{H}} + \|\hat{\phi}_i\|_{\mathcal{H}}\right),$$

and therefore

$$\|\Sigma^n - \hat{\Sigma}\|_{\mathrm{HS}} \leq \frac{1}{n}\sum_{i=1}^{n}\|\phi_i - \hat{\phi}_i\|_{\mathcal{H}}\left(\|\phi_i\|_{\mathcal{H}} + \|\hat{\phi}_i\|_{\mathcal{H}}\right).$$

Taking expectations and applying Cauchy–Schwarz's inequality to each term gives

$$\mathbb{E}\|\Sigma^n - \hat{\Sigma}\|_{\mathrm{HS}} \leq \frac{1}{n}\sum_{i=1}^{n}\sqrt{\mathbb{E}\|\phi_i - \hat{\phi}_i\|_{\mathcal{H}}^2}\ \sqrt{\mathbb{E}\left(\|\phi_i\|_{\mathcal{H}} + \|\hat{\phi}_i\|_{\mathcal{H}}\right)^2}.$$

The fourth-moment Assumption 3.1 implies the second moment bound $\mathbb{E}\|\Phi(\mu)\|_{\mathcal{H}}^2 \leq R^{1/2}$. Using this and the inequality $(a+b)^2 \leq 2(a^2+b^2)$, we obtain that

$$\sqrt{\mathbb{E}\big(\|\phi_i\|_{\mathcal{H}} + \|\hat{\phi}_i\|_{\mathcal{H}}\big)^2} \leq \sqrt{2\,\mathbb{E}\|\phi_i\|_{\mathcal{H}}^2 + 2\,\mathbb{E}\|\hat{\phi}_i\|_{\mathcal{H}}^2} \leq 2R^{1/4}.$$

Substituting this into the previous bound gives

$$
\begin{aligned}
\mathbb{E}\|\Sigma^n - \hat{\Sigma}\|_{\mathrm{HS}} &\leq 2R^{1/4} n^{-1} \sum_{i=1}^{n} \sqrt{\mathbb{E}\|\phi_i - \hat{\phi}_i\|_{\mathcal{H}}^2} \\
&\leq 2R^{1/4} n^{-1} \sum_{i=1}^{n} r_m(\Phi) \\
&= 2R^{1/4}\, r_m(\Phi),
\end{aligned}
$$

which concludes the proof.

## B. Proof of Theorem 3.4

In this section, we will use for clarity bold symbols $\boldsymbol{\mu}, \boldsymbol{b}, \cdots$ to denote random objects. Furthermore, $\hat{\Sigma} = \hat{\Sigma}((\boldsymbol{X}_{ij})_{1 \leq i \leq n, 1 \leq j \leq m})$ denotes an estimator of $\Sigma_\mu$ based on the samples $(\boldsymbol{X}_{ij})_{1 \leq i \leq n, 1 \leq j \leq m}$.

### B.1. General reduction scheme for proving minimax rates

We will follow the classical scheme for proving minimax bounds on $\mathbb{E}\|\Sigma_\mu - \hat{\Sigma}\|_{\mathrm{HS}}$, see Section 2.2 in (Tsybakov, 2003). We start by using Markov's inequality:

$$\inf_{\hat{\Sigma}} \sup_{\mu \in \mathcal{D}(R)} \mathbb{E}\Big[n^{1/2}\,\|\Sigma_\mu - \hat{\Sigma}\|_{\mathrm{HS}}\Big] \geq C \inf_{\hat{\Sigma}} \sup_{\mu \in \mathcal{D}(R)} \mathbb{P}\Big(\|\Sigma_\mu - \hat{\Sigma}\|_{\mathrm{HS}} \geq Cn^{-1/2}\Big) \tag{12}$$

And we will reduce the problem to two ($M = 2$) hypothesis:

$$C \inf_{\hat{\Sigma}} \sup_{\mu \in \mathcal{D}(R)} \mathbb{P}\Big(\|\Sigma_\mu - \hat{\Sigma}\|_{\mathrm{HS}} \geq Cn^{-1/2}\Big) \geq C \inf_{\hat{\Sigma}} \max_{k \in \{1,2\}} \mathbb{P}^{(k)}\big(\|\Sigma^{(k)} - \hat{\Sigma}\|_{\mathrm{HS}} \geq Cn^{-1/2}\big), \tag{13}$$

where for $k = 1, 2$, $\mathbb{P}^{(k)}$ will denote the probability measure of the data $(\boldsymbol{X}_{ij})_{1 \leq i \leq n, 1 \leq j \leq m}$ under the hypothesis $H_k$, $\boldsymbol{\mu}^{(k)}$ will denote a specific random measure under $H_k$, and $\Sigma^{(k)} = \mathbb{E}[\Phi(\boldsymbol{\mu}^{(k)}) \otimes \Phi(\boldsymbol{\mu}^{(k)})]$ denotes the covariance operator of the random embedding $\Phi(\boldsymbol{\mu}^{(k)})$. These will be defined precisely in Section B.2 below. Inequality (Tsybakov, 2003)[Equation 2.8] states that if

$$\|\Sigma^{(1)} - \Sigma^{(2)}\|_{\mathrm{HS}} \geq 2Cn^{-1/2}, \tag{14}$$

then for any estimator $\hat{\Sigma}$,

$$\mathbb{P}^{(j)}\big(\|\hat{\Sigma} - \Sigma^{(j)}\|_{\mathrm{HS}} \geq Cn^{-1/2}\big) \geq \mathbb{P}^{(j)}\big(\psi^* \neq j\big), \qquad \forall j \in \{1,2\}$$

where $\psi^* = \operatorname*{argmin}_{k \in \{1,2\}} \|\hat{\Sigma} - \Sigma^{(k)}\|_{\mathrm{HS}}$. It then follows that if (14) holds, then

$$C \inf_{\hat{\Sigma}} \max_{k \in \{1,2\}} \mathbb{P}^{(k)}\big(\|\hat{\Sigma} - \Sigma^{(k)}\|_{\mathrm{HS}} \geq Cn^{-1/2}\big) \geq C \inf_{\psi \in \{1,2\}} \max_{k \in \{1,2\}} \mathbb{P}^{(k)}(\psi \neq k). \tag{15}$$

Additionally, Theorem 2.2 in (Tsybakov, 2003) states that if

$$\mathrm{KL}(\mathbb{P}^{(1)}, \mathbb{P}^{(2)}) \leq \alpha < \infty \tag{16}$$

where KL is the Kullback-Leibler divergence, then

$$\inf_{\psi \in \{1,2\}} \max_{k \in \{1,2\}} \mathbb{P}^{(k)}(\psi \neq k) \geq \max\Big(\frac{e^{-\alpha}}{4}, \frac{1 - \sqrt{\alpha/2}}{2}\Big). \tag{17}$$

Therefore, if we construct two hypothesis which satisfy (14) and (16), then combining inequalities (12), (13), (15), and (17) gives:

$$\inf_{\hat{\Sigma}} \sup_{\mu \in \mathcal{D}(R)} \mathbb{E}\Big[n^{1/2}\,\|\Sigma_\mu - \hat{\Sigma}\|_{\mathrm{HS}}\Big] \geq C \max\Big(\frac{e^{-\alpha}}{4}, \frac{1 - \sqrt{\alpha/2}}{2}\Big). \tag{18}$$

## B.2. Probability model

Let $s \geq 1$ be a real number. Our two hypothesis $H_k, k \in \{1, 2\}$, will be separated in the following way:

$$s^{(2)} = s, \qquad s^{(1)} = s + \varepsilon, \qquad \varepsilon = n^{-1/2}.$$

For $k \in \{1, 2\}$ we define the following random variables (mutually independent):

$$\mathbf{b}^{(k)} \sim \mathcal{N}(0,\, s^{(k)} I_d) \qquad \mathbf{Z}_{ij}^{(k)} \sim \mathcal{N}(0,\, I_d), \qquad 1 \leq i \leq n,\, 1 \leq j \leq m.$$

Let $\boldsymbol{b}_1^{(k)}, \cdots, \boldsymbol{b}_n^{(k)}$ be independent copies of $\boldsymbol{b}^{(k)}$. The data is sampled according to the following model:

$$\boldsymbol{X}_{ij}^{(k)} = \boldsymbol{b}_i^{(k)} + \boldsymbol{Z}_{ij}^{(k)} \in \mathbb{R}^d, \qquad 1 \leq i \leq n, \quad 1 \leq j \leq m. \tag{19}$$

For each $k \in \{1, 2\}$, if we denote $\boldsymbol{\mu}_i^{(k)}$ the measure from which $(\boldsymbol{X}_{ij}^{(k)}|\boldsymbol{b}_i^{(k)})_{1 \leq j \leq m}$ are sampled , it follows, from model (19) that $\boldsymbol{\mu}_1^{(k)}, \cdots, \boldsymbol{\mu}_n^{(k)}$ are independent copies of the random measure $\boldsymbol{\mu}^{(k)} = \mathcal{N}(\boldsymbol{b}^{(k)},\, I_d)$. For each $1 \leq i \leq n$, we form the concatenated vectors by stacking the $m$ vectors $\boldsymbol{X}_{ij}^{(k)}$ and the $m$ vectors $\boldsymbol{Z}_{ij}^{(k)}$:

$$\boldsymbol{X}_i^{(k)} = (\boldsymbol{X}_{i1}^{(k)}, \cdots, \boldsymbol{X}_{im}^{(k)})^T = \sum_{j=1}^m e_j \otimes \boldsymbol{X}_{ij}^{(k)} \in \mathbb{R}^{dm} \qquad \boldsymbol{Z}_i^{(k)} = (\boldsymbol{Z}_{i1}^{(k)}, \cdots, \boldsymbol{Z}_{im}^{(k)})^T = \sum_{j=1}^m e_j \otimes \boldsymbol{Z}_{ij}^{(k)} \in \mathbb{R}^{dm},$$

where $e_j \in \mathbb{R}^m$ is the $j$-th canonical basis vector and $\otimes$ denotes the Kronecker product. Then the random vectors $\boldsymbol{X}_1^{(k)}, \cdots, \boldsymbol{X}_n^{(k)}$ are independent and can be written as:

$$\boldsymbol{X}_i^{(k)} = \sum_{j=1}^m e_j \otimes \boldsymbol{b}_i^{(k)} + \sum_{j=1}^m e_j \otimes \boldsymbol{Z}_{ij}^{(k)}$$
$$= \mathbb{1}_m \otimes \boldsymbol{b}_i^{(k)} + \boldsymbol{Z}_i^{(k)}$$

where $\mathbb{1}_m = (1, \cdots, 1)^T \in \mathbb{R}^m$. For $k \in \{1, 2\}$, let $\mathbb{P}^{(k)}$ be the probability measure of the data in model (19) under hypothesis $H_k$. The following Lemma, proven in Section B.5, allows to bound the Kullback Leibler divergence between $\mathbb{P}^{(1)}$ and $\mathbb{P}^{(2)}$ by a constant.

**Lemma B.1.** *The deconditioned law of the $\boldsymbol{X}_i^{(k)}$ is:*

$$\boldsymbol{X}_i^{(k)} \sim \mathcal{N}\Big(0,\, I_{dm} + s^{(k)} \mathbb{1}_m \mathbb{1}_m^T \otimes I_d\Big).$$

*Furthermore, the Kullback Leibler divergence between $\mathbb{P}^{(1)}$ and $\mathbb{P}^{(2)}$ can be bounded as:*

$$\mathrm{KL}(\mathbb{P}^{(1)}, \mathbb{P}^{(2)}) \leq \frac{d}{4}. \tag{20}$$

## B.3. Separation of hypothesis

In this section, we prove that condition (14) holds for the KME, LOT and SW embeddings. First, recall that $\mathcal{D}(R)$ denotes the class of random measures $\mu$ supported on $\mathbb{R}^d$ satisfying $\mathbb{E}\|\Phi(\mu)\|_{\mathcal{H}}^4 \leq R$.

### B.3.1. KME

**Lemma B.2.** *For $k \in \{1, 2\}$, let $\Sigma^{(k)}$ be the covariance operator of the random embedding $\Phi^{\mathrm{KME}}(\boldsymbol{\mu}^{(k)})$ for the linear kernel. If $R \geq 4(d^2 + 2d)$ and if*

$$1 \leq s \leq \sqrt{\frac{R}{d^2 + 2d}} - 1, \tag{21}$$

*then, $\boldsymbol{\mu}^{(k)} \in \mathcal{D}(R)$. Furthermore,*

$$\|\Sigma^{(1)} - \Sigma^{(2)}\|_{\mathrm{HS}} \geq 2C^{\mathrm{KME}} n^{-1/2} \tag{22}$$

*where $C^{\mathrm{KME}} = \frac{\sqrt{d}}{2}$.*

### B.3.2. LOT EMBEDDING

**Lemma B.3.** *For $k \in \{1, 2\}$, let $\Sigma^{(k)}$ be the covariance operator of the random embedding $\Phi^{\mathrm{LOT}}(\boldsymbol{\mu}^{(k)})$. If $R \geq 4(d^2 + 2d)$ and if*

$$1 \leq s \leq \sqrt{\frac{R}{d^2 + 2d}} - 1, \tag{23}$$

*then, $\boldsymbol{\mu}^{(k)} \in \mathcal{D}(R)$. Furthermore,*

$$\|\Sigma^{(1)} - \Sigma^{(2)}\|_{\mathrm{HS}} \geq 2C^{\mathrm{LOT}} n^{-1/2}, \tag{24}$$

*where $C^{\mathrm{LOT}} = \frac{\sqrt{d}}{2}$.*

### B.3.3. SLICED-WASSERSTEIN

**Lemma B.4.** *For $k \in \{1, 2\}$, let $\Sigma^{(k)}$ be the covariance operator of the random embedding $\Phi^{\mathrm{SW}}(\boldsymbol{\mu}^{(k)})$. If $R \geq 17$ and if*

$$1 \leq s \leq \frac{-8 + \sqrt{12R - 8}}{6}. \tag{25}$$

*then, $\boldsymbol{\mu}^{(k)} \in \mathcal{D}(R)$. Furthermore,*

$$\|\Sigma^{(1)} - \Sigma^{(2)}\|_{\mathrm{HS}} \geq 2C^{\mathrm{SW}} n^{-1/2}, \tag{26}$$

*where $C^{\mathrm{SW}} = \frac{1}{2\sqrt{d}}$.*

## B.4. Conclusion of the proof of Theorem 3.4

Combining Lemma B.1 with Lemmas B.2, B.3 and B.4, and choosing $\alpha = d/4$ in (16), we obtain that for each of the three embeddings considered, choosing $s$ accordingly to respectively (21), (23) and (25) ensures that the two hypothesis constructed satisfy both (14) and (16). Therefore, for each of the three embeddings, we have:

$$\inf_{\hat{\Sigma}} \sup_{\mu \in \mathcal{D}(R)} \mathbb{E}\left[ n^{1/2} \|\Sigma_\mu - \hat{\Sigma}\|_{\mathrm{HS}} \right] \geq C \max\left( \frac{e^{-d/4}}{4}, \frac{1 - \sqrt{d/8}}{2} \right), \tag{27}$$

where $C = C^{\mathrm{KME}}$ for the KME embedding, $C = C^{\mathrm{LOT}}$ for the LOT embedding and $C = C^{\mathrm{SW}}$ for the SW embedding. This concludes the proof of Theorem 3.4.

## B.5. Proof of Lemma B.1

*Proof of Lemma B.1.* By construction, we have that $\boldsymbol{X}_i^{(k)} | \boldsymbol{b}_i^{(k)} \sim \mathcal{N}(\mathbb{1}_m \otimes \boldsymbol{b}_i^{(k)}, \, I_{dm})$. As $\boldsymbol{b}_i^{(k)}$ and $\boldsymbol{Z}_i^{(k)}$ are Gaussian vectors and independent, and as $\boldsymbol{X}_i^{(k)}$ is a linear function of that pair, $\boldsymbol{X}_i^{(k)}$ is Gaussian in $\mathbb{R}^{dm}$. We can then deduce that $\boldsymbol{X}_i^{(k)}$ is also a Gaussian vectors with:

$$\begin{aligned} \mathbb{E}[\boldsymbol{X}_i^{(k)}] &= \mathbb{E}\left[ \mathbb{E}\left[ \boldsymbol{X}_i^{(k)} | \boldsymbol{b}_i^{(k)} \right] \right] \\ &= \mathbb{E}\left[ \mathbb{1}_m \otimes \boldsymbol{b}_i^{(k)} \right] \\ &= \mathbb{1}_m \otimes \mathbb{E}\left[ \boldsymbol{b}_i^{(k)} \right] = 0 \end{aligned}$$

and by the law of total variance, see (Blitzstein & Hwang, 2019)[Theorem 9.5.5],

$$
\begin{aligned}
\text{Cov}(\boldsymbol{X}_i^{(k)}) &= \mathbb{E}\big[\text{Cov}(\boldsymbol{X}_i^{(k)}|\,\boldsymbol{b}_i^{(k)})\big] + \text{Cov}\big(\mathbb{E}[\boldsymbol{X}_i^{(k)}|\,\boldsymbol{b}_i^{(k)}]\big) \\
&= \mathbb{E}[I_{dm}] + \text{Cov}(\mathbb{1}_m \otimes \boldsymbol{b}_i^{(k)}) \\
&= I_{dm} + \mathbb{1}_m \mathbb{1}_m^T \otimes \text{Cov}(\boldsymbol{b}_i^{(k)}) \\
&= I_{dm} + s^{(k)} \mathbb{1}_m \mathbb{1}_m^T \otimes I_d.
\end{aligned}
$$

Therefore, the probability measure $\mathbb{P}^{(k)}$ of the data vector $(\boldsymbol{X}_1^{(k)}, \ldots, \boldsymbol{X}_n^{(k)})$ in model (19) under hypothesis $H_k$ is the product of $n$ independent Gaussian measures $\mathbb{P}_i^{(k)} = \mathcal{N}\big(0,\ I_{dm} + s^{(k)} \mathbb{1}_m \mathbb{1}_m^T \otimes I_d\big), 1 \le i \le n$. The Kullback Leibler divergence between two Gaussians in $\mathbb{R}^p$ can be written as follows:

$$
\text{KL}\big(\mathcal{N}(\eta^{(1)}, K^{(1)}), \mathcal{N}(\eta^{(2)}, K^{(2)})\big) = \frac{1}{2}\Big((\eta^{(2)}-\eta^{(1)})^T (K^{(2)})^{-1}(\eta^{(2)}-\eta^{(1)}) + \text{Tr}\big((K^{(2)})^{-1}K^{(1)}\big) - \ln\frac{|K^{(1)}|}{|K^{(2)}|} - p\Big) \quad (28)
$$

In our case, the dimension is $p = dm$, the means are $0_{dm}$ and $K^{(k)} = I_{dm} + s^{(k)} \mathbb{1}_m \mathbb{1}_m^T \otimes I_d$. To compute the inverse of $K^{(k)}$, we start by writing:

$$
\begin{aligned}
K^{(k)} &= I_{dm} + s^{(k)}(\mathbb{1}_m \otimes I_d)(\mathbb{1}_m^T \otimes I_d) \\
&= I_{dm} + (\mathbb{1}_m \otimes I_d)s^{(k)}I_d(\mathbb{1}_m^T \otimes I_d).
\end{aligned}
$$

The Woodbury matrix identity tells us that:

$$
(A + UCV)^{-1} = A^{-1} - A^{-1}U(C^{-1} + VA^{-1}U)^{-1}VA^{-1}.
$$

Applying this identity to $A = I_{dm}, U = \mathbb{1}_m \otimes I_d, C = s^{(k)}I_d$ and $V = \mathbb{1}_m^T \otimes I_d$ yields:

$$
\begin{aligned}
(K^{(k)})^{-1} &= I_{dm} - I_{dm}(\mathbb{1}_m \otimes I_d)\Big(\frac{1}{s^{(k)}}I_d + (\mathbb{1}_m^T \otimes I_d)I_{dm}(\mathbb{1}_m \otimes I_d)\Big)^{-1}(\mathbb{1}_m^T \otimes I_d)I_{dm} \\
&= I_{dm} - (\mathbb{1}_m \otimes I_d)\Big(\frac{1}{s^{(k)}}I_d + (\mathbb{1}_m^T \mathbb{1}_m \otimes I_d)\Big)^{-1}(\mathbb{1}_m^T \otimes I_d) \\
&= I_{dm} - (\mathbb{1}_m \otimes I_d)\Big(\frac{1}{s^{(k)}} + m\Big)^{-1}(\mathbb{1}_m^T \otimes I_d) \\
&= I_{dm} - \frac{s^{(k)}}{1 + ms^{(k)}}(\mathbb{1}_m \mathbb{1}_m^T \otimes I_d)
\end{aligned}
$$

Then the product gives:

$$
\begin{aligned}
(K^{(2)})^{-1}K^{(1)} &= \Big(I_{dm} - \frac{s^{(2)}}{1 + ms^{(2)}}(\mathbb{1}_m \mathbb{1}_m^T \otimes I_d)\Big)\Big(I_{dm} + s^{(1)}\mathbb{1}_m \mathbb{1}_m^T \otimes I_d\Big) \\
&= I_{dm} + s^{(1)}\mathbb{1}_m \mathbb{1}_m^T \otimes I_d - \frac{s^{(2)}}{1 + ms^{(2)}}(\mathbb{1}_m \mathbb{1}_m^T \otimes I_d) - \frac{s^{(1)}s^{(2)}}{1 + ms^{(2)}}(\mathbb{1}_m \mathbb{1}_m^T \otimes I_d)(\mathbb{1}_m \mathbb{1}_m^T \otimes I_d) \\
&= I_{dm} + s^{(1)}\mathbb{1}_m \mathbb{1}_m^T \otimes I_d - \frac{s^{(2)}}{1 + ms^{(2)}}(\mathbb{1}_m \mathbb{1}_m^T \otimes I_d) - \frac{s^{(1)}s^{(2)}}{1 + ms^{(2)}}(\mathbb{1}_m \mathbb{1}_m^T \mathbb{1}_m \mathbb{1}_m^T \otimes I_d) \\
&= I_{dm} + s^{(1)}\mathbb{1}_m \mathbb{1}_m^T \otimes I_d - \frac{s^{(2)}}{1 + ms^{(2)}}(\mathbb{1}_m \mathbb{1}_m^T \otimes I_d) - \frac{ms^{(1)}s^{(2)}}{1 + ms^{(2)}}(\mathbb{1}_m \mathbb{1}_m^T \otimes I_d) \\
&= I_{dm} + \Big(s^{(1)} - \frac{s^{(2)}}{1 + ms^{(2)}} - \frac{ms^{(1)}s^{(2)}}{1 + ms^{(2)}}\Big)(\mathbb{1}_m \mathbb{1}_m^T \otimes I_d) \\
&= I_{dm} + \frac{s^{(1)} - s^{(2)}}{1 + ms^{(2)}}(\mathbb{1}_m \mathbb{1}_m^T \otimes I_d).
\end{aligned}
$$

Then we can compute the trace of this product:

$$\mathrm{Tr}\Big((K^{(2)})^{-1}K^{(1)}\Big) = \mathrm{Tr}\Big(I_{dm}\Big) + \frac{s^{(1)} - s^{(2)}}{1 + ms^{(2)}}\mathrm{Tr}\Big(\mathbb{1}_m\mathbb{1}_m^T \otimes I_d\Big)$$

$$= dm + \frac{s^{(1)} - s^{(2)}}{1 + ms^{(2)}}\mathrm{Tr}\Big(\mathbb{1}_m\mathbb{1}_m^T\Big)\mathrm{Tr}\Big(I_d\Big)$$

$$= dm + \frac{s^{(1)} - s^{(2)}}{1 + ms^{(2)}}dm$$

$$= d\Big(m - 1 + 1 + \frac{ms^{(1)} - ms^{(2)}}{1 + ms^{(2)}}\Big)$$

$$= d\Big(m - 1 + \frac{1 + ms^{(1)}}{1 + ms^{(2)}}\Big)$$

where we used that for two matrices $A$, $B$, $\mathrm{Tr}(A \otimes B) = \mathrm{Tr}(A)\mathrm{Tr}(B)$. We now need to determine the determinant $\big|K^{(k)}\big|$. Using that for two square matrices $A$ and $B$ of respective size $n$ and $m$, $|A \otimes B| = |A|^m \cdot |B|^n$, we have that:

$$\big|K^{(k)}\big| = \big|(I_m + s^{(k)}\mathbb{1}_m\mathbb{1}_m^T) \otimes I_d\big| = \big|I_m + s^{(k)}\mathbb{1}_m\mathbb{1}_m^T\big|^d \cdot 1^m.$$

The matrix $I_m + s^{(k)}\mathbb{1}_m\mathbb{1}_m^T$ has eigenvalues 1 with multiplicity $m - 1$ and $1 + ms^{(k)}$ with multiplicity 1. This gives:

$$\ln\Big(\frac{|K^{(1)}|}{|K^{(2)}|}\Big) = \ln\Big(\frac{1^{d(m-1)}(1 + ms^{(1)})^d}{1^{d(m-1)}(1 + ms^{(2)})^d}\Big)$$

$$= d\ln\Big(\frac{1 + ms^{(1)}}{1 + ms^{(2)}}\Big)$$

Putting the terms together, and recalling that $s^{(2)} = s$ and $s^{(1)} = s + \varepsilon$ for some $s \geq 1$, we obtain:

$$\mathrm{KL}(\mathbb{P}_i^{(1)}, \mathbb{P}_i^{(2)}) = \frac{d}{2}\Big(m - 1 + \frac{1 + ms^{(1)}}{1 + ms^{(2)}} - \ln\Big(\frac{1 + ms^{(1)}}{1 + ms^{(2)}}\Big) - m\Big)$$

$$= \frac{d}{2}\Big(\frac{1 + ms^{(1)}}{1 + ms^{(2)}} - \ln\Big(\frac{1 + ms^{(1)}}{1 + ms^{(2)}}\Big) - 1\Big)$$

$$= \frac{d}{2}\Big(\frac{m\varepsilon}{1 + ms} - \ln\Big(1 + \frac{m\varepsilon}{1 + ms}\Big)\Big)$$

We notice the following inequality:

$$\forall x \geq 0, \quad x - \ln(1 + x) = \int_0^x \frac{t}{t + 1}\mathrm{d}t \leq \int_0^x t\,\mathrm{d}t = \frac{x^2}{2}$$

Then:

$$\mathrm{KL}(\mathbb{P}_i^{(1)}, \mathbb{P}_i^{(2)}) \leq \frac{d}{4}\frac{m^2}{(1 + ms)^2}\varepsilon^2 \leq \frac{d}{4}\varepsilon^2,$$

The inequality $\frac{m^2}{(1+ms)^2} \leq 1$ is verified as we assumed $s \geq 1$. Finally, using that $\varepsilon = n^{-1/2}$,

$$\mathrm{KL}(\mathbb{P}^{(1)}, \mathbb{P}^{(2)}) = \sum_{i=1}^n \mathrm{KL}(\mathbb{P}_i^{(1)}, \mathbb{P}_i^{(2)}) \leq \frac{nd}{4}\varepsilon^2 = \frac{d}{4}.$$

$\square$

## B.6. Proofs of Section B.3

*Proof of Lemma B.2.* The interval $[1, \sqrt{\frac{R}{d^2+2d}} - 1]$ is non-empty when $R \geq 4(d^2 + 2d)$. The KME of the probability measure $\boldsymbol{\mu}^{(k)} = \mathcal{N}(\boldsymbol{b}^{(k)}, I_d)$ is:

$$\forall x \in \mathbb{R}^d, \qquad \Phi^{\mathrm{KME}}(\boldsymbol{\mu}^{(k)})(x) = \int_{\mathbb{R}^d} x^T y \mathrm{d}\boldsymbol{\mu}^{(k)}(y) = x^T \boldsymbol{b}^{(k)}$$

We have that:

$$\mathbb{E}\|\Phi^{\mathrm{KME}}(\boldsymbol{\mu}^{(k)})\|_{L^2(\rho)}^4 = \mathbb{E}\Big\langle k(\cdot, \boldsymbol{b}^{(k)}), k(\cdot, \boldsymbol{b}^{(k)}) \Big\rangle_{\mathcal{H}}^2 = \mathbb{E}\big[k(\boldsymbol{b}^{(k)}, \boldsymbol{b}^{(k)})^2\big] = \mathbb{E}\|\boldsymbol{b}^{(k)}\|^4$$

Since $\boldsymbol{b}^{(k)} = \sqrt{s^{(k)}} Z$ with $Z \sim \mathcal{N}(0, I_d)$, we have $\|b^{(k)}\|^2 = s^{(k)}\|Z\|^2$. The random variable $\|Z\|^2 \sim \chi_d^2$ follows a chi-squared distribution with $d$ degrees of freedom. Using the fourth moment, we have that $\mathbb{E}\|Z\|^4 = d(d+2)$. Therefore,

$$\mathbb{E}\|\Phi^{\mathrm{KME}}(\boldsymbol{\mu}^{(k)})\|_{L^2(\rho)}^4 = (d^2 + 2d)(s^{(k)})^2 \leq (d^2 + 2d)(s^{(1)} + \varepsilon)^2 \leq (d^2 + 2d)(s^{(1)} + 1)^2.$$

Therefore, thanks to inequality (21), for $k \in \{1, 2\}$, $\mathbb{E}\|\Phi^{\mathrm{KME}}(\boldsymbol{\mu}^{(k)})\|_{L^2(\rho)}^4 \leq R$ and hence $\boldsymbol{\mu}^{(k)} \in \mathcal{D}(R)$.

Using Proposition D.1, the covariance operator $\Sigma^{(k)}$ admits the following eigendecomposition:

$$\Sigma^{(k)} = \mathrm{Var}(\boldsymbol{b}_1^{(k)}) \sum_{i=1}^d f_i \otimes f_i,$$

with $f_i(x) = x^T e_i$. Then, we have that:

$$\begin{aligned}
\|\Sigma^{(1)} - \Sigma^{(2)}\|_{\mathrm{HS}}^2 &= (s^{(1)} - s^{(2)})^2 \Big\|\sum_{i=1}^d f_i \otimes f_i\Big\|_{\mathrm{HS}}^2 \\
&= (s^{(1)} - s^{(2)})^2 \sum_{i=1}^d \|f_i \otimes f_i\|_{\mathrm{HS}}^2 \\
&= (s^{(1)} - s^{(2)})^2 \sum_{i=1}^d \|f_i\|_{\mathcal{H}}^4 \\
&= (s^{(1)} - s^{(2)})^2 d \\
&= dn^{-1}.
\end{aligned}$$

Taking $C = C^{\mathrm{KME}} = \frac{\sqrt{d}}{2}$ concludes the proof. $\qquad\square$

*Proof of Lemma B.3.* The interval $[1, \sqrt{\frac{R}{d^2+2d}} - 1]$ is non empty when $4(d^2 + 2d) \leq R$. Using the well-known formula for optimal transport between Gaussians $\rho = \mathcal{N}(0, I_d)$ and $\boldsymbol{\mu}^{(k)} = \mathcal{N}(\boldsymbol{b}^{(k)}, I_d)$ (see for instance Equation 2.40 in (Peyré et al., 2019)), we have that the LOT embedding of $\boldsymbol{\mu}^{(k)}$ is given by:

$$\forall x \in \mathbb{R}^d, \qquad \Phi^{\mathrm{LOT}}(\boldsymbol{\mu}^{(k)})(x) = (I_d - I_d)x + \boldsymbol{b}^{(k)} = \boldsymbol{b}^{(k)}.$$

We need to check that for $k \in \{1, 2\}$, $\boldsymbol{\mu}^{(k)} \in \mathcal{D}(R)$. We have that

$$\mathbb{E}\|\Phi^{\mathrm{LOT}}(\boldsymbol{\mu}^{(k)})\|_{L^2(\rho)}^4 = \mathbb{E}\Big(\int_{\mathbb{R}^d} \|\boldsymbol{b}^{(k)}\|^2 \mathrm{d}\rho(x)\Big)^2 = \mathbb{E}\|\boldsymbol{b}^{(k)}\|^4.$$

Using the same argument as in the proof of Lemma B.2, we have that $\mathbb{E}\|\boldsymbol{b}^{(k)}\|^4 = (d^2 + 2d)(s^{(k)})^2$. Therefore,

$$\mathbb{E}\|\Phi^{\mathrm{LOT}}(\boldsymbol{\mu}^{(k)})\|_{L^2(\rho)}^4 = (d^2 + 2d)(s^{(k)})^2 \leq (d^2 + 2d)(s^{(1)} + \varepsilon)^2 \leq (d^2 + 2d)(s^{(1)} + 1)^2.$$

Therefore, thanks to inequality (23), for $k \in \{1, 2\}$, $\mathbb{E}\|\Phi^{\mathrm{LOT}}(\boldsymbol{\mu}^{(k)})\|_{L^2(\rho)}^4 \leq R$ and hence $\boldsymbol{\mu}^{(k)} \in \mathcal{D}(R)$.

Using Proposition D.2, the covariance operator $\Sigma^{(k)}$ admits the following eigendecomposition:

$$\Sigma^{(k)} = \mathrm{Var}(\boldsymbol{b}^{(k)}) \sum_{i=1}^{d} f_i \otimes f_i = s^{(k)} \sum_{i=1}^{d} f_i \otimes f_i$$

with $f_i(x) = e_i$. Then, we have that:

$$
\begin{aligned}
\|\Sigma^{(1)} - \Sigma^{(2)}\|_{\mathrm{HS}}^2 &= \left\|(s^{(1)} - s^{(2)}) \sum_{i=1}^{d} f_i \otimes f_i\right\|_{\mathrm{HS}}^2 \\
&= (s^{(1)} - s^{(2)})^2 \sum_{i=1}^{d} \|f_i \otimes f_i\|_{\mathrm{HS}}^2 \\
&= (s^{(1)} - s^{(2)})^2 \sum_{i=1}^{d} \|f_i\|_{L^2(\rho)}^4 \\
&= (s^{(1)} - s^{(2)})^2 d \\
&= \varepsilon^2 d \\
&= dn^{-1}.
\end{aligned}
$$

Taking $C = C^{\mathrm{LOT}} = \frac{\sqrt{d}}{2}$ concludes the proof. $\qquad\square$

*Proof of Lemma B.4.* The Sliced–Wasserstein embedding of $\boldsymbol{\mu}^{(k)}$ is the function

$$\forall t \in [0, 1], \forall \theta \in \mathbb{S}^{d-1}, \qquad \Phi^{\mathrm{SW}}(\boldsymbol{\mu}^{(k)})(t, \theta) = \theta^\top \boldsymbol{b}^{(k)} + \sqrt{2}\,\mathrm{erf}^{-1}(2t - 1),$$

The squared norm of this embedding is given by:

$$\|\Phi^{\mathrm{SW}}(\boldsymbol{\mu}^{(k)})\|_{L^2([0,1]\times\mathbb{S}^{d-1})}^2 = \int_{\mathbb{S}^{d-1}} \int_0^1 \left(\theta^\top \boldsymbol{b}^{(k)} + \sqrt{2}\,\mathrm{erf}^{-1}(2t-1)\right)^2 \mathrm{d}t\,\mathrm{d}\sigma(\theta).$$

Expanding and using that $\int_0^1 \mathrm{erf}^{-1}(2t-1)\,\mathrm{d}t = 0$ and $\int_0^1 \left(\sqrt{2}\,\mathrm{erf}^{-1}(2t-1)\right)^2 \mathrm{d}t = 2 \cdot \frac{1}{2} = 1$ (the latter follows because $\sqrt{2}\,\mathrm{erf}^{-1}(2t-1)$ is a standard normal quantile), and that $\int_{\mathbb{S}^{d-1}} \theta\theta^\top \,\mathrm{d}\sigma(\theta) = \frac{1}{d} I_d$, we obtain

$$\|\Phi^{\mathrm{SW}}(\boldsymbol{\mu}^{(k)})\|_{L^2([0,1]\times\mathbb{S}^{d-1})}^2 = \int_{\mathbb{S}^{d-1}} (\boldsymbol{b}^{(k)})^\top \theta\theta^\top \boldsymbol{b}^{(k)} \,\mathrm{d}\sigma(\theta) + 1 = \frac{\|\boldsymbol{b}^{(k)}\|^2}{d} + 1.$$

Hence

$$\left\|\Phi^{\mathrm{SW}}(\boldsymbol{\mu}^{(k)})\right\|_{L^2}^4 = \frac{\|\boldsymbol{b}^{(k)}\|^4}{d^2} + \frac{2\|\boldsymbol{b}^{(k)}\|^2}{d} + 1,$$

and taking expectation over $\boldsymbol{b}^{(k)}$ yields

$$\mathbb{E}\left\|\Phi^{\mathrm{SW}}(\boldsymbol{\mu}^{(k)})\right\|_{L^2}^4 = \frac{\mathbb{E}\|\boldsymbol{b}^{(k)}\|^4}{d^2} + \frac{2\mathbb{E}\|\boldsymbol{b}^{(k)}\|^2}{d} + 1.$$

For the Gaussian $\boldsymbol{b}^{(k)} \sim \mathcal{N}(0, s^{(k)} I_d)$, we have $\mathbb{E}\|\boldsymbol{b}^{(k)}\|^2 = s^{(k)} d$ and $\mathbb{E}\|\boldsymbol{b}^{(k)}\|^4 = (s^{(k)})^2(d^2 + 2d)$. Substituting gives

$$\mathbb{E}\left\|\Phi^{\mathrm{SW}}(\boldsymbol{\mu}^{(k)})\right\|_{L^2}^4 = \left(1 + \frac{2}{d}\right)(s^{(k)})^2 + 2s^{(k)} + 1.$$

To ensure $\boldsymbol{\mu}^{(k)} \in \mathcal{D}(R)$ for $k = 1, 2$ it suffices to bound this fourth moment by $R$ for the worst-case variance. If $s^{(1)} = s + \varepsilon$ and $\varepsilon \leq 1$ we may use the simple bound $s^{(1)} \leq s + 1$ and also the inequality $1 + \frac{2}{d} \leq 3$ to obtain the sufficient condition

$$3(s+1)^2 + 2(s+1) + 1 \leq R.$$

Expanding the left-hand side yields

$$3(s+1)^2 + 2(s+1) + 1 = 3s^2 + 8s + 6.$$

Solving the quadratic inequality $3s^2 + 8s + 6 \le R$ gives the (larger) root

$$s_{\max} = \frac{-8 + \sqrt{12R - 8}}{6},$$

so all $s$ with $s \le s_{\max}$ satisfy the inequality. Imposing also $s \ge 1$ we get a nonempty feasible interval $[1, s_{\max}]$ precisely when

$$s_{\max} \ge 1 \quad \Longleftrightarrow \quad R \ge 17.$$

Therefore, the condition

$$R \ge 17 \qquad \text{and} \qquad 1 \le s \le \frac{-8 + \sqrt{12R - 8}}{6}$$

is sufficient to guarantee $\mathbb{E}\|\Phi^{\mathrm{SW}}(\boldsymbol{\mu}^{(k)})\|_{L^2}^4 \le R$ for $k = 1, 2$, hence $\boldsymbol{\mu}^{(k)} \in \mathcal{D}(R)$.

Using Proposition D.3, the covariance operator $\Sigma^{(k)}$ the following eigendecomposition:

$$\Sigma^{(k)} = \frac{1}{d}\mathrm{Var}(\boldsymbol{b}_1^{(k)}) \sum_{i=1}^{d} f_i \otimes f_i,$$

with $f_i(t, \theta) = \sqrt{d}\theta_i$. Then, we have that

$$\begin{aligned}
\|\Sigma^{(1)} - \Sigma^{(2)}\|_{\mathrm{HS}}^2 &= (s^{(1)} - s^{(2)})^2 \frac{1}{d^2}\left\|\sum_{i=1}^{d} f_i \otimes f_i\right\|_{\mathrm{HS}}^2 \\
&= (s^{(1)} - s^{(2)})^2 \frac{1}{d^2}\sum_{i=1}^{d}\|f_i \otimes f_i\|_{\mathrm{HS}}^2 \\
&= (s^{(1)} - s^{(2)})^2 \frac{1}{d^2}\sum_{i=1}^{d}\|f_i\|_{L^2([0,1]\times\mathbb{S}^{d-1})}^4 \\
&= \frac{1}{d}(s^{(1)} - s^{(2)})^2 \\
&= \frac{1}{d}n^{-1}.
\end{aligned}$$

Taking $C = C^{\mathrm{SW}} = \frac{1}{2\sqrt{d}}$ concludes the proof. $\qquad\square$

## C. Proof of Theorem 3.8

We recall the definitions of the covariance operators associated to $\phi$, $\phi_i$ and $\hat{\phi}_i$:

$$\Sigma = \mathbb{E}[\phi \otimes \phi] \qquad \Sigma^n = \frac{1}{n}\sum_{i=1}^{n}\phi_i \otimes \phi_i \qquad \hat{\Sigma} = \frac{1}{n}\sum_{i=1}^{n}\hat{\phi}_i \otimes \hat{\phi}_i.$$

Given Assumption 3.1, the covariance operator $\Sigma$ is trace-class and therefore compact. The empirical covariance operators $\Sigma^n$ and $\hat{\Sigma}$ are also compact, as they have finite rank (at most $n$). By the spectral theorem for compact self-adjoint operators (see Theorem E.3), these covariance operators admit the following spectral representations:

$$\Sigma = \sum_{j\ge 0}\lambda_j P_j, \qquad \Sigma^n = \sum_{j\ge 0}\lambda_j^n P_j^n \qquad \hat{\Sigma} = \sum_{j\ge 0}\hat{\lambda}_j \hat{P}_j,$$

where $(\lambda_j)_{j\ge 0}$, $(\lambda_j^n)_{j\ge 0}$ and $(\hat{\lambda}_j)_{j\ge 0}$ are positive eigenvalues sorted in decreasing order, and the $P_j$'s, $P_j^n$'s and $\hat{P}_j$'s are rank one projectors. Given $P \in \mathcal{P}_q$, we define the following PCA reconstruction errors

$$R(P) = \mathbb{E}\|\phi - P\phi\|_{\mathcal{H}}^2, \qquad R^n(P) = \frac{1}{n}\sum_{i=1}^{n}\|\phi_i - P\phi_i\|_{\mathcal{H}}^2, \qquad \hat{R}(P) = \frac{1}{n}\sum_{i=1}^{n}\|\hat{\phi}_i - P\hat{\phi}_i\|_{\mathcal{H}}^2 \qquad (29)$$

and note the corresponding minimizers

$$P_{\leq q} \in \underset{P \in \mathcal{P}_q}{\operatorname{argmin}} R(P), \qquad P_{\leq q}^n \in \underset{P \in \mathcal{P}_q}{\operatorname{argmin}} R^n(P), \qquad \hat{P}_{\leq q} \in \underset{P \in \mathcal{P}_q}{\operatorname{argmin}} \hat{R}(P).$$

These optimal projectors can be expressed from the spectral representations of the covariance operators.

$$P_{\leq q} = \sum_{j=1}^{q} P_j, \qquad P_{\leq q}^n = \sum_{j=1}^{q} P_j^n, \qquad \hat{P}_{\leq q} = \sum_{j=1}^{q} \hat{P}_j.$$

To establish Theorem 3.8, we prove the following Lemmas C.1, C.2 and C.3. We first start with Lemma C.1 by rewriting the excess risk $\mathcal{E}_q^{\mathrm{PCA}}$ and splitting it into two terms that we will bound separately.

**Lemma C.1.**

$$\mathcal{E}_q^{\mathrm{PCA}} \leq \underbrace{\mathbb{E}\Big[\langle \Sigma - \Sigma^n, P_{\leq q} - \hat{P}_{\leq q}\rangle_{\mathrm{HS}}\Big]}_{(i)} + \underbrace{\mathbb{E}\Big[\langle \Sigma^n - \hat{\Sigma}, P_{\leq q} - \hat{P}_{\leq q}\rangle_{\mathrm{HS}}\Big]}_{(ii)} \tag{30}$$

Here, we have divided the task into two sources of error: **(i)** the error due to the sampling of the $n$ measures $\mu_1, \cdots, \mu_n$ and **(ii)** the error due to the sampling of $m$ points from each measure. We then focus with Lemma C.2 on bounding **(i)**.

**Lemma C.2** (Bounding **(i)**)**.** *If $\phi$ is subgaussian and Assumption 3.1 is verified, we have*

$$\mathbb{E}\big[\langle \Sigma - \Sigma^n, P_{\leq q} - \hat{P}_{\leq q}\rangle_{\mathrm{HS}}\big] \lesssim \sum_{j=1}^{q} \max\left\{\sqrt{\frac{\lambda_j \sum_{k \geq j} \lambda_k}{n}}, \frac{\sum_{k \geq j} \lambda_k}{n}\right\}$$

The second part consists in bounding **(ii)**, which is done in Lemma C.3 below.

**Lemma C.3** (Bounding **(ii)**)**.** *Under Assumption 3.1, we have*

$$\mathbb{E}\big[\langle \Sigma^n - \hat{\Sigma}, P_{\leq q} - \hat{P}_{\leq q}\rangle_{\mathrm{HS}}\big] \leq 4R^{1/4}\sqrt{q}\, r_m(\Phi)$$

Combining Lemmas C.2 and C.3 yields the expected result for Theorem 3.8.

### C.1. Proof of Lemma C.1

The reconstruction errors defined in (29) can be expressed in terms of the corresponding covariance operators:

$$R(P) = \langle \Sigma, I - P\rangle_{\mathrm{HS}} \qquad R^n(P) = \langle \Sigma^n, I - P\rangle_{\mathrm{HS}} \qquad \hat{R}(P) = \langle \hat{\Sigma}, I - P\rangle_{\mathrm{HS}}.$$

These equalities for the reconstruction error is a well known result in PCA which we recall in Lemma C.5 in Section C.5. As $\hat{P}_{\leq q}$ is a minimizer of $\hat{R}$, we have that

$$\hat{R}(P_{\leq q}) - \hat{R}(\hat{P}_{\leq q}) = \langle \hat{\Sigma}, \hat{P}_{\leq q} - P_{\leq q}\rangle_{\mathrm{HS}} \geq 0. \tag{31}$$

This allows us to split the error into two terms.

$$\begin{aligned}
R(\hat{P}_{\leq q}) - R(P_{\leq q}) &= \langle \Sigma, I - \hat{P}_{\leq q}\rangle_{\mathrm{HS}} - \langle \Sigma, I - P_{\leq q}\rangle_{\mathrm{HS}} \\
&= \langle \Sigma, P_{\leq q} - \hat{P}_{\leq q}\rangle_{\mathrm{HS}} \\
&\overset{(31)}{\leq} \langle \Sigma, P_{\leq q} - \hat{P}_{\leq q}\rangle_{\mathrm{HS}} + \langle \hat{\Sigma}, \hat{P}_{\leq q} - P_{\leq q}\rangle_{\mathrm{HS}} \\
&= \langle \Sigma - \hat{\Sigma}, P_{\leq q} - \hat{P}_{\leq q}\rangle_{\mathrm{HS}} \\
&= \langle \Sigma - \Sigma^n + \Sigma^n - \hat{\Sigma}, P_{\leq q} - \hat{P}_{\leq q}\rangle_{\mathrm{HS}} \\
&= \langle \Sigma - \Sigma^n, P_{\leq q} - \hat{P}_{\leq q}\rangle_{\mathrm{HS}} + \langle \Sigma^n - \hat{\Sigma}, P_{\leq q} - \hat{P}_{\leq q}\rangle_{\mathrm{HS}}
\end{aligned}$$

## C.2. Proof of Lemma C.2

We start by splitting the left term into two:

$$\mathbb{E}\Big[\langle \Sigma - \Sigma^n, P_{\leq q} - \hat{P}_{\leq q}\rangle_{\mathrm{HS}}\Big] = \mathbb{E}\Big[\langle \Sigma - \Sigma^n, P_{\leq q}\rangle_{\mathrm{HS}}\Big] + \mathbb{E}\Big[\langle \Sigma^n - \Sigma, \hat{P}_{\leq q}\rangle_{\mathrm{HS}}\Big]$$

We now adapt the argument of the proof of Proposition 2.5 in (Reiss & Wahl, 2020) to our setting. We note $\Delta = \Sigma^n - \Sigma$. As $P_{\leq q}$ is deterministic, we have that $\mathbb{E}[\langle \Sigma - \Sigma^n, P_{\leq q}\rangle_{\mathrm{HS}}] = \langle \Sigma - \mathbb{E}[\Sigma^n], P_{\leq q}\rangle_{\mathrm{HS}} = 0$. We obtain:

$$\mathbb{E}\Big[\langle \Sigma - \Sigma^n, P_{\leq q} - \hat{P}_{\leq q}\rangle_{\mathrm{HS}}\Big] = \mathbb{E}\Big[\langle \Delta, \hat{P}_{\leq q}\rangle_{\mathrm{HS}}\Big]$$
$$\leq \mathbb{E}\Big[\sup_{P \in \mathcal{P}_q} \langle \Delta, P\rangle_{\mathrm{HS}}\Big]$$
$$= \mathbb{E}\Big[\sup_{P \in \mathcal{P}_q} \mathrm{Tr}(\Delta P)\Big]$$

A corollary of the min-max Theorem E.4 provides a variational characterization of partial traces:

$$\sup_{V_q \subset \mathcal{H}} \mathrm{Tr}(\Delta P_{V_q}) = \sum_{j=1}^{q} \lambda_j(\Delta), \tag{32}$$

where $V_q$ denotes a subspace of $\mathcal{H}$ of dimension $q$ and $P_{V_q}$ is the orthogonal projection onto $V_q$. Let $P_{\geq j}$ be the orthogonal complement of $P_{<j}$, i.e. $P_{\geq j} = I - P_{<j}$. Combining (32) with Lemma E.5 in Section C.5, we obtain

$$\sup_{P \in \mathcal{P}_q} \mathrm{Tr}(\Delta P) \leq \sum_{j=1}^{q} \lambda_1(P_{\geq j}\Delta P_{\geq j})$$
$$\leq \sum_{j=1}^{q} \max\{|\lambda_1(P_{\geq j}\Delta P_{\geq j})|, |\lambda_2(P_{\geq j}\Delta P_{\geq j})|, \cdots\}$$
$$= \sum_{j=1}^{q} \|P_{\geq j}\Delta P_{\geq j}\|_{\mathrm{op}},$$

where last equality comes from the fact that for self-adjoint bounded linear operators, the spectral radius is equal to the operator norm, denoted $\|\cdot\|_{\mathrm{op}}$. We define the covariance operator $\Sigma_j$ of $P_{\geq j}\phi$ as:

$$\Sigma_j = \mathbb{E}\big[P_{\geq j}\phi \otimes P_{\geq j}\phi\big]$$
$$= P_{\geq j}\mathbb{E}[\phi \otimes \phi]P_{\geq j}$$
$$= P_{\geq j}\Sigma P_{\geq j},$$

where the second equality comes from Lemma C.6. Now, let us notice that $P_{\geq j}$ is constructed from the spectral representation of $\Sigma$, and therefore exactly projects on the subspace spanned by the last eigenvectors of $\Sigma$. This means that we have $\lambda_1(\Sigma_j) = \lambda_j(P_{\geq j}\Sigma P_{\geq j}) = \|\Sigma_j\|_{\mathrm{op}}$. We obtain:

$$\mathrm{tr}(\Sigma_j) = \sum_{k \geq j} \lambda_k.$$

We can also define $\Sigma_j^n$ as the empirical covariance operator of the $P_{\geq j}\phi_i$'s and we have that $\Sigma_j^n = P_{\geq j}\Sigma^n P_{\geq j}$. We can observe that

$$\Sigma_j^n - \Sigma_j = P_{\geq j}\Sigma^n P_{\geq j} - P_{\geq j}\Sigma P_{\geq j}$$
$$= P_{\geq j}\Delta P_{\geq j}.$$

We now need the random variable $\phi$ to be pregaussian for using the moment bound (Koltchinskii & Lounici, 2017).

**Definition C.4.** Let $\phi$ be a weakly square integrable and centered random variable in $\mathcal{H}$. We say $\phi$ is pregaussian if there exists a centered Gaussian random variable in $\mathcal{H}$ which has the same covariance operator than $\phi$.

Assumption 3.1 implies that the covariance operator of $\phi$ is trace-class, which itself implies that $\phi$ is pregaussian. Furthermore, subgaussianity of $\phi$ and $\phi_i$ also imply subgaussianity of $P_{\geq j}\phi$ and $P_{\geq j}\phi_i$. We can now apply Theorem 4 in (Koltchinskii & Lounici, 2017) to $\Sigma_j^n - \Sigma_j$:

$$\mathbb{E}[\|P_{\geq j}\Delta P_{\geq j}\|_{\mathrm{op}}] = \mathbb{E}[\|\Sigma_j - \Sigma_j^n\|_{\mathrm{op}}]$$

$$\lesssim \|\Sigma_j\|_{\mathrm{op}} \max\left\{\sqrt{\frac{(\mathbb{E}\|P_{\geq j}\phi\|_{\mathcal{H}})^2}{\|\Sigma_j\|_{\mathrm{op}} n}}, \frac{(\mathbb{E}\|P_{\geq j}\phi\|_{\mathcal{H}})^2}{\|\Sigma_j\|_{\mathrm{op}} n}\right\}$$

$$= \max\left\{\sqrt{\frac{\|\Sigma_j\|_{\mathrm{op}}(\mathbb{E}\|P_{\geq j}\phi\|_{\mathcal{H}})^2}{n}}, \frac{(\mathbb{E}\|P_{\geq j}\phi\|_{\mathcal{H}})^2}{n}\right\}$$

$$= \max\left\{\sqrt{\frac{\lambda_j(\mathbb{E}\|P_{\geq j}\phi\|_{\mathcal{H}})^2}{n}}, \frac{(\mathbb{E}\|P_{\geq j}\phi\|_{\mathcal{H}})^2}{n}\right\}$$

Now, using that $(\mathbb{E}\|X\|)^2 \leq \mathbb{E}\|X\|^2$ and that $\mathrm{Tr}(\mathbb{E}[X \otimes X]) = \mathbb{E}\|X\|^2$:

$$\mathbb{E}[\|P_{\geq j}\Delta P_{\geq j}\|_{\mathrm{op}}] \lesssim \max\left\{\sqrt{\frac{\lambda_j\mathbb{E}\|P_{\geq j}\phi\|_{\mathcal{H}}^2}{n}}, \frac{\mathbb{E}\|P_{\geq j}\phi\|_{\mathcal{H}}^2}{n}\right\}$$

$$= \max\left\{\sqrt{\frac{\lambda_j\mathrm{Tr}(\Sigma_j)}{n}}, \frac{\mathrm{Tr}(\Sigma_j)}{n}\right\}$$

$$= \max\left\{\sqrt{\frac{\lambda_j\sum_{k\geq j}\lambda_k}{n}}, \frac{\sum_{k\geq j}\lambda_k}{n}\right\}$$

Putting everything together, we can finally bound our first term:

$$\mathbb{E}\Big[\langle\Sigma - \Sigma^n, P_{\leq q} - \hat{P}_{\leq q}\rangle_{\mathrm{HS}}\Big] \leq \mathbb{E}\Big[\sum_{j=1}^{q}\|P_{\geq j}\Delta P_{\geq j}\|_{\mathrm{op}}\Big]$$

$$\lesssim \sum_{j=1}^{q}\max\left\{\sqrt{\frac{\lambda_j\sum_{k\geq j}\lambda_k}{n}}, \frac{\sum_{k\geq j}\lambda_k}{n}\right\}.$$

### C.3. Proof of Lemma C.3

We start by writing:

$$\mathbb{E}\Big[\langle\Sigma^n - \hat{\Sigma}, P_{\leq q} - \hat{P}_{\leq q}\rangle_{\mathrm{HS}}\Big] = \mathbb{E}\Big[\langle\Sigma^n - \hat{\Sigma}, P_{\leq q}\rangle_{\mathrm{HS}} - \langle\Sigma^n - \hat{\Sigma}, \hat{P}_{\leq q}\rangle_{\mathrm{HS}}\Big]$$

$$\leq 2\sup_{P\in\mathcal{P}_q}\mathbb{E}\Big[|\langle\Sigma^n - \hat{\Sigma}, P\rangle_{\mathrm{HS}}\Big]$$

Using Cauchy-Schwarz, we have that

$$\mathbb{E}\Big[\langle\Sigma^n - \hat{\Sigma}, P_{\leq q} - \hat{P}_{\leq q}\rangle_{\mathrm{HS}}\Big] \leq 2\sup_{P\in\mathcal{P}_q}\mathbb{E}\Big[\|\Sigma^n - \hat{\Sigma}\|_{\mathrm{HS}}\|P\|_{\mathrm{HS}}\Big]$$

$$= 2\sqrt{q}\,\mathbb{E}\|\Sigma^n - \hat{\Sigma}\|_{\mathrm{HS}}$$

From Lemma A.2, we have that under Assumption 3.1,

$$\mathbb{E}\|\Sigma^n - \hat{\Sigma}\|_{\mathrm{HS}} \leq 2R^{1/4}r_m(\Phi).$$

Then, under Assumption 3.1, we have:

$$\mathbb{E}\Big[\langle \Sigma^n - \hat{\Sigma}, P_{\leq q} - \hat{P}_{\leq q}\rangle_{\mathrm{HS}}\Big] \leq 4R^{1/4}\sqrt{q}r_m(\Phi).$$

## C.4. Proof of Corollary 3.9

We have a standard identity for the population and empirical projectors

$$\|P_{\leq q} - \hat{P}_{\leq q}\|_{\mathrm{HS}}^2 = 2\|(I - P_{\leq q})\hat{P}_{\leq q}\|_{\mathrm{HS}}^2 = 2\|\sum_{k>q} P_k\hat{P}_{\leq q}\|_{\mathrm{HS}}^2 = 2\sum_{k>q}\|P_k\hat{P}_{\leq q}\|_{\mathrm{HS}}^2.$$

Using Lemma 2.6 in (Reiss & Wahl, 2020), the PCA excess risk satisfies

$$\mathcal{E}_q^{\mathrm{PCA}} = \sum_{j\leq q}(\lambda_j - \lambda_{q+1})\|P_j\hat{P}_{>q}\|_{\mathrm{HS}}^2 + \sum_{k>q}(\lambda_{q+1} - \lambda_k)\|P_k\hat{P}_{\leq q}\|_{\mathrm{HS}}^2$$

In particular,

$$\mathcal{E}_q^{\mathrm{PCA}} \geq \sum_{k>q}(\lambda_{q+1} - \lambda_k)\|P_k\hat{P}_{\leq q}\|_{\mathrm{HS}}^2 \geq (\lambda_{q+1} - \lambda_q)\sum_{k>q}\|P_k\hat{P}_{\leq q}\|_{\mathrm{HS}}^2,$$

as we assumed $\lambda_q \geq \lambda_{q+1}$. Combining the two, we obtain

$$\|P_{\leq q} - \hat{P}_{\leq q}\|_{\mathrm{HS}}^2 \leq \frac{2\mathcal{E}_q^{\mathrm{PCA}}}{\lambda_{q+1} - \lambda_q},$$

which is a variant of the Davis-Kahan sin-$\theta$ theorem (Davis & Kahan, 1969).

## C.5. Technical details

**Lemma C.5.** *Let $\phi$ be a centered random variable in the Hilbert space $\mathcal{H}$ and $\Sigma = \mathbb{E}[\phi \otimes \phi]$ its covariance operator. For an orthogonal projection $P : \mathcal{H} \to \mathcal{H}$, we have:*

$$\mathbb{E}\|\phi - P\phi\|_{\mathcal{H}}^2 = \langle \Sigma, I - P\rangle_{\mathrm{HS}}$$

*Proof.*

$$
\begin{aligned}
\mathbb{E}\|\phi - P\phi\|_{\mathcal{H}}^2 &= \mathbb{E}\Big[\|\phi\|_{\mathcal{H}}^2 + \|P\phi\|_{\mathcal{H}}^2 - 2\langle \phi, P\phi\rangle_{\mathcal{H}}\Big]\\
&= \mathbb{E}\Big[\langle \phi, \phi\rangle_{\mathcal{H}} + \langle P\phi, P\phi\rangle_{\mathcal{H}} - 2\langle \phi, P\phi\rangle_{\mathcal{H}}\Big]\\
&= \mathbb{E}\Big[\langle \phi, \phi\rangle_{\mathcal{H}} + \langle \phi, PP\phi\rangle_{\mathcal{H}} - 2\langle \phi, P\phi\rangle_{\mathcal{H}}\Big]\\
&= \mathbb{E}\Big[\langle \phi, \phi\rangle_{\mathcal{H}} + \langle \phi, P\phi\rangle_{\mathcal{H}} - 2\langle \phi, P\phi\rangle_{\mathcal{H}}\Big]\\
&= \mathbb{E}\Big[\langle \phi, \phi\rangle_{\mathcal{H}} - \langle \phi, P\phi\rangle_{\mathcal{H}}\Big]\\
&= \mathbb{E}\Big[\langle \phi, (I - P)\phi\rangle_{\mathcal{H}}\Big]\\
&\stackrel{(33)}{=} \mathbb{E}\Big[\mathrm{Tr}\big((I - P)(\phi \otimes \phi)\big)\Big]\\
&= \mathbb{E}\Big[\langle I - P, \phi \otimes \phi\rangle_{\mathrm{HS}}\Big]\\
&= \langle I - P, \Sigma\rangle_{\mathrm{HS}}.
\end{aligned}
$$

$\square$

**Lemma C.6.** *Let $P : \mathcal{H} \to \mathcal{H}$ be an orthogonal projection and $\phi$ be a random variable on $\mathcal{H}$. Then,*

$$\mathbb{E}[P\phi \otimes P\phi] = P\mathbb{E}[\phi \otimes \phi]P.$$

*Proof.* Using the definition of the tensor product, let $f \in \mathcal{H}$. Then,

$$
\begin{aligned}
\mathbb{E}[P\phi \otimes P\phi]f &= \mathbb{E}[\langle P\phi, f \rangle_{\mathcal{H}} P\phi] \\
&= \mathbb{E}[\langle \phi, Pf \rangle_{\mathcal{H}} P\phi] \\
&= \mathbb{E}[P\langle \phi, Pf \rangle_{\mathcal{H}} \phi] \\
&= P\mathbb{E}[\langle \phi, Pf \rangle_{\mathcal{H}} \phi] \\
&= P\mathbb{E}[(\phi \otimes \phi)(Pf)] \\
&= P\mathbb{E}[\phi \otimes \phi]Pf
\end{aligned}
$$

$\square$

### C.5.1. PROOF OF COROLLARY 3.10

We focus on the first term of the risk bound in Theorem 3.8, that is:

$$
\sum_{j=1}^{q} \max \left\{ \sqrt{\frac{\lambda_j \sum_{k \geq j} \lambda_k}{n}}, \frac{\sum_{k \geq j} \lambda_k}{n} \right\}.
$$

**Polynomial decay.** The case of polynomially decaying eigenvalues corresponds to $\lambda_j \asymp j^{-\alpha}$ for some $\alpha > 1$ and for all $j \geq 1$.

As $\sum_{k \geq j} \lambda_k \asymp \sum_{k \geq j} k^{-\alpha} \asymp j^{1-\alpha}$, we have that:

$$
\sqrt{\frac{\lambda_j \sum_{k \geq j} \lambda_k}{n}} \asymp \sqrt{\frac{j^{-\alpha} j^{1-\alpha}}{n}} = \sqrt{\frac{j^{1-2\alpha}}{n}}, \quad \text{and} \quad \frac{\sum_{k \geq j} \lambda_k}{n} \asymp \frac{j^{1-\alpha}}{n}.
$$

Notice that the two terms are equal when $j = n$ and the square root term is larger when $j \leq n$. Therefore, if we assume $q \leq n$, which is reasonable as $q$ is usually small compared to $n$, we have that:

$$
\sum_{j=1}^{q} \max \left\{ \sqrt{\frac{\lambda_j \sum_{k \geq j} \lambda_k}{n}}, \frac{\sum_{k \geq j} \lambda_k}{n} \right\} \asymp \sum_{j=1}^{q} \sqrt{\frac{j^{1-2\alpha}}{n}} \asymp \frac{1}{\sqrt{n}} \begin{cases} 1, & \alpha > 3/2, \\ \log(q), & \alpha = 3/2, \\ q^{\frac{3}{2}-\alpha}, & \alpha < 3/2. \end{cases}
$$

Therefore, assuming a polynomial decay in the eigenvalues $\lambda_j \asymp j^{-\alpha}$ for $\alpha > 3/2$ and $q \leq n$, we have that the first term of the risk is bounded by a term of order $\sqrt{1/n}$.

**Exponential decay.** Let us now assume that the eigenvalues decay exponentially, i.e. $\lambda_j \asymp e^{-\alpha j}$ for some $\alpha > 0$ and for all $j \geq 1$. In that case, we have that $\sum_{k \geq j} \lambda_k \asymp \sum_{k \geq j} e^{-\alpha k} \asymp \frac{e^{-\alpha j}}{1-e^{-\alpha}}$. This gives:

$$
\sqrt{\frac{\lambda_j \sum_{k \geq j} \lambda_k}{n}} \asymp \sqrt{\frac{e^{-\alpha j} e^{-\alpha j}}{n(1-e^{-\alpha})}} = \sqrt{\frac{e^{-2\alpha j}}{n(1-e^{-\alpha})}}, \quad \text{and} \quad \frac{\sum_{k \geq j} \lambda_k}{n} \asymp \frac{e^{-\alpha j}}{n(1-e^{-\alpha})}.
$$

The two terms are equal when $n = \frac{1}{1-e^{-\alpha}}$. We therefore have that:

$$
\sum_{j=1}^{q} \max \left\{ \sqrt{\frac{\lambda_j \sum_{k \geq j} \lambda_k}{n}}, \frac{\sum_{k \geq j} \lambda_k}{n} \right\} \asymp \begin{cases} \frac{e^{-\alpha}(1-e^{-\alpha q})}{n(1-e^{-\alpha})} \asymp \frac{1}{n}, & n \leq \frac{1}{1-e^{-\alpha}}, \\ \frac{e^{-\alpha}(1-e^{-\alpha q})}{\sqrt{n}(1-e^{-\alpha})^{3/2}} \asymp \frac{1}{\sqrt{n}}, & n \geq \frac{1}{1-e^{-\alpha}}. \end{cases}
$$

Hence, assuming an exponential decay in the eigenvalues $\lambda_j \asymp e^{-\alpha j}$ for $\alpha > 0$, we have that the first term of the risk is bounded by a term of order $1/n$ as soon as $n$ is smaller than $\frac{1}{1-e^{-\alpha}}$, and $1/\sqrt{n}$ otherwise.

## D. Gaussian measures

We now study the special case where $\mu = \mathcal{N}(\boldsymbol{b}, \boldsymbol{S})$, with $\boldsymbol{b} \in \mathbb{R}^d$ a random vector and $\boldsymbol{S} = \mathrm{diag}(s_1^2, \cdots, s_d^2)$ a random diagonal matrix with entries $s_1^2, \cdots, s_d^2 > 0$. We assume the coordinates of $\boldsymbol{b}$ (resp. $\boldsymbol{s} = (s_1, \cdots, s_d)$) are independent random variables and we also assume that $\boldsymbol{b}$ and $\boldsymbol{s}$ are independent. In this section, we study the spectrum of the covariance operator corresponding to the different embeddings $\Phi(\mu)$ in this special gaussian case.

**Proposition D.1** (KME). *Let $\mu = \mathcal{N}(\boldsymbol{b}, \boldsymbol{S})$ with $\boldsymbol{S} = \mathrm{diag}(s_1^2, \cdots, s_d^2)$ and $\boldsymbol{b} = (\boldsymbol{b}_1, \cdots, \boldsymbol{b}_d)$, where the $\boldsymbol{b}_i$'s are mutually independent, the $\boldsymbol{s}_i$'s are mutually independent, and $\boldsymbol{b}$ and $\boldsymbol{s}$ are independent. Then, the KME of $\mu$ is*

$$\forall x \in \mathcal{X}, \quad \Phi^{\mathrm{KME}}(\mu)(x) = x^T \boldsymbol{b},$$

*Furthermore, the covariance operator of $\Phi^{\mathrm{KME}}(\mu)$ for the linear kernel admits the following eigendecomposition:*

$$\Sigma = \sum_{i=1}^{d} \mathrm{Var}(\boldsymbol{b}_i) f_i \otimes f_i,$$

*where for all $1 \leq i \leq d$, $f_i(x) = x_i$.*

*Proof.* Let us recall that, given a positive definite kernel $k$ and corresponding RKHS $\mathcal{H}_k$, the KME of a probability measure $\mu$ is:

$$\Phi^{\mathrm{KME}}(\mu) = \int k(x, \cdot)\, d\mu(x) \in \mathcal{H}_k.$$

For this example, we choose the linear kernel $k(x, y) = x^T y$. In this case, the KME boils down to:

$$\forall x \in \mathcal{X}, \quad \Phi^{\mathrm{KME}}(\mu)(x) = \int x^T y\, d\mu(y) = x^T \boldsymbol{b}.$$

For $1 \leq i \leq d$, let $f_i(x) = x_i$. Then $\Phi^{\mathrm{KME}}(\mu)(x) = \sum_{i=1}^{d} \boldsymbol{b}_i x_i = \sum_{i=1}^{d} \boldsymbol{b}_i f_i(x)$. Let us compute its covariance operator

$$\begin{aligned}
\Sigma &= \mathbb{E}\big[\Phi^{\mathrm{KME}}(\mu) \otimes \Phi^{\mathrm{KME}}(\mu)\big] - \mathbb{E}\big[\Phi^{\mathrm{KME}}(\mu)\big] \otimes \mathbb{E}\big[\Phi^{\mathrm{KME}}(\mu)\big] \\
&= \mathbb{E}\left[\sum_{i=1}^{d} \boldsymbol{b}_i f_i \otimes \sum_{j=1}^{d} \boldsymbol{b}_j f_j\right] - \mathbb{E}\left[\sum_{i=1}^{d} \boldsymbol{b}_i f_i\right] \otimes \mathbb{E}\left[\sum_{j=1}^{d} \boldsymbol{b}_j f_j\right] \\
&= \sum_{i=1}^{d} \sum_{j=1}^{d} \mathbb{E}[\boldsymbol{b}_i \boldsymbol{b}_j] f_i \otimes f_j - \sum_{i=1}^{d} \sum_{j=1}^{d} \mathbb{E}[\boldsymbol{b}_i]\, \mathbb{E}[\boldsymbol{b}_j] f_i \otimes f_j \\
&= \sum_{i=1}^{d} \sum_{j=1}^{d} \Big(\mathbb{E}[\boldsymbol{b}_i \boldsymbol{b}_j] - \mathbb{E}[\boldsymbol{b}_i]\, \mathbb{E}[\boldsymbol{b}_j]\Big) f_i \otimes f_j \\
&= \sum_{i=1}^{d} \sum_{j=1}^{d} \mathrm{Cov}(\boldsymbol{b})_{ij} f_i \otimes f_j \\
&= \sum_{i=1}^{d} \mathrm{Var}(\boldsymbol{b}_i) f_i \otimes f_i.
\end{aligned}$$

The $(f_i)_{1 \leq i \leq d}$ are therefore the eigenfunctions of $\Sigma$ with corresponding eigenvalues $\mathrm{Var}(\boldsymbol{b}_i)$. Furthermore, one can check that $\forall 1 \leq i \leq d, \|f_i\|_{\mathcal{H}_k} = 1$ and $\forall 1 \leq j \leq d$ with $j \neq i, \langle f_i, f_j \rangle_{\mathcal{H}_k} = 0$. $\square$

**Proposition D.2** (LOT). *Let $\mu = \mathcal{N}(\boldsymbol{b}, \boldsymbol{S})$ with $\boldsymbol{S} = \mathrm{diag}(s_1^2, \cdots, s_d^2)$ and $\boldsymbol{b} = (\boldsymbol{b}_1, \cdots, \boldsymbol{b}_d)$, where the $\boldsymbol{b}_i$'s are mutually independent, the $\boldsymbol{s}_i$'s are mutually independent, and $\boldsymbol{b}$ and $\boldsymbol{s}$ are independent. Then, the LOT embedding of $\mu$ with reference measure $\rho = \mathcal{N}(0, I_d)$ is*

$$\forall x \in \mathcal{X}, \quad \Phi^{\mathrm{LOT}}(\mu)(x) = \sum_{i=1}^{d} (\boldsymbol{s}_i - 1) x_i e_i + \boldsymbol{b}.$$

*Furthermore, the covariance operator of $\Phi^{\mathrm{LOT}}(\boldsymbol{\mu})$ admits the following eigendecomposition:*

$$\Sigma = \sum_{i=1}^{d} \mathrm{Var}(\boldsymbol{b}_i) f_i \otimes f_i + \sum_{i=1}^{d} \mathrm{Var}(\boldsymbol{s}_i) g_i \otimes g_i,$$

*where for all $1 \leq i \leq d$, $f_i(x) = e_i$ and $g_i(x) = x_i e_i$.*

*Proof.* The LOT embedding with reference measure $\rho$ of a probability measure $\mu$ is:

$$\Phi^{\mathrm{LOT}}(\mu) = T_\mu - \mathrm{Id} \in L^2(\rho).$$

In the Gaussian case, with $\rho = \mathcal{N}(0, I_d)$ and $\mu = \mathcal{N}(\boldsymbol{b}, \boldsymbol{S})$, the LOT embedding boils down to:

$$\Phi^{\mathrm{LOT}}(\mu)(x) = (\boldsymbol{S}^{1/2} - I_d)x + \boldsymbol{b} = \sum_{i=1}^{d} (\boldsymbol{s}_i - 1) x_i e_i + \boldsymbol{b}$$

For $1 \leq i \leq d$, let $f_i(x) = e_i$ and $g_i(x) = x_i e_i$.

$$\Phi^{\mathrm{LOT}}(\boldsymbol{\mu}) = \sum_{i=1}^{d} \boldsymbol{b}_i f_i + \sum_{i=1}^{d} (\boldsymbol{s}_i - 1) g_i.$$

We can now compute the covariance operator $\Sigma$ of the random variable $\Phi^{\mathrm{LOT}}(\boldsymbol{\mu})$.

$$\Sigma = \mathbb{E}\Big[\Phi^{\mathrm{LOT}}(\boldsymbol{\mu}) \otimes \Phi^{\mathrm{LOT}}(\boldsymbol{\mu})\Big] - \mathbb{E}\Big[\Phi^{\mathrm{LOT}}(\boldsymbol{\mu})\Big] \otimes \mathbb{E}\Big[\Phi^{\mathrm{LOT}}(\boldsymbol{\mu})\Big]$$

$$= \mathbb{E}\Big[\big(\sum_{i=1}^{d} \boldsymbol{b}_i f_i + \sum_{i=1}^{d} (\boldsymbol{s}_i - 1) g_i\big) \otimes \big(\sum_{j=1}^{d} \boldsymbol{b}_j f_j + \sum_{j=1}^{d} (\boldsymbol{s}_j - 1) g_j\big)\Big] - \mathbb{E}\Big[\sum_{i=1}^{d} \boldsymbol{b}_i f_i + \sum_{i=1}^{d} (\boldsymbol{s}_i - 1) g_i\Big] \otimes \mathbb{E}\Big[\sum_{j=1}^{d} \boldsymbol{b}_j f_j + \sum_{j=1}^{d} (\boldsymbol{s}_j - 1) g_j\Big]$$

Since $\boldsymbol{b}$ and $\boldsymbol{s}$ are independent, all cross-terms between $f_i$ and $g_j$ vanish and we obtain:

$$\Sigma = \sum_{i=1}^{d} \mathbb{E}[\boldsymbol{b}_i^2] f_i \otimes f_i + \sum_{i=1}^{d} \mathbb{E}[(\boldsymbol{s}_i - 1)^2] g_i \otimes g_i - \sum_{i=1}^{d} \mathbb{E}[\boldsymbol{b}_i]^2 f_i \otimes f_i - \sum_{i=1}^{d} \mathbb{E}[(\boldsymbol{s}_i - 1)]^2 g_i \otimes g_i$$

$$= \sum_{i=1}^{d} \mathrm{Var}(\boldsymbol{b}_i) f_i \otimes f_i + \sum_{i=1}^{d} \mathrm{Var}(\boldsymbol{s}_i - 1) g_i \otimes g_i$$

$$= \sum_{i=1}^{d} \mathrm{Var}(\boldsymbol{b}_i) f_i \otimes f_i + \sum_{i=1}^{d} \mathrm{Var}(\boldsymbol{s}_i) g_i \otimes g_i$$

One can check that $\|f_i\|_{L^2(\rho)} = \|g_i\|_{L^2(\rho)} = 1$, and that $\forall i \neq j, \langle f_i, g_j \rangle_{L^2(\rho)} = \langle f_i, f_j \rangle_{L^2(\rho)} = \langle g_i, g_j \rangle_{L^2(\rho)} = \langle f_i, g_i \rangle_{L^2(\rho)} = 0$.

$\square$

**Proposition D.3** (SW). *Let $\mu = \mathcal{N}(\boldsymbol{b}, s^2 I_d)$ with $\boldsymbol{b} = (\boldsymbol{b}_1, \cdots, \boldsymbol{b}_d)$, where the $\boldsymbol{b}_i$'s are mutually independent, and $\boldsymbol{b}$ and $\boldsymbol{s}$ are independent. Then, the SW embedding of $\boldsymbol{\mu}$ is*

$$\forall t \in [0, 1], \forall \theta \in \mathbb{S}^{d-1}, \quad \Phi^{\mathrm{SW}}(\boldsymbol{\mu})(t, \theta) = \theta^T \boldsymbol{b} + \sqrt{2}s\, \mathrm{erf}^{-1}(2t - 1).$$

*Furthermore, the covariance operator of $\Phi^{\mathrm{SW}}(\boldsymbol{\mu})$ admits the following eigendecomposition:*

$$\Sigma = \frac{1}{d} \sum_{i=1}^{d} \mathrm{Var}(\boldsymbol{b}_i) f_i \otimes f_i + \mathrm{Var}(\boldsymbol{s}) g \otimes g,$$

*where $g(t, \theta) = \sqrt{2}\mathrm{erf}^{-1}(2t - 1)$ and for all $1 \leq i \leq d$, $f_i(t, \theta) = \sqrt{d}_i$.*

*Proof.* The Sliced Wasserstein embedding of a probability measure is:

$$\Phi^{\mathrm{SW}}(\mu)(t,\theta) = F_{\mu,\theta}^{-1}(t),$$

where $F_{\mu,\theta}^{-1}$ is the quantile function of $P_\#^\theta \mu$. When $\boldsymbol{\mu} = \mathcal{N}(\boldsymbol{b}, \boldsymbol{S})$, the projection has a closed form $P_\#^\theta \boldsymbol{\mu} = \mathcal{N}(\theta^T \boldsymbol{b}, \theta^T \boldsymbol{S}\theta)$. Furthermore, for a one-dimensional Gaussian measure $\nu = \mathcal{N}(m, \sigma)$ with $m \in \mathbb{R}$ the mean and $\sigma^2 \in \mathbb{R}_+$ the variance, the quantile function also has a closed form $F_\nu^{-1}(t) = \theta^T b + \sigma\sqrt{2}\mathrm{erf}^{-1}(2t-1)$, where $\mathrm{erf}^{-1}$ is the inverse error function. This gives:

$$\forall \theta \in \mathbb{S}^{d-1}, \forall t \in [0,1], \qquad F_{\mu,\theta}^{-1}(t) = \theta^T \boldsymbol{b} + \sqrt{2\theta^T \boldsymbol{S}\theta}\,\mathrm{erf}^{-1}(2t-1).$$

If $\boldsymbol{S} = \boldsymbol{s}^2 I_d$, then

$$\forall \theta \in \mathbb{S}^{d-1}, \forall t \in [0,1], \qquad \Phi^{\mathrm{SW}}(\boldsymbol{\mu})(t,\theta) = \theta^T \boldsymbol{b} + \sqrt{2}\boldsymbol{s}\,\mathrm{erf}^{-1}(2t-1).$$

For $1 \le i \le d$, let $f_i(t,\theta) = \sqrt{d}\theta_i$ and $g(t,\theta) = \sqrt{2}\mathrm{erf}^{-1}(2t-1)$. We can rewrite the embedding as:

$$\Phi^{\mathrm{SW}}(\boldsymbol{\mu})(t,\theta) = \frac{1}{\sqrt{d}}\sum_{i=1}^d \boldsymbol{b}_i f_i(\theta) + \boldsymbol{s}g(t,\theta).$$

We can now compute the covariance operator $\Sigma$ of the random variable $\Phi^{\mathrm{SW}}(\boldsymbol{\mu})$.

$$\Sigma = \mathbb{E}\Big[\Phi^{\mathrm{SW}}(\boldsymbol{\mu}) \otimes \Phi^{\mathrm{SW}}(\boldsymbol{\mu})\Big] - \mathbb{E}\Big[\Phi^{\mathrm{SW}}(\boldsymbol{\mu})\Big] \otimes \mathbb{E}\Big[\Phi^{\mathrm{SW}}(\boldsymbol{\mu})\Big]$$

$$= \mathbb{E}\Big[\big(\frac{1}{\sqrt{d}}\sum_{i=1}^d \boldsymbol{b}_i f_i + \boldsymbol{s}g\big) \otimes \big(\frac{1}{\sqrt{d}}\sum_{j=1}^d \boldsymbol{b}_j f_j + \boldsymbol{s}g\big)\Big] - \mathbb{E}\Big[\frac{1}{\sqrt{d}}\sum_{i=1}^d \boldsymbol{b}_i f_i + \boldsymbol{s}g\Big] \otimes \mathbb{E}\Big[\frac{1}{\sqrt{d}}\sum_{j=1}^d \boldsymbol{b}_j f_j + \boldsymbol{s}g\Big]$$

Since $\boldsymbol{b}$ and $\boldsymbol{s}$ are independent, all cross-terms between $f_i$ and $g$ vanish and we obtain:

$$\Sigma = \frac{1}{d}\sum_{i=1}^d \mathbb{E}[\boldsymbol{b}_i^2]f_i \otimes f_i + \mathbb{E}[\boldsymbol{s}^2]g \otimes g - \frac{1}{d}\sum_{i=1}^d \mathbb{E}[\boldsymbol{b}_i]^2 f_i \otimes f_i - \mathbb{E}[\boldsymbol{s}]^2 g \otimes g$$

$$= \frac{1}{d}\sum_{i=1}^d \mathrm{Var}(\boldsymbol{b}_i)f_i \otimes f_i + \mathrm{Var}(\boldsymbol{s})g \otimes g.$$

One can check that $\|f_i\|_{L^2([0,1]\times\mathbb{S}^{d-1})} = \|g\|_{L^2([0,1]\times\mathbb{S}^{d-1})} = 1$, and that $\forall i \ne j, \langle f_i, g \rangle_{L^2([0,1]\times\mathbb{S}^{d-1})} = \langle f_i, f_j \rangle_{L^2([0,1]\times\mathbb{S}^{d-1})} = 0$.

$\square$

# E. Operator theory on Hilbert spaces

In this section, we take a closer look at linear continuous maps on Hilbert spaces, often called *bounded linear operators*. For a more general treatment, we refer to (Conway, 1990) and (Reed & Simon, 1978). Let $\mathcal{H}$ be a Hilbert space with inner-product $\langle \cdot, \cdot \rangle_\mathcal{H}$ and norm $\|\cdot\|_\mathcal{H}$. For a bounded linear operator $A : \mathcal{H} \to \mathcal{H}$, the operator norm $\|\cdot\|_{\mathrm{op}}$ can be defined in several ways:

$$\|A\|_{\mathrm{op}} = \inf_{f \in \mathcal{H}}\{c \ge 0 \;:\; \|Af\|_\mathcal{H} \le c\|f\|_\mathcal{H}\}$$

$$= \sup_{f \in \mathcal{H}}\{\|Af\|_\mathcal{H} : \|f\|_\mathcal{H} \le 1\}$$

$$= \sup_{f \in \mathcal{H}}\{\|Af\|_\mathcal{H} \;:\; \|f\|_\mathcal{H} = 1\}$$

$$= \sup_{f \in \mathcal{H}}\Big\{\frac{\|Af\|_\mathcal{H}}{\|f\|_\mathcal{H}} \;:\; f \ne 0\Big\}$$

A useful definition for a positive bounded linear operator $A : \mathcal{H} \to \mathcal{H}$ is its trace:

$$\mathrm{Tr}(A) = \sum_{i \geq 0} \langle Ae_i, e_i \rangle_{\mathcal{H}},$$

where $e_1, e_2, \cdots$ is an orthonormal basis of $\mathcal{H}$. A bounded linear operator $A$ is then called *trace class* when

$$\mathrm{Tr}(|A|) < +\infty.$$

Another norm defined for a bounded operator $A : \mathcal{H} \to \mathcal{H}$ is the *Hilbert-Schmidt norm:*

$$\|A\|_{\mathrm{HS}}^2 = \sum_{i \geq 0} \|Ae_i\|_{\mathcal{H}}^2,$$

where $e_1, e_2, \cdots$ is an orthonormal basis of $\mathcal{H}$. For finite-dimensional Euclidean spaces, the Hilbert-Schmidt norm corresponds to the Frobenius norm. An *Hilbert-Schmidt operator* is a bounded linear operator which has finite Hilbert-Schmidt norm. One can then define the *Hilbert-Schmidt inner product* between two Hilbert-Schmidt operators $A$ and $B$ as

$$\langle A, B \rangle_{\mathrm{HS}} = \mathrm{Tr}(B^*A) = \sum_{i \geq 0} \langle Ae_i, Be_i \rangle_{\mathcal{H}},$$

where $B^*$ denotes the adjoint operator of $B$. For $f, g \in \mathcal{H}$, define the operator $f \otimes g : \mathcal{H} \to \mathcal{H}$ such that for $h \in \mathcal{H}$, $(f \otimes g)(h) = \langle f, h \rangle_{\mathcal{H}} g$. This rank-one operator is Hilbert-Schmidt and satisfies

$$\mathrm{Tr}(A(f \otimes g)) = \langle Ag, f \rangle_{\mathcal{H}} \tag{33}$$

for any bounded linear operator $A : \mathcal{H} \to \mathcal{H}$.

We now introduce compact operators, which are closely analogous to matrices acting on finite-dimensional spaces.

**Definition E.1.** Let $A : \mathcal{H} \to \mathcal{H}$ be a bounded linear operator. The operator $A$ is compact if for every bounded set $B \subset \mathcal{H}$, the closure of $A(B)$ is compact in $\mathcal{H}$.

Equivalently, the operator $A$ is compact if for every bounded sequence $f_n$ in $\mathcal{H}$, $Af_n$ contains a convergent subsequence. One can summarize the relationships between some classes of operators for an infinite-dimensional Hilbert space $\mathcal{H}$ in the following way:

$$\{\text{finite rank}\} \subseteq \{\text{trace class}\} \subseteq \{\text{Hilbert} - \text{Schmidt}\} \subseteq \{\text{compact}\}$$

As this work deals with PCA, we are particularly concerned with projections, which are linear operators $P : \mathcal{H} \to \mathcal{H}$ that verify $P^2 = P$. The following definition allows us to introduce orthogonal projections.

**Definition E.2.** An operator $A : \mathcal{H} \to \mathcal{H}$ is called self-adjoint if for every $f, g \in \mathcal{H}$,

$$\langle Af, g \rangle_{\mathcal{H}} = \langle f, Ag \rangle_{\mathcal{H}}$$

In Hilbert spaces, a projection is orthogonal if and only if it is self-adjoint. An orthogonal projection is bounded operator.

Let us now discuss spectral theory. In linear algebra, a real symmetric matrix is diagonalizable via an orthogonal matrix. This result, known as the spectral theorem, can be extended to self-adjoint compact operators.

**Theorem E.3.** *Let $A : \mathcal{H} \to \mathcal{H}$ be a self-adjoint compact operator. Then, there is an orthonormal basis $e_1, e_2, \cdots$ of $\mathcal{H}$ consisting of eigenvectors of A. Each eigenvalue is real.*

Therefore, for any self-adjoint compact operator $A$, there exists an orthonornmal basis $e_1, e_2, \cdots \in \mathcal{H}$ and eigenvalues $\lambda_1, \lambda_2, \cdots \in \mathbb{R}$ such that

$$A = \sum_{i \geq 0} \lambda_i \cdot e_i \otimes e_i,$$

The following result is known under the name min-max theorem or Courant-Fischer-Weyl min-max principle and gives a variational characterization of eigenvalues of compact, self-adjoint operators on Hilbert spaces.

**Theorem E.4.** *Let $A$ be a compact, self-adjoint operator on a Hilbert space $\mathcal{H}$ with eigenvalues $\lambda_1 \geq \lambda_2 \geq \cdots$. Then,*

$$\lambda_k = \max_{V_k} \min_{\substack{v \in V_k \\ \|v\|_{\mathcal{H}} = 1}} \langle Av, v \rangle_{\mathcal{H}}$$

*and,*

$$\lambda_k = \min_{V_{k-1}} \max_{\substack{v \perp V_{k-1} \\ \|v\|_{\mathcal{H}} = 1}} \langle Av, v \rangle_{\mathcal{H}},$$

*where $V_k \subseteq \mathcal{H}$ denotes a $k$-dimensional subspace of $\mathcal{H}$.*

From this theorem, we deduce a corollary which will be useful in our proofs.

**Corollary E.5.** *Let $A : \mathcal{H} \to \mathcal{H}$ be a self-adjoint, compact operator. Let $u_1, u_2, \cdots$ be an orthonormal basis of $\mathcal{H}$ and define the rank-one projectors $P_k = u_k \otimes u_k$. Define $P_{<j} = \sum_{1 \leq k < j} P_k$ and $P_{\geq j} = \sum_{k \geq j} P_k$. Then,*

$$\lambda_j(A) \leq \lambda_1(P_{\geq j} A P_{\geq j})$$

*Proof.* The min-max Theorem E.4 applied to $k = 1$ implies that

$$\lambda_1(P_{\geq j} A P_{\geq j}) = \min_{V_0 \subset \mathcal{H}} \max_{\substack{u \perp V_0 \\ \|u\|_{\mathcal{H}} = 1}} \langle P_{\geq j} A P_{\geq j} u, u \rangle_{\mathcal{H}},$$

where $V_0$ is a $0$-dimensional subspace of $\mathcal{H}$. It is therefore necessarily the set $\{0_{\mathcal{H}}\}$.

$$\lambda_1(P_{\geq j} A P_{\geq j}) = \max_{\substack{u \in \mathcal{H} \\ \|u\|_{\mathcal{H}} = 1}} \langle P_{\geq j} A P_{\geq j} u, u \rangle_{\mathcal{H}}$$

$$= \max_{\substack{u \in \mathcal{H} \\ \|u\|_{\mathcal{H}} = 1}} \langle A P_{\geq j} u, P_{\geq j} u \rangle_{\mathcal{H}}$$

$$= \max_{\substack{u \in \mathcal{H} \\ \|u\|_{\mathcal{H}} = 1 \\ v = P_{\geq j} u}} \langle Av, v \rangle_{\mathcal{H}}$$

We notice that:

$$\left\{ v \in \mathcal{H} \mid \exists u \in \mathcal{H}, \|u\|_{\mathcal{H}} = 1, v = P_{\geq j} u \right\} \supseteq \left\{ v \in \mathrm{Im}(P_{\geq j}) \mid \|v\|_{\mathcal{H}} = 1 \right\}$$

Indeed, if we take $v \in \mathrm{Im}(P_{\geq j})$ with $\|v\|_{\mathcal{H}} = 1$, then $v = P_{\geq j} v$ and we directly have the inclusion. This gives:

$$\lambda_1(P_{\geq j} A P_{\geq j}) \geq \max_{\substack{v \in \mathrm{Im}(P_{\geq j}) \\ \|v\|_{\mathcal{H}} = 1}} \langle Av, v \rangle_{\mathcal{H}}$$

Now, let us observe that the orthogonal set to $\mathrm{Im}(P_{\geq j})$ is $\mathrm{Im}(P_{<j})$ which is of dimension $j - 1$.

$$\lambda_1(P_{\geq j} A P_{\geq j}) \geq \max_{\substack{v \perp \mathrm{Im}(P_{<j}) \\ \|v\|_{\mathcal{H}} = 1}} \langle Av, v \rangle_{\mathcal{H}}$$

$$\geq \min_{V_{j-1}} \max_{\substack{v \perp V_{j-1} \\ \|v\|_{\mathcal{H}} = 1}} \langle Av, v \rangle_{\mathcal{H}}$$

$$= \lambda_j(A)$$

which concludes the proof.

$\square$

# F. Rates of convergence of $\mathbb{E}\|\phi_i - \hat{\phi}_i\|_{\mathcal{H}}^2$

Recall that Lemma C.3 bounds the second term of the PCA risk by our bounds given in Theorems 3.3 and 3.8 that both depend on the quantity $\mathbb{E}\|\phi_i - \hat{\phi}_i\|_{\mathcal{H}}^2$, where $\hat{\phi}_i = \Phi(\hat{\mu}_i)$, and $\hat{\mu}_i$ is an estimator of $\mu_i$ based on $m_i$ samples. In this appendix, we wish to review existing rates of convergence of $\mathbb{E}\|\phi_i - \hat{\phi}_i\|_{\mathcal{H}}^2$, which will depend on the chosen embedding $\Phi$. We denote $B(x, r)$ the ball centered at $x$ of radius $r$. For any $a, b \in \mathbb{R}$, $a \wedge b = \min\{a, b\}$.

## F.1. Kernel mean embedding

**Theorem F.1.** *Let $k$ be a continuous positive definite kernel on a separable topological space $\mathcal{X}$ such that $\sup_{x \in \mathcal{X}} k(x, x) = K < \infty$. Let $Y_1, \cdots, Y_m$ be $m$ i.i.d. samples from a probability measure $\mu$ on $\mathcal{X}$. Define the empirical estimator of the kernel mean embedding as:*

$$\Phi^{\mathrm{KME}}(\hat{\mu}) = \frac{1}{m} \sum_{j=1}^{m} k(Y_j, \cdot).$$

*Then,*

$$\mathbb{E}[\|\Phi^{\mathrm{KME}}(\mu) - \Phi^{\mathrm{KME}}(\hat{\mu})\|_{\mathcal{H}_k}^2] \lesssim m^{-1}.$$

The following proof is inspired by the proof of Theorem 2.22 in (Chewi et al., 2024), which we rewrite here for completeness.

*Proof.* We have that

$$
\begin{aligned}
\mathbb{E}[\|\Phi^{\mathrm{KME}}(\mu) - \Phi^{\mathrm{KME}}(\hat{\mu})\|_{\mathcal{H}_k}^2] &= \mathbb{E}\left[\left\|\int_{\mathcal{X}} k(y, \cdot)\mathrm{d}\mu(y) - \frac{1}{m}\sum_{j=1}^{m} k(Y_j, \cdot)\right\|_{\mathcal{H}_k}^2\right] \\
&= \mathbb{E}\left[\left\|\frac{1}{m}\sum_{j=1}^{m} k(Y_j, \cdot) - \int_{\mathcal{X}} k(y, \cdot)\mathrm{d}\mu(y)\right\|_{\mathcal{H}_k}^2\right] \\
&= \frac{1}{m}\mathbb{E}\left[\left\|k(Y_1, \cdot) - \int_{\mathcal{X}} k(y, \cdot)\mathrm{d}\mu(y)\right\|_{\mathcal{H}_k}^2\right] \\
&= \frac{1}{m}\left(\mathbb{E}\left\|k(Y_1, \cdot)\right\|_{\mathcal{H}_k}^2 - \left\|\int_{\mathcal{X}} k(y, \cdot)\mathrm{d}\mu(y)\right\|_{\mathcal{H}_k}^2\right) \\
&\leq \frac{1}{m}\mathbb{E}\left\|k(Y_1, \cdot)\right\|_{\mathcal{H}_k}^2 \\
&= \frac{1}{m}\mathbb{E}k(Y_1, Y_1) \\
&\leq \frac{K}{m}
\end{aligned}
$$

$\square$

## F.2. LOT

In this section, we consider two absolutely continuous probability measures $\rho$ and $\mu$ supported on $\mathcal{X} \subset \mathbb{R}^d$, with respective densities $f$ and $g$. According to Brenier's theorem (Brenier, 1991), there exists an OT map $T$ between $\rho$ and $\mu$ which is the gradient of a convex function $\varphi$. In practice, we do not directly observe $\rho$ and $\mu$ but samples of $\rho$ and $\mu$. From these samples, one can derive estimators $\hat{T}$ of $T$ and study the rate of convergence in terms of the following loss function

$$\mathbb{E}\|\phi_i - \hat{\phi}_i\|_{\mathcal{H}}^2 = \mathbb{E}\|\hat{T} - T\|_{L^2(\rho)}^2.$$

We list a few assumptions which will be mentionned in some of those rates.

**Assumption F.2.** The set $\mathcal{X}$ is compact and convex with nonempty interior such that $\mathcal{X} \subseteq [0, 1]^d$.

**Assumption F.3.** There exists $\varepsilon_0, \delta_0 > 0$ such that for all $x \in \mathcal{X}$ and $\varepsilon \in (0, \varepsilon_0)$, we have $\mathcal{L}(B(x, \varepsilon) \cap \mathcal{X}) \geq \delta_0 \mathcal{L}(B(x, \varepsilon))$, with $\mathcal{L}$ the Lebesgue measure on $\mathbb{R}^d$.

**Assumption F.4.** The Brenier potential $\varphi$ is in $C^2(\mathcal{X})$ and for some $\lambda > 0$, $(1/\lambda)I_d \preceq \nabla^2\varphi(x) \preceq \lambda I_d$ for all $x \in \mathcal{X}$.

### F.2.1. ONE-SAMPLE PROBLEM

In the one-sample problem, we suppose that $\rho$ is a known distribution and $\mu$ is an unknown distribution from which an i.i.d. sample $Y_1, \cdots, Y_m$ is observed. As $\rho$ is absolutely continuous, there exists an OT map between $\rho$ and any sample-based

estimator of $\mu$. The first natural estimator is the empirical measure

$$\hat{\mu} = \frac{1}{m} \sum_{j=1}^{m} \delta_{Y_j}.$$

Let $\hat{T}$ denote the OT map between $\rho$ and $\hat{\mu}$. In this semi-discrete setup, (Manole et al., 2024) show the following bound on the risk of $\hat{T}$:

**Theorem F.5** (Corollary 7 in (Manole et al., 2024))**.** *If Assumptions F.2 and F.4 hold, then*

$$\mathbb{E}\|\hat{T} - T\|_{L^2(\rho)}^2 \lesssim \begin{cases} m^{-1/2}, & d \leq 3, \\ m^{-1/2}\log m, & d = 4, \\ m^{-2/d}, & d \geq 5. \end{cases}$$

The authors of (Manole et al., 2024) also focus on the case where $\mu$ admits a smooth density $g$. We define for any $M, \gamma > 0$:

$$C^s(\mathcal{X}; M, \gamma) = \{f \in C^s(\mathcal{X}) \ : \ \|f\|_{C^s(\mathcal{X})} \leq M, f \geq 1/\gamma \text{ over } \mathcal{X}\},$$

where $C^s(\mathcal{X})$ is the Hölder space. Let $\hat{\mu}$ be the measure of the wavelet density estimator $\hat{g}$. Let $\hat{T}$ be the OT map between $\rho$ and $\hat{\mu}$.

**Theorem F.6** (Theorem 10 in (Manole et al., 2024))**.** *Let $s > 1$ and $M, \gamma > 0$. Let $\rho$ and $\mu$ be two absolutely continuous probability measures on $\mathcal{X} = [0,1]^d$ and assume the density $g$ of $\mu$ is in $C^{s-1}([0,1]^d; M, \gamma)$. Let $J_m$ be the truncation level of the truncated wavelet estimator of $g$ and assume $2^{J_m} \asymp m^{1/(d+2(s-1))}$. If Assumption F.4 holds, then*

$$\mathbb{E}\|\hat{T} - T\|_{L^2(\rho)}^2 \lesssim \begin{cases} 1/m, & d = 1, \\ (\log m)^2/m, & d = 2, \\ m^{-\frac{2s}{2(s-1)+d}}, & d \geq 3. \end{cases}$$

F.2.2. TWO-SAMPLE PROBLEM

We now suppose both $\rho$ and $\mu$ are unknown absolutely continuous probability measures. Let $X_1, \cdots, X_{m_0} \sim \rho$ and $Y_1, \cdots, Y_m \sim \mu$ be i.i.d. samples.

**Barycentric projection.** Define the empirical measures

$$\hat{\rho} = \frac{1}{m_0} \sum_{j=1}^{m_0} \delta_{X_j}, \qquad \hat{\mu} = \frac{1}{m} \sum_{j=1}^{m} \delta_{Y_j}.$$

A Monge map between $\hat{\rho}$ and $\hat{\mu}$ might not exist when $m \neq m_0$ but an OT plan $\pi$ always does. From $\pi$, a transport map $\hat{T}_\pi$ is constructed with the barycentric projection (10).

**Theorem F.7** (Corollary 2.3 in (Deb et al., 2021))**.** *If $\mu$ and $\nu$ are compactly supported, $T$ is $L$-Lipschitz and $\mathbb{E}_{X_1 \sim \hat{\rho}}[\exp(t\|X_1\|^\alpha)] < \infty$ for some $t > 0, \alpha > 0$, then,*

$$\mathbb{E}\|\hat{T}_\pi - T\|_{L^2(\hat{\rho})}^2 \lesssim \kappa_{m,m_0} \text{ with } \quad \kappa_{m,m_0} = \begin{cases} m^{-1/2} + m_0^{-1/2}, & d = 2,3, \\ m^{1/2}\log(1+m) + m_0^{-1/2}\log(1+m_0), & d = 4, \\ m^{-2/d} + m_0^{-2/d}, & d \geq 5. \end{cases}$$

**One-nearest neighbor estimator.** The barycentric projection map is only defined on the support of $\hat{\rho}$ that is on $X_1, \cdots, X_{m_0}$. The one-nearest neighbor extrapolation allows to extend this map to out-of-sample points. Let $V_j$ be the Voronoi cell centered at $X_j$, defined as:

$$V_j = \{x \in \mathcal{X} \ : \ \|x - X_j\| \leq \|x - X_k\|, \ \forall k \neq j\}.$$

The one-nearest neighbor estimator of $T$ can then be defined by:

$$\hat{T}(x) = \sum_{j=1}^{m_0} I(x \in V_j) T_\pi(x_j),$$

where $I$ is the indicator function. (Manole et al., 2024) show the following bound on the risk of $\hat{T}$.

**Theorem F.8** (Proposition 15 in (Manole et al., 2024)). *If the density $f$ of $\rho$ is bounded and Assumptions F.2, F.3 and F.4 hold, then*

$$\mathbb{E}\|\hat{T} - T\|^2_{L^2(\rho)} \lesssim (\log m_0)^2 \kappa_{m \wedge m_0} \qquad \kappa_n = \begin{cases} n^{-1/2}, & d \leq 3, \\ n^{-1/2} \log n, & d = 4, \\ n^{-2/d}, & d \geq 5. \end{cases}$$

**Smooth OT map estimation.** Let $\hat{\rho}$ be an estimator of $\rho$ based on $m_0$ samples, such that $\hat{\rho}$ is absolutely continuous. Let $\hat{\mu}$ be any estimator of $\mu$ based on $m$ samples. As $\hat{\rho}$ is absolutely continuous, there exists a unique transport map $\hat{T}$ between $\hat{\rho}$ and $\hat{\mu}$.

**Theorem F.9** (Theorem 4 in (Balakrishnan & Manole, 2025)). *Suppose we have Assumptions F.2 and F.4. Let $s, \gamma, M > 0$. We assume that $f$ and $g$ are in $C^s(\mathcal{X}; M, \gamma)$ and vanish at the boundary of $\mathcal{X}$ up to order $\lfloor s \rfloor$. Then,*

$$\mathbb{E}\|\hat{T} - T\|^2_{L^2(\rho)} \lesssim \kappa_{m_0 \wedge m} \qquad \kappa_n = \begin{cases} 1/n, & d = 1, \\ \log n/n, & d = 2, \\ n^{-\frac{2(s+1)}{2s+d}}, & d \geq 3. \end{cases}$$

This convergence rates matches known minimax lower bounds for estimating OT maps between Hölder continuous densities up to a log factor when $d = 2$ (Hütter & Rigollet, 2021).

**Entropic OT.** OT suffers from high computational costs: in the discrete case, solving the OT problem between points clouds of $n$ points has complexity $O(n^3 \log n)$. In this context, the seminal paper by (Cuturi, 2013) introduces entropic regularization, which enable the fast computation of optimal transport distances using the Sinkhorn algorithm. Entropic OT consists in modifying problem 8 by adding a penalization term based on the entropy of the coupling:

$$\min_{\pi \in \Pi(\rho, \mu)} \int_{\mathcal{X} \times \mathcal{X}} \|x - y\|^2 \mathrm{d}\pi(x, y) + \varepsilon \mathrm{KL}(\pi | \rho \otimes \mu), \tag{34}$$

where $\varepsilon > 0$, $\rho \otimes \mu$ is the product measure and $\mathrm{KL}(\alpha|\beta) = \int \log \frac{\mathrm{d}\alpha}{\mathrm{d}\beta} \mathrm{d}\alpha$ when $\alpha \in \mathcal{M}(\mathcal{X})$ is absolutely continuous with respect to $\beta \in \mathcal{M}(\mathcal{X})$.

In (Pooladian et al., 2023), the authors compute an entropic OT map $T_\varepsilon$ via computing the barycentric projection of the optimal entropic coupling. Denoting $\hat{\rho}$ and $\hat{\mu}$ the empirical measures associated with two $m$-samples from $\rho$ and $\mu$, we note $\hat{T}_\varepsilon$ the transport map obtained by computing the barycentric projection of the optimal entropic coupling between $\hat{\rho}$ and $\hat{\mu}$. They show the following:

**Theorem F.10.** *Suppose $\rho$ has a compact convex support $\Omega \subset B(0, R)$ with a bounded density $f$. If $\mu$ is a discrete measure with supported included in $B(0, R)$, $\varepsilon \asymp m^{-1/2}$ and $m$ is large enough, then*

$$\mathbb{E}\|\hat{T}_\varepsilon - T_0\|^2_{L^2(\rho)} \lesssim m^{-1/2}.$$

### F.3. Sliced Wasserstein embedding

**Theorem F.11.** *Let $\mu$ be a probability measure supported on $\mathcal{X} \subseteq \mathbb{R}^d$ with finite-second moment. Let $\hat{\mu}$ be the empirical measure based on $m$ i.i.d. samples $Y_1, \cdots, Y_m$ of $\mu$. Then,*

$$\mathbb{E}[\|\Phi^{\mathrm{SW}}(\mu) - \Phi^{\mathrm{SW}}(\hat{\mu})\|^2_{L^2([0,1] \times \mathbb{S}^{d-1})}] \lesssim m^{-1}.$$

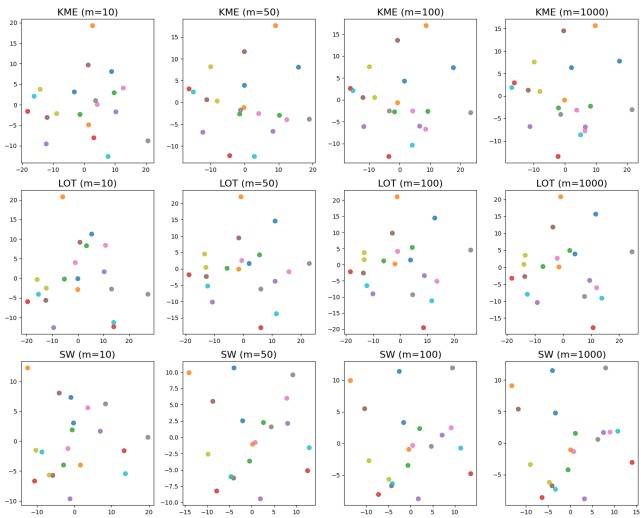

*Figure 7.* 2D PCA representation of $n = 20$ gaussian measures for different subsamples sizes $m$ and three embeddings (KME, LOT, SW). Each plot shows the projection onto the first two principal components.

*Proof.* By definition of the sliced Wasserstein distance, we have:

$$\mathbb{E}\mathrm{SW}_2^2(\mu, \hat{\mu}) = \int_{\mathbb{S}^{d-1}} \mathbb{E}W_2^2(P_\#^\theta \mu, P_\#^\theta \hat{\mu})\mathrm{d}\sigma(\theta).$$

Notice that $P^\theta(Y_1), \cdots, P^\theta(Y_m)$ are i.i.d samples from $P_\#^\theta \mu$ and $P_\#^\theta \hat{\mu}$ is the empirical measure based on these samples. As, the projected law $P_\#^\theta \mu$ has finite second moment, the one-dimensional rate $\mathbb{E}W_2^2(P_\#^\theta \mu, P_\#^\theta \hat{\mu}) \lesssim m^{-1}$ holds (see e.g. (Fournier & Guillin, 2015) for general bounds on the rate of convergence of empirical measures in Wasserstein distance in low dimension).

$$\mathbb{E}\mathrm{SW}_2^2(\mu, \hat{\mu}) \lesssim \int_{\mathbb{S}^{d-1}} m^{-1}\mathrm{d}\sigma(\theta) \lesssim m^{-1}.$$

However, the distance between the embedded measures through the sliced-Wasserstein embedding exactly corresponds to the sliced-Wasserstein distance between these measures. Hence

$$\mathbb{E}[\|\Phi^{\mathrm{SW}}(\mu) - \Phi^{\mathrm{SW}}(\hat{\mu})\|_{L^2([0,1]\times\mathbb{S}^{d-1})}^2] = \mathbb{E}[\mathrm{SW}_2^2(\mu, \hat{\mu})] \lesssim m^{-1}.$$

$\square$

# G. Additional experiments

## G.1. Numerical experiments on simulated data

On our simulated dataset from Section 4.1, we demonstrate that relatively few samples per measure can yield accurate and stable PCA representations. Fixing $n = 20$ and visualizing the resulting PCA for different values of $m \in \{10, 50, 100, 1000\}$ in Figure 7, we observe that the representation stabilizes around $m = 50$, which is nearly identical to the $m = 1000$ case. This confirms that moderate values of $m$ are sufficient to obtain high-quality PCA.

## G.2. Image dataset

We consider a dataset of RGB images from Sentinel-2 satellite imagery. Each image is viewed as a discrete probability measure on the RGB space, that is each pixel is represented by a point $\mathbb{R}^3$. Pixel intensities are normalized to lie in $[0, 1]^3$. We extract $n = 30$ images of size $64 \times 64$ pixels from different land cover types. We subsample $m$ pixels from each image to compute the embeddings. We consider the uniform measure on $[0, 1]^3$ as the reference measure, and we sample $m_0 = 100$ points from it. For KME, we use the RBF kernel with bandwidth $\sigma = 0.5$, evaluated on the $m_0 = 100$ points. For the LOT embedding, we compute the barycentric projection from the empirical reference measure $\hat{\rho}_{m_0}$ to each data as described in

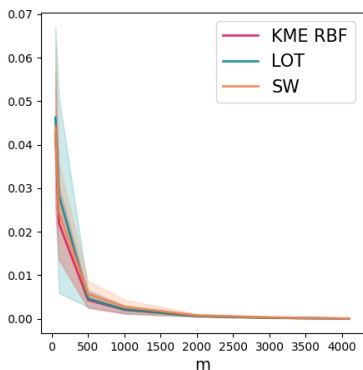

*Figure 8.* Mean Procrustes disparity and standard deviation for different subsample sizes on the image dataset.

Section 4.1. Finally, the number of quantiles and the number of projections for the SW embedding are respectively $T = 10$ and $p = 10$. We visualize the 2-dimensional PCA representation for each embedding and different subsample sizes in Figure 9. We also compute the mean Procrustes disparity between PCA representations obtained from different random subsamples of the same size, as described in Section 4, and plot the results in Figure 8.

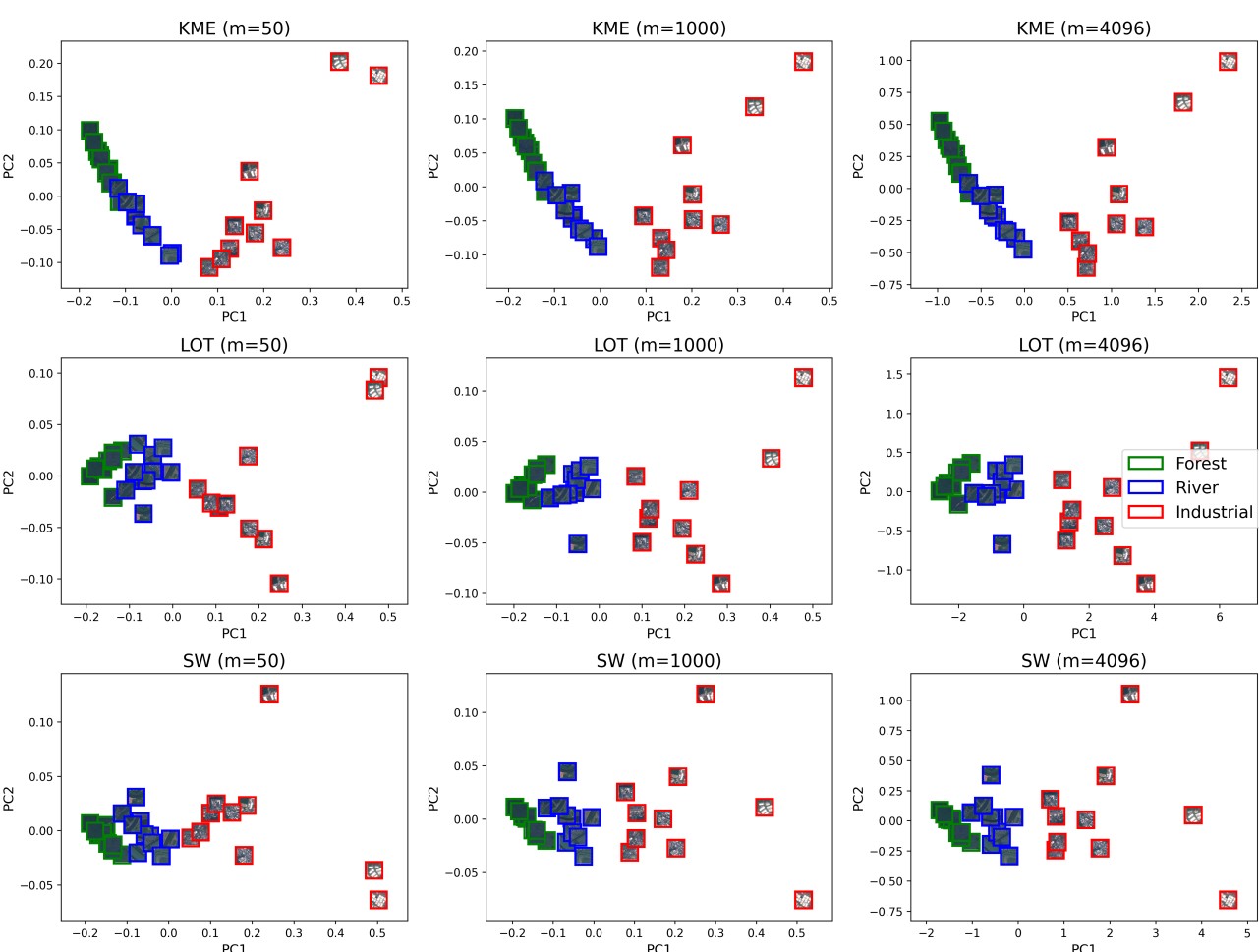

*Figure 9.* 2D PCA representation of the image dataset for different subsamples sizes $m$ and three embeddings (KME, LOT, SW). Each plot shows the projection onto the first two principal components.

