# OpenReview forum: "PCA of Probability Measures: Sparse and Dense Sampling Regimes"
_ICML.cc/2026/Conference — ICML 2026 regular_

### Official Review · Reviewer_PXpC · 2026-03-10

**Soundness:** 3
**Presentation:** 3
**Significance:** 3
**Originality:** 3
**Overall Recommendation:** 4
**Confidence:** 5

**Summary:**

This paper studies PCA of probability measures, establishing convergence rate in terms of number of probability measures (n) and number of samples from each measure (m). The theory suggests a transition from sparse region (small m) to a dense regime (large m). In the dense case, the empirical covariance error has a minimal optimal rate.

**Compliance With Llm Reviewing Policy:**

Affirmed.

**Final Justification:**

Authors addressed my concerns so I raised the score to 4.

**Key Questions For Authors:**

This seems similar to "weakness", please see details there.

**Limitations:**

I didn't find any discussion on limitations, which could have been discussed in Section 5.

**Strengths And Weaknesses:**

**Strength**

The paper is technically sound.

The main theorem is a general result in the sense that it doesn't depend on specific embedding method.

**Weakness**

It's not clear to me why people care about PCA for probability measures. The only motivation is in the second sentence of the first paragraph of Section 1. However, it's still not clear why we need to do PCA, what's the outputs of PCA, and how can we make use of the PCA outputs.

Throughout the paper, the sample size $m$ for each measure is assumed to be constant, without any justification. The authors should either explain why this assumption is valid or relax this assumption, or at least explain why this assumption is (not) essential.

The theoretical results seem incomplete to me. Think about PCA, the main goals are to 1) obtain the projection (matrix), which helps to interpret each loading/factor and the importance of each feature and 2) obtain the low-dimensional representations. However, in the current manuscript, the only theoretical results are regarding the error of using empirical covariance to approximate population covariance (Theorem 3.3, 3.4) and the reconstruction error (Theorem 3.7 and Corollary 3.8). I understand that these two tasks in PCA are trivial but not for the framework in this manuscript. However, more interesting results are about the error of the estimated projection $P_{\leq q}$.

There is a (big) gap between the simulation study and the claims from the author(s). My understanding is that Figure 1 is to verify the rate in terms of $n$ in the dense case with a fixed $m$, which is 1/2 according to the theory. The authors claim that "Figure 1 displays
the results, confirming the expected $n^{−1/2} decay for both metrics across all three embeddings". However, the empirical rate is not clear at all. An alternative is to make log-log plot and fit a linear regression and report the slope, which is expected to be close to -1/2. Such analysis is missing for both Figure 1 and Figure 2.

There is again a big gap between real data analysis in Section 4.2 and the theory. On one hand, the entire theoretical framework is about asymptotics. On the other hand, $n$ in both examples in Section 4.2 are super small ($n=63$ for flow cytometry and $n=20$ for the 3-D shapes). Is there any example where both $n$ and $m$ are big so that asymptotics is meaningful?

Also in the real data analysis, the ultimate goal of the analysis is not clear. Is the point here to show the 2-d representation helps with clustering? If the main goal is to show how many subsamples are "good enough", there is an entire literature in subsampling (also called design), but there is no comparison to those results. In general I feel the real data analysis is superficial.

---

> ### Author Rebuttal · Authors · 2026-03-30
>
> We thank the reviewer for the detailed feedback.
>
> > **"It's not clear to me why people care about PCA of probability measures."**
>
> We agree that the motivation for PCA of probability measures should be clarified. PCA in the context of probability measures has been widely studied [1,2]. The goal is to identify the main modes of variability, producing outputs analogous to classical PCA: principal directions in the embedded space, component scores, and modes of variation that can be mapped back to the space of distributions. These outputs enable low-dimensional representation, interpretation of variability, and feature extraction for downstream tasks such as clustering or classification.
>
> We refer to our response to Reviewer 4que for more details.
>
> > **About the sample size $m$**
>
> For simplicity, we assume a constant number of samples $m$ per measure, but our results extend to varying $m_i$. In that case, the within-measure approximation error term $r_m(\Phi)$ becomes
>
> $$
> \frac{1}{n}\sum\limits_{i=1}^n r_{m_i}(\Phi) \leq r_{m_{\min}}(\Phi), \qquad m_{\min} := \min\limits_{1\leq i\leq n} m_i,
> $$
>
> since $r_m(\Phi)$ is nonincreasing. We will add a remark to clarifiy this in the revision.
>
> > **"The theoretical results seem incomplete to me. [...] more interesting results are about the error of the estimated projection"**
>
> We thank the reviewer for this comment. We believe that bounding the covariance operator is central, as principal projections and low-dimensional representations rely on its eigenstructure, and the reconstruction error is a standard PCA criterion [3].
>
> We agree that an explicit control of the projection error would further strengthen the contributions. Using Lemma 2.6 in [3] and a standard identity,
>
> $$
> \\mathcal{E}^{\\mathrm{PCA}}\_q \\geq (\\lambda\_{q+1} - \\lambda\_q) \\sum\\limits\_{k>q}  \\Vert P\_k \\hat{P}\_{\\leq q} \\Vert^2_{\\mathrm{HS}} = (\\lambda\_{q+1} - \\lambda\_q) \\frac{\Vert P_{\leq q} - \\hat{P}\_{\leq q}\Vert^2_{\mathrm{HS}} }{2},
> $$
>
> we obtain
>
> $$
> \Vert P_{\leq q} - \hat{P}\_{\leq q}\Vert^2_{\mathrm{HS}}  \leq \frac{2\mathcal{E}^{\mathrm{PCA}}_q}{\\lambda\_{q+1} - \\lambda\_q},
> $$
>
> This variant of the Davis-Kahan sin-$\theta$ theorem shows that the projector error inherits the bounds from Theorem 3.7. We will include this result as a corollary in the revision.
>
> > **About the rates in Figures 1 and 2**
>
> We thank the reviewer for this suggestion. To address this, we performed an additional analysis based on log-log plots and linear regression to estimate the empirical convergence rates, see table below.
>
> |                   |    |  KME    | LOT    | SW     |
> |-------------------|------|--------|--------|--------|
> | **Covariance estimation** | fixed *m*, varying *n* | -0.49  | -0.48  | -0.49  |
> |                   | fixed *n*, varying *m* | -0.37  | -0.53  | -0.36  |
> | **Excess risk**    | fixed *m*, varying *n* | -1.02  | -1.06  | -1.09  |
> |                   | fixed *n*, varying *m* | -0.68  | -0.76  | -0.70  |
>
> We observe that the excess-risk curves exhibit slopes close to -1 as a function of $n$, which is consistent with our theory since our bounds allow either an $n^{-1/2}$ or an $n^{-1}$ decay. Also, when varying $m$, the observed slopes are close to $-1/2$, although not as close as the theory would suggest. A more extensive exploration covering a wider range of $m$ and $n$ would likely better isolate this regime, but we were limited here due to time constraints.
>
> > **About the real data analysis**
>
> We would like to clarify that our bounds in Theorems 3.3 and 3.7 are **non-asymptotic**, and hold for any finite values of $n$ and $m$, explicitely quantifying the dependence on both parameters. The notions of "sparse" and "dense" regimes arise from the relative dominance of the terms $n^{-1/2}$ and $r_m(\Phi)$ in these finite-sample bounds, rather than from an asymptotic limit.
>
> The 2D representations in our real-data experiments assess the stability of PCA under subsampling. We focus on $n$ probablity measures, each observed through $m$ samples, a setting that differs from classical vector-based design and subsampling frameworks. We are not aware of references that study the interplay between the number of measures $n$ and the number of samples $m$ in PCA for probability measures. We would be grateful if the reviewer could point us to specific references to help better position our contribution.
>
> ### References
>
> [1] Seguy, V. and Cuturi, M. Principal geodesic analysis for probability measures under the optimal transport metric. Advances in Neural Information Processing Systems.
>
> [2] Vesseron, N., Cazelles, E., Brigant, A.L. and Klein, T. On the Wasserstein geodesic principal component analysis of probability measures.
>
> [3] Reiss, M. and Wahl, M. Nonasymptotic upper bounds for the reconstruction error of PCA. The Annals of Statistics.

---

> > ### Author Rebuttal · Reviewer_PXpC · 2026-04-01
> >
> > I'd like to thank the authors for addressing my concerns carefully. I am more than happy to raise my score. I strongly recommend the authors to include the additional results in the next version which make the paper much stronger, including:
> > 1. more motivation
> > 2. error bound for epsilon
> > 3. empirical supports to the rates.

---

### Official Review · Reviewer_EY5E · 2026-03-11

**Soundness:** 3
**Presentation:** 3
**Significance:** 3
**Originality:** 3
**Overall Recommendation:** 3
**Confidence:** 4

**Summary:**

This paper studies PCA of random probability measures under a two-stage sampling model: one first samples $n$ measures, and then approximates each measure using $m$ samples. The authors analyze an embedding-based PCA framework in Hilbert spaces and derive rates for covariance estimation and PCA excess risk that scale as $n^{-1/2}+m^{-\alpha}$, where $\alpha$ depends on the embedding. The main message is a sparse-versus-dense sampling transition: when \(m\) is large enough, performance is governed by the standard $n^{-1/2}$ rate, while for small $m$, the quality of estimating each individual measure becomes the bottleneck. The paper also establishes minimax optimality in the dense regime and provides empirical results consistent with the theory.

**Compliance With Llm Reviewing Policy:**

Affirmed.

**Final Justification:**

This paper studies a meaningful problem and offers a valuable double-asymptotic analysis with an interpretable sparse-to-dense transition. In the submitted version, there was a mismatch between the LOT theory and the LOT experiments, and some results relied on restrictive boundedness assumptions. The rebuttal was helpful and addressed my main concern, particularly by acknowledging the LOT inconsistency and clarifying the interpretation of the LOT experiments. In light of this clarification, I increased my soundness score from 2 to 3. However, it also confirmed that this inconsistency was real in the submitted version. As a result, my overall evaluation did not change, and I keep my original recommendation of weak reject.

**Key Questions For Authors:**

* In the experiments, the LOT embedding is implemented using an empirical reference measure and, when$m_0 \neq m$, a barycentric projection derived from the optimal transport plan. However, the discussion of the rate alignment for $d=2$ states that all three embeddings have matching sampling rates because $m^{-1/d} = m^{-1/2}$ when $d=2$. This seems inconsistent with Table 1, where the LOT rate for the barycentric-projection estimator with empirical reference measure is listed as $m^{-1/4}$, not $m^{-1/d}$. Could the authors clarify exactly which LOT estimator is covered by the theory and which convergence rate should be expected in the experiments?

* The excess-risk analysis relies on a uniformly bounded embedding, and for LOT/SW this is justified by assuming the underlying space $\mathcal{X}$ is bounded. How essential is this assumption? Do the authors expect the main PCA excess-risk result to extend to unbounded settings under weaker moment conditions?

**Limitations:**

Yes.

**Strengths And Weaknesses:**

Strengths:

* The paper studies a meaningful problem and provides a clean double-asymptotic analysis that separates the roles of the number of measures $n$ and the number of samples per measure $m$. This yields an interpretable sparse-to-dense transition and a unified framework for comparing several important distribution embeddings.

* The paper gives a unified statistical comparison across several important embeddings—kernel mean embeddings, LOT, and sliced Wasserstein—within the same double-sampling framework. That makes the sparse/dense transition more broadly useful, rather than being tied to a single representation choice.

Weaknesses:

* There is a mismatch between the LOT theory and the LOT experiments. The experimental section appears to implement LOT with an empirical reference measure and barycentric projection, while Table 1 indicates that this estimator has a different convergence rate from the $m^{-1/d}$ rate used to justify the $d=2$ alignment and interpret the empirical trends. See questions below.

* The theoretical guarantees rely on fairly restrictive boundedness assumptions. In particular, the excess-risk analysis assumes a uniformly bounded embedding, which for LOT and SW is justified by assuming the underlying support is bounded.

---

> ### Author Rebuttal · Authors · 2026-03-30
>
> Thank you for your time and feedback. Below we address each concern in detail:
>
> > **"Could the authors clarify exactly which LOT estimator is covered by the theory and which convergence rate should be expected in the experiments? "**
>
> The reviewer is correct that there is a mismatch between the rate used in the discussion and the estimator implemented in our experiments. For simplicity, these rates are stated in Table 1 in a general form that are valid when $d\geq 5$. Low-dimensional settings, particularly when $d=2$, can exhibit different statistical behavior. In our experiments, we are in a fully discrete regime, where the transport map is obtained via the barycentric projection of the optimal transport plan. As shown in [1], this estimator leads to rates on the order of $m^{-1/4}$, when the true OT map is Lipschitz and the support of the target measure compact. Therefore, the rate alignment argument in dimension $d=2$ does not apply to the estimator used in our experiments. We agree that this is an inconsistency in the current presentation. We will clarify this point in the revised version.
>
> > **"Do the authors expect the main PCA excess-risk result to extend to unbounded settings under weaker moment conditions?"**
>
> We thank the reviewer for raising this important point. We agree that the boundedness assumption (Assumption 3.6) can be relaxed. In fact, the excess risk upper bound can be proven under weaker moment conditions. Instead of assuming that the embedding is uniformely bounded, it is sufficient to assume the fourth-moment condition of Assumption 3.1, that is,
>
> $$
> \mathbb{E}\Vert\phi\Vert^4_{\mathcal{H}}\leq R,
> $$
>
> together with a subgaussianity assumption on $\phi = \Phi(\mu)$. This can be obtained by modifying the proof of Theorem 3.7 in Appendix C, which is based on Lemmas C.1, C.2 and C.3. Lemma C.1 only divides the risk into two terms, and does not rely on boundedness.  Lemma C.2 requires a concentration inequality for covariance operators, which can be obtained with subgaussianity and pregaussianity. We therefore assume $\phi$ is subgaussian, and the fourth-moment assumption ensures the pregaussian property. Lemma C.3 can be in fact simplified and does not need boundedness. Using Cauchy-Schwarz, we have that:
>
> $$
> \langle \Sigma^n - \hat{\Sigma}, P\rangle_{\mathrm{HS}}
> \leq \Vert \Sigma^n - \hat{\Sigma} \Vert_{\mathrm{HS}} \Vert P \Vert_{\mathrm{HS}},
> $$
>
> Using that $\Vert P\Vert_{\mathrm{HS}} = \sqrt{q}$ and that $\mathbb{E}\Vert\Sigma^n - \hat{\Sigma}\Vert_{\mathrm{HS}} \leq 2R^{1/4} r_m(\Phi)$ from Lemma A.2, we can conlude that
>
> $$
> \\mathbb{E} \\bigl[{\\left\\langle \\Sigma^n - \\hat{\\Sigma}, P\_{\\leq q} - \\hat{P}\_{\\leq q} \\right\\rangle}_{\\mathrm{HS}} \\bigr] \leq \mathbb{E}\bigl[2 \sup\limits\_{P \in P\_q}  |\langle \Sigma^n - \hat{\Sigma}, P\rangle\_{\mathrm{HS}} |\bigr] \leq 4R^{1/4}\sqrt{q}r\_m(\Phi).
> $$
>
> This yields the same contribution as in Theorem 3.7. We will clarify this point in the paper and explicitely state Theorem 3.7 under these weaker assumptions.
>
> ### References
>
> [1] Deb, N., Ghosal, P. and Sen, B., 2021. Rates of estimation of optimal transport maps using plug-in estimators via barycentric projections. Advances in Neural Information Processing Systems, 34, pp.29736-29753.

---

> > ### Author Rebuttal · Reviewer_EY5E · 2026-04-02
> >
> > Thank you for your rebuttal. I had a follow up question: Since the $d=2$ rate-alignment argument does not apply to the LOT estimator used in the experiments, could the authors clarify what claim the LOT experiments are intended to support? In particular, should these results be interpreted as validating a corresponding theoretical rate, or only as empirical behavior of the implemented discrete barycentric-projection estimator?

---

> > > ### Author Response · Authors · 2026-04-02
> > >
> > > We once again thank the reviewer for their careful reading of our paper and for pointing out this important point.
> > >
> > > The goal of this experiment is to illustrate the empirical behavior of the practical LOT estimator used in our implementation. In particular, our implementation relies on a barycentric projection of a discrete optimal transport plan, for which existing results suggest a convergence rate of order $m^{-1/4}$.
> > >
> > > Interestingly, in our experiments, we observe a decay closer to $m^{-1/2}$, matching the behavior of the other embeddings. This suggests that the $m^{-1/4}$ rate may be pessimistic. Indeed, to the best of our knowledge, this rate is not minimax optimal. Closing the gap between the rates achieved in practice and the optimal statistical rates is actually a very active area of research in optimal transport [1, 2, 3].
> > >
> > > We will add this discussion in the revision.
> > >
> > > [1] J.-C. Hütter and P. Rigollet. Minimax rates of estimation for smooth optimal transport maps.
> > >
> > > [2] A.-A. Pooladian, V. Divol, and J. Niles-Weed. Minimax estimation of discontinuous optimal transport maps: The semi-discrete case. In International Conference on Machine Learning,
> > >
> > > [3] Manole, T., Balakrishnan, S., Niles-Weed, J. and Wasserman, L., 2024. Plugin estimation of smooth optimal transport maps. The Annals of Statistics.
> > >
> > > ### **EDIT**
> > >
> > > Thank you again for you time and feedback. We have addressed your latest question and would greatly appreciate if you could let us know whether our clarification resolved your concern. If there are still aspects that you feel should be improved, please let us know.

---

### Official Review · Reviewer_4que · 2026-03-12

**Soundness:** 3
**Presentation:** 3
**Significance:** 3
**Originality:** 2
**Overall Recommendation:** 4
**Confidence:** 2

**Summary:**

In this paper, the authors consider the theoretical error of PCA, where the goal is to approximate the population covariance matrix by the empirical covariance matrix. For the true random measure $\mu$, $n$ independent copies of $\mu$ are sampled first, and then for each copy $m$ independent samples are drawn. The empirical covariance based on an embedding into a Hilbert space. Under mild assumptions on the measure and the embedding, the authors proved the bounds of the approximation by the empirical covariance in terms of the Hilbert-Schmidt norm difference and the excess risk in the double asymptotic regime as both $n$ and $m$ tend to infinity with possibly different speed. Numerical experiments are also provided.

**Compliance With Llm Reviewing Policy:**

Affirmed.

**Final Justification:**

The authors answered my question adequately. I will maintain my score.

**Key Questions For Authors:**

- In 1.1.(iii), I would like to know more about the validity of the assumption that the eigenvalues of $\Sigma$ exhibit a polynomial decay (with the decay faster than $j^{-\alpha}$ for some $\alpha > 3/2$.)
- It would be good if the reason why the Hilbert-Schmidt norm difference is considered in this work, instead of other norms, e.g., the operator norm. Do you expect that the phase transition in 1.1.(i) and the optimality in 1.1.(ii) would fail if other norms are used?

**Limitations:**

Yes

**Strengths And Weaknesses:**

Strengths
- The results establish the phase transition from the sparse regime to dense regime, and the error estimate is near-optimal in the dense regime.
- The theoretical analysis is thoroughly done.
- The manuscript is well-written and the results are well-presented.

Weaknesses
- While the results are interesting, it is not well-discussed how and why the results may be useful in machine learning.
- The novelty of the work should be explained further.

---

> ### Author Rebuttal · Authors · 2026-03-30
>
> Thank you for your positive feedback and insightful comments.
>
> > **"While the results are interesting, it is not well-discussed how and why the results may be useful in machine learning."**
>
> The motivation behind our work is twofold:
>
> 1. There is a growing of interest in distributional data analysis, where observations are probability measures rather than vectors. This setting naturally arises in many machine learning applications, including demographics, economics, imaging, text, spatio-temporal processes, and biomedical applications.
> 2. However, standard functional PCA  (FPCA) fails on probability measures: due to nonlinear geometry, FPCA components do not remain within the space of probability measures, rendering their interpretation complex [1].
>
> Our contribution is to study this problem in a statistical learning setting where one observes $n$ probability measures, each through $m$ samples. This setting is highly relevant in practice: in applications such as flow cytometry, each data point is a large point cloud, and computing Hilbert space embeddings can be computationally expensive. Our results show that beyond a certain threshold on $m$, increasing the number of samples per distributions does not improve accuracy, allowing computational savings via subsampling without degrading PCA performance.
>
> > **"The novelty of the work should be discussed further."**
>
> The behaviour of PCA in a double asymptotic regime has been previously studied in the context of  FPCA, where one observes $n$ functions through $m$ discretization points. Extending such results to probability measures is however non-trivial, as the data are no longer functions but distributions that must first be embedded into a Hilbert space. To the best of our knowledge, the statistical analysis of PCA for probability measures in this double sampling regime has not been addressed before.
>
> Our main contribution is to establish convergence rates for key PCA quantities of the form $n^{-1/2} + m^{-\alpha}$ which explicitly quantify the interplay between inter-measure $(n)$ and intra-measure $(m)$ variability. This reveals a transition phenomenon between sparse and dense sampling regimes, analogous to functional PCA, but previously unexplored for distributional data.
>
> Our results provide guidance on how to choose the number of samples $m$ as a function of $n$, and we demonstrate experimentally that significant subsampling can preserve PCA accuracy, which is highly relevant in large-scale applications.
>
> > **"I would like to know more about the validity of the assumption that the eigenvalues of $\Sigma$ exhibit a polynomial decay."**
>
> We thank the reviewer for raising this point. Polynomial or exponential eigenvalue decay is standard in FPCA and is closely tied to regularity properties of the underlying random functions. In this setting, the covariance operator can be viewed as an integral operator with kernel $K(s,t)$, whose smoothness governs the decay of its eigenvalues: smoother kernels lead to faster decay. This connection is well documented in the literature on FPCA and Gaussian processes. Assuming $\alpha > 3/2$ corresponds to a mild smoothness condition on the embedded random object $\Phi(\mu)$. Such assumptions are classical and  are commonly used to control truncation errors in spectral approximations. We will clarify this point and add references in the revised version.
>
> > **"It would be good if the reason why the Hilbert-Schmidt norm difference is considered in this work, instead of other norms, e.g., the operator norm."**
>
> We use the Hilbert--Schmidt (HS) norm mainly because it is the natural norm induced by the inner product on operators. That said, our results can be interpreted in operator norm as well.
>
> First, since $\Vert A\Vert_{\mathrm{op}} \leq \Vert A\Vert_{\mathrm{HS}}$ for any compact operator $A$, any upper bound proved in HS norm (Theorem 3.3) immediately yields an operator-norm upper bound with the same rate.
>
> Second, for the minimax lower bound (Theorem 3.4), the proof in Appendix B is based on a two-hypothesis testing problem where we construct covariance operators separated in HS norm. If one measures separation in operator norm instead, the same pair of alternatives can be used, and for each embedding, the covariance operator can be written $\\Sigma^{(k)} = C s^{(k)}\\sum\\limits\_{i=1}^d f\_i\\otimes f\_i$ with $\Vert \\sum\\limits\_{i=1}^d f\_i\\otimes f\_i\Vert_{\mathrm{op}} = 1$. This yields the following operator-norm separation:
> $$
> \Vert \Sigma^{(1)} - \Sigma^{(2)}\Vert^2_{\mathrm{op}}  = C(s^{(1)} - s^{(2)})^2 = Cn^{-1}.
> $$
>
> Therefore, the separation hypothesis needed in Section B.3 remains valid with different constants. We will add a short remark in the revised version to clarify this point.
>
>
> ### References
>
> [1] Cazelles, E., Seguy, V., Bigot, J., Cuturi, M. and Papadakis, N., 2018. Geodesic PCA versus log-PCA of histograms in the Wasserstein space. SIAM Journal on Scientific Computing, 40(2).

---

> > ### Author Rebuttal · Reviewer_4que · 2026-04-01
> >
> > The authors answered my question adequately.

---

> > > ### Author Response · Authors · 2026-04-07
> > >
> > > Thank you for your positive answer. We have carefully addressed all of your comments and revised the manuscript accordingly (in particular, we have clarified the use of the operator norm as suggested.
> > >
> > > We noticed that your overall score remains unchanged, and we were wondering whether there might still be aspects of the paper that could be improved. If you have any additional suggestions or remaining concerns, we would be grateful if you could share them, as they would help us strengthen the paper.

---

### Official Review · Reviewer_dMP3 · 2026-03-12

**Soundness:** 3
**Presentation:** 4
**Significance:** 3
**Originality:** 3
**Overall Recommendation:** 5
**Confidence:** 2

**Summary:**

The work studies asymptotic behavior in covariance estimation and PCA as a function of the number of measures, the number of samples per measure, and the choice of embedding. The authors show that these two sources of error bound the covariance estimation error as well as the excess risk in PCA. The authors then shown empirically on generated data. Applied to three real datasets, authors qualitatively show that 2D PCA quickly converges to a stable representation with respect to number of samples.

**Compliance With Llm Reviewing Policy:**

Affirmed.

**Final Justification:**

This is a technically sound work with direct implications to questions of data sampling. I maintain my original score of accept.

**Key Questions For Authors:**

1. Using the paper’s example, if each person represents a measure, is it reasonable to assume that people of varying ages, genders, illnesses, etc are independent and drawn from the same base measure? Please discuss the validity of the iid mu_i assumption in practice, e.g. for cases where it is a natural assumption and for cases where it may be inappropriate to apply. This discussion will help clarify the utility and/or limitations of the theoretical results.
2. Could you empirically evaluate performance as a function of dataset dimension? E.g., for some embedding, fixed number of PCA dimensions, and fixed number of measures and samples (for both dense and sparse regimes), please evaluate covariance estimation error and PCA excess risk as the dataset dimension increases. This will provide a complementary analysis of your theoretical results in high-dimensional scenarios which are also of practical interest in when applying PCA.
3. Relatedly, can you perform an empirical analysis on PCA excess risk for fixed number of measures, samples, and dataset dimension over increasing number of PCA dimensions? This will test the bound in Theorem 3.7 when q is large.

**Limitations:**

yes

**Strengths And Weaknesses:**

Strengths: This paper provides rigorous theoretical analysis of a relevant estimation problem in a doubly asymptotic regime whose results have direct use in applied data analysis. In particular, the discussion in Sec. 3.3.1 highlights how the preceding theorems yield a principled way to select number of samples for PCA (given fixed number of measures), providing a solution to practitioners’ perennial question of how many data points to sample/collect. Real data evaluation on varied open source datasets shows robustness and helps with reproducibility. The paper is very well written and motivated, especially for such a technical work. Proofs are written cleanly with explanations at each step.

Weaknesses:
- The results are proved when mu_i are iid samples from a base measure mu. While I understand this condition on mu_i facilitates theoretical analysis, it is unclear if this is a justified assumption in real data scenarios.
- The paper focuses on PCA, a method often associated to dimension reduction and high-dimensional data, but provides very little analysis and evaluation regarding varying dimensions.
- A significant portion of empirical evaluation relies on Procrustes disparity which is insufficiently defined, e.g., what is ‘d’ (around line 380)? Fig. 4 is unlabeled so it is unclear which plot is mean and which is standard deviation.
- The authors use \lesssim notation throughout the text without definition. This notation seems rather nonstandard for this venue and may be confused by general readers.

---

> ### Author Rebuttal · Authors · 2026-03-30
>
> Thank you for your positive review and your detailed comments. We address each point below.
>
> > **"Please discuss the iid $\mu_i$ assumption in practice [...]"**
>
> The assumption that the measures $\mu_i$ are i.i.d copies of a base measure $\mu$ can be indeed misleading. A more precise formulation is to consider $\mathbb{P}\in\mathcal{P}(\mathcal{P}(\mathbb{R}^d))$, i.e a probability measure on the space of probability measures supported on $\mathbb{R}^d$ and to assume $\mu_i \overset{\mathrm{iid}}{\sim}\mathbb{P}$. Under this perspective, the i.i.d assumption does not mean that all individuals are identical, rather that they are independently drawn from a population distribution, *which can be heterogeneous*. The heterogeneity can be naturally encoded in $\mathbb{P}$. For instance, it can be modeled as a mixture
>
> $$
> \mathbb{P} = \sum\limits_{k=1}^K p_k \pi_k,
> $$
>
> where $p = (p_1,\cdots,p_K)$ are the mixture weights and each component $\pi_k$ corresponds to a subpopulation (e.g. varying ages, genders...).
>
> > **"A significant portion of empirical evaluation relies on Procrustes disparity which is insufficiently defined."**
>
> Let $Y_k^{(m)}\in\mathbb{R}^{n\times 2}$ denote the two-dimensional PCA representation obtained from the $k$-th random subsample of size $m$. We compute the Procrustes disparity $d_{kl}^{(m)} = d(Y^{(m)}_k , Y_l^{(m)})$ where
>
> $$
> d(Y_k^{(m)}, Y_l^{(m)}) = \min\limits_{R,s,t} \|Y_k^{(m)} - sY_l^{(m)}R-\mathbf{1} t^T\|^2_{F},
> $$
>
> where the minimum is taken over rotations $R\in\mathbb{R}^{2\times 2}$, scaling factors $s\in\mathbb{R}_+$, and translations $t\in\mathbb{R}^2$. Here $\|\cdot\|_F$ denotes the Frobenius norm and $\mathbf{1}\in\mathbb{R}^n$ is the vector of ones. The quantity $d$ measures the dissimilarity between the datasets $Y_k^{(m)}$ and $Y_l^{(m)}$ after optimal alignement. Regarding Figure 4, the solid line represents the mean Procrustes disparity $\overline{d}^{(m)}$ and the shaded region indicates standard deviation around the mean. We will clarify the definition of the Procrustes disparity and revise the caption of Figure 4.
>
> > **"Could you empirically evaluate performance as a function of dataset dimension ?"**
>
> We thank the reviewer for this insightful suggestion. We agree that understanding the impact of the ambient dimension $d$ is important, especially in light of the different expected behaviors for the considered embeddings. To address this point, we conducted an additional experiment where we study how the covariance estimation error and the PCA excess risk vary with the ambient dimension $d$ while keeping $n$ and $m$ fixed. For each $d$, we generate $n=100$ random Gaussian measures, each sampled through $m=100$ points, estimate their embeddings, and compute both the covariance estimation error and the PCA excess risk. Results are averaged over 10 repetitions.
>
> | d  | KME (mean ± std)×100 | LOT (mean ± std)×100 | SW (mean ± std)×100 |
> |----|----------------------|----------------------|---------------------|
> | 2  | 1.43 ± 0.31          | 2.4 ± 0.17           | 0.71 ± 0.12         |
> | 5  | 2.6 ± 0.32           | 11.94 ± 0.13         | 0.66 ± 0.15         |
> | 10 | 4.29 ± 0.32          | 34.72 ± 0.38         | 0.55 ± 0.05         |
> | 50 | 12.64 ± 0.78         | 235.64 ± 0.97        | 0.52 ± 0.03         |
>
> | d  | KME (mean ± std)×100 | LOT (mean ± std)×100 | SW (mean ± std)×100 |
> |----|----------------------|----------------------|---------------------|
> | 2  | 0.02 ± 0.03          | 0.07 ± 0.02          | 0.02 ± 0.01         |
> | 5  | 0.13 ± 0.12          | 0.38 ± 0.05          | 0.01 ± 0.01         |
> | 10 | 0.17 ± 0.11          | 0.71 ± 0.05          | 0.02 ± 0.01         |
> | 50 | 0.71 ± 0.11          | 0.93 ± 0.01          | 0.02 ± 0.01         |
>
> We observe a clear dependence on the dimension that is consistent with our findings. In particular, the LOT embedding ($r_m(\Phi) \asymp m^{-1/d}$) exhibits a significant degradation as $d$ increases, while KME and SW ($r_m(\Phi) \asymp m^{-1/2}$) remain essentially stable.
>
> > **"Can you perform an empirical analysis on PCA excess risk [...] over increasing number of PCA dimensions ?"**
>
> Following the reviewer's suggestion, we performed an additional experiment to study the behavious of the PCA excess risk as a function of the number of components $q$, while keeping $n$, $m$ and $d$ fixed. We report the results in the following table.
>
> | q  | KME (mean ± std)×100 | LOT (mean ± std)×100 | SW (mean ± std)×100 |
> |----|----------------------|----------------------|---------------------|
> | 1  | 16.23 ± 11.66        | 4.13 ± 0.11          | 0.13 ± 0.07         |
> | 2  | 24.03 ± 8.57         | 8.28 ± 0.15          | 0.44 ± 0.23         |
> | 5  | 49.22 ± 8.39         | 20.20 ± 0.32         | 2.31 ± 0.19         |
> | 10 | 53.79 ± 8.54         | 39.09 ± 0.47         | 7.91 ± 0.00         |
>
> We observe in all cases that the excess risk grows with $q$, which was expected from Theorem 3.7.

---

> > ### Author Rebuttal · Reviewer_dMP3 · 2026-04-02
> >
> > I thank the authors for their comprehensive response.

---

### Decision · Program_Chairs · 2026-04-30

**Decision:**

Accept (regular)

**Comment:**

Reviewers are generally positive about this work. Some of the key points raised are as follows.

Strengths:
- Relevant problem
- Theory gives actual practical guidance on how many points to collect
- Method is evaluated on a comprehensive set of datasets
- Work reveals interesting phase transition behavior
- Reviewers described paper as well-written

Limitations:
- Some mismatch between theory and experiments
- Some minor writing issues, such as \lessim, while reasonably clear from context, not being precisely defined

I think on balance of these the paper is ready to be accepted, given the issues worth correcting are minor enough that they can be corrected at the camera ready.